

# Advanced characterization of aerosol properties from measurements of spectral optical depth using the GRASP algorithm.

Benjamin Torres[1,2], Oleg Dubovik[1], David Fuertes[1,2], Gregory Schuster[3], Victoria Eugenia Cachorro[4], Tatsiana Lapyonok[1], Philippe Goloub[1], Luc Blarel[1], Africa Barreto[5], Marc Mallet[6], Carlos Toledano[4], and Didier Tanré[1]

[1]Laboratoire d'Optique Amosphérique, Université des Sciences et Technologies de Lille, Villeneuve d'Ascq, France.
[2]GRASP-SAS, Remote sensing developments, Université des Sciences et Technologies de Lille, Villeneuve d'Ascq, France.
[3]NASA Langley Research Center, Hampton, Virginia, USA.
[4]Group of Atmospheric Optics, Valladolid University, Valladolid, Spain
[5]Izaña Atmospheric Research Center, Spanish Meteorological Agency, Tenerife, Spain.
[6]CNRM UMR 3589, Météo-France/CNRS, Toulouse, France

*Correspondence to:* Dr. Benjamin Torres
(benjamin.torres@grasp-sas.com)

**Abstract.** This study evaluates the potential of using aerosol optical depth ($\tau_a$) measurements to characterize the microphysical and optical properties of atmospheric aerosols. With this aim, we used the recently developed GRASP (Generalized Retrieval of Aerosol and Surface Properties) code for numerical testing of six different aerosol models with three different aerosol loads. We found that bimodal log-normal size distributions serve as useful input assumptions, especially when the measurements have

5  inadequate spectral coverage and/or limited accuracy, such as lunar photometry. The direct numerical simulations indicate that the GRASP-AOD retrieval provides modal aerosol optical depths (fine and coarse) to within 0.01 of the input values. The retrieval of the fine mode radius, width, and volume concentration is stable and precise if the real part of the refractive index is known. The coarse mode properties are less accurate, but they are significantly improved when additional a priori information is available. In addition to these numerical studies, we used optical depth observations at six AERONET locations

10  to validate our results with the standard AERONET inversion products. Differences in the fine mode volume median radii for the GRASP-AOD and AERONET inversions are less than 0.02 $\mu$m at sites dominated by the fine mode for all cases, although they are typically less than 0.01 $\mu$m when $\tau_a(440) > 0.3$. The comparison of the coarse mode volume median radii shows larger differences than the fine mode at the same sites, with values typically between 0.2-0.3 $\mu$m. The comparison of coarse mode volume median radii between GRASP-AOD and AERONET improves for sites dominated by desert dust

15  aerosol, with differences of less than 0.2 $\mu$m in most cases. The retrieved values of the fine-mode $\tau_a(500)$ using GRASP-AOD are generally between those values obtained by the standard AERONET inversion and the values obtained by the advance AERONET Spectral Deconvolution Algorithm (SDA), with differences typically lower than 0.02 between GRASP-AOD and both algorithms. Finally, we present some examples of application of GRASP-AOD inversion using moon-photometry and the airborne PLASMA sun-photometer during ChArMEx summer 2013 campaign in the western Mediterranean.



## 1 Introduction

Aerosol optical depth ($\tau_a$) measurements that are complemented with angular radiance measurements are routinely used by the scientific community to infer the microphysical and optical properties of atmospheric aerosols (e.g., the Aerosol Robotic Network, or AERONET). Unfortunately, the availability of suitable $\tau_a$ measurements far exceeds the availability of suitable

radiance scans, mainly because of partly cloudy conditions and scattering angle limitations. Additionally, nighttime $\tau_a$ measurements are becoming increasingly available through lunar photometry, while nighttime radiance scans are useless. Thus, it is desirable to derive meaningful aerosol information from spectral optical depth measurements when complementary radiance measurements are not available.

The goal of this work is to evaluate and demonstrate the full potential of using spectral optical depth measurement for

characterizing detailed properties of aerosol. Indeed, the magnitude and spectral dependence of optical depth is known to be related to amount and size distribution of aerosol particles. Therefore, ground-based observations of solar radiation attenuation was one on the first type of measurements analyzed in pioneering studies devoted to atmospheric remote sensing algorithms (e.g. see Angstrom, 1929, 1961; Yamamoto and Tanaka, 1969; Quenzel, 1970; Grassl, 1971; King et al., 1978). However, in the last few decades, the focus of remote sensing retrieval development shifted towards analysis of more complex observations:

angular and polarimetric properties of transmitted and reflected diffuse radiation. In principle, such observations have high sensitivity that allows complete and accurate characterization of the aerosol features. However, modeling and interpretation of the diffuse radiation is significantly more complex than the analysis of the direct Sun beam. Therefore, the complexity and efficiency of the retrieval algorithms have significantly improved compared to the those originally developed for the interpretation of aerosol optical depth. At the same time, in a number of practical situations the interpretation of $\tau_a$ data alone

remain interesting for the community. Moreover, due to evolution of ground-based instrumentation and infrastructure, the amount of available observations constantly increases. Therefore, in current study, we have decided to revisit this problem and provide a complete analysis and illustration of $\tau_a$ inversion using state-of-the-art retrieval approaches and software.

With this purpose, we make use of the recently developed GRASP (Generalized Retrieval of Aerosol and Surface Properties, see Dubovik et al. 2014) algorithm and software. The code implements statistically optimized fitting of diverse observations

using multi-term least square method (LSM) concept (e.g. see Dubovik and King, 2000; Dubovik, 2004). Correspondingly the retrieval is organized as a solution search in continuous space of solutions without traditional use of pre-calculated look-up tables. The GRASP algorithm is highly versatile and can be applied to a large variety of remote sensing measurements (e.g. sun-photometer, lidars, satellite images, etc). The GRASP concept, originated from the Dubovik and King (2000) algorithm, has been successfully used for 15 years to process observations of AErosol RObothic NETwork (AERONET) of ground-based

sun/sky-radiometers (Holben et al., 1998) engendering a large number of worldwide climatologies (to cite some Dubovik et al., 2002a; Smirnov et al., 2002a, b; Eck et al., 2005, 2010; Giles et al., 2012; Toledano et al., 2012). During this period the algorithm has certainly evolved and several useful modifications were realized (Dubovik et al., 2000, 2002b, 2006; Li et al., 2009). The GRASP development inherited of all these retrieval advances.



In addition, the GRASP algorithm has a highly flexible forward model that makes it a convenient tool for sensitivity and tuning studies (Dubovik et al., 2014). This flexibility was one of motivating factors for performing these studies. Indeed, the lack of scattering information in $\tau_a$ observation obliges the retrieval developer to search for an optimum aerosol model and an adequate set of a priori constraints (parameterization of size distribution, etc.). For example, this study suggests that reasonable results can be obtained by approximating size distributions as bimodal log-normals, which can be described with only 6 parameters (volume median radius ($r_{V_i}$), standard deviation ($\sigma_{V_i}$) and volume concentration ($C_{V_i}$) for fine and coarse mode) instead of using binned size distributions (22 bins in the case of AERONET standard inversion). Moreover, we assume that the complex refractive index is known. Both assumptions and their consequences are discussed in detail in the sensitivity analysis presented in section 3.

A practical motivation for the present study is the large amount of "optical depth only" measurements that exist in the ground-based networks. For instance, AERONET includes around forty direct Sun measurements per day in its standardized sequence of measurements (in cloudless conditions); only about eight of these sequences are coincident with sky-radiance measurements that are suitable as input to the AERONET inversion code[1]. Hence, there is a large amount of data containing only direct Sun measurements which are not used apart from the characterization of the aerosol load. Moreover, many AERONET sites are plagued by several months of partial cloudiness (especially in wintertime). In these situations, there are no angular measurements of sky-radiance suitable for the retrieval of detailed aerosol properties, and only a few direct Sun measurements are available. As a result, the aerosol loads are the only data reported at many sites, and further information about aerosol properties is absent for extended time periods.

Similar to AERONET, the $\tau_a$ data are often the most common measurement data provided by other ground-based networks as well , such as the SKYNET-PREDE network of sky radiometers (Takamura and Nakajima, 2004). Furthermore, some networks provide only $\tau_a$ data. For example, the Maritime Aerosol Network (Smirnov et al., 2009) provides $\tau_a$ measurements over the oceans taken by Microtops handheld sun-photometers. This network is federated with AERONET and uses the same calibration procedure and data processing. Another example is the Global Atmospheric Watch GAW-PFR (Wehrli, 2005), operated by the World Optical Depth Research and Calibration Center (WORCC), which provides only quality-assured spectral $\tau_a$ data, albeit with nearly continuous temporal coverage. Outside of the standardized networks, many other instruments offer only aerosol optical depth measurements at similar spectral ranges: airborne photometers (AATS-14, as in Schmid et al. 2003, PLASMA by Karol et al. 2013 and others), spectroradiometers (Cachorro and De Frutos 1994, Martinez-Lozano et al. 1999), etc.

Night measurements, which have been largely developed in recent years, represent another interesting example where $\tau_a$ data are typically the only information available about the columnar aerosol properties (apart from the backscattering provided by collocated lidars). These night observations are of great interest, especially at polar regions where they offer a solution to infer aerosol load during winter months. There are two main groups depending on the light source: star-photometry (Herber et al.,

---

[1]In reality there are around sixteen sky measurements per day: eight almucantar and eight principal plane. We have counted eight set of measurements since AERONET products includes only almucantar inversions to guarantee the quality of the retrievals. More information about AERONET standardized sequence of measurements can be found in http://aeronet.gsfc.nasa.gov/new_web/Documents/AERONETcriteria_final1_excerpt.pdf and about principal plane retrievals in Torres et al. (2014).





2002; Perez-Ramirez et al., 2008; Perez-Ramirez et al., 2011; Baibakov et al., 2015), and more recently, lunar photometry (Barreto et al., 2013, 2016). In some conditions, these techniques show the same accuracy as regular Sun measurements (Perez-Ramirez et al., 2008; Barreto et al., 2013), although the absence of worldwide networks with well-defined standardized protocols limits the diffusion of these results.

Aerosol optical depth measurements provide key information about climate effects, tropospheric corrections for remote sensing, visibility, etc. (Dubovik et al., 2002a). Moreover, even though $\tau_a$ refers to the total load of the whole atmospheric column, several studies have demonstrated a strong correlation of $\tau_a$ with the near-surface atmospheric concentration of particulate matter, $PM_x$ (where $x$ is the upper cut-off aerodynamic diameter, typically 2.5 or 10 $\mu m$, and $PM_x$ is mass per unit volume of air in $\mu g/cm^3$ ). $PM_x$ has a detrimental affect on human health, and is commonly monitored at the ground level in worldwide

networks (Rohen et al., 2011; Bennouna et al., 2016). For these reasons, many studies during recent decades use $\tau_a$ retrieved by satellite sensors to estimate and forecast the surface $PM_x$ (Kacenelenbogen et al., 2006; Koelemeijer et al., 2006; Vidot et al., 2007; Rohen et al., 2011).

Nevertheless, the total aerosol loading given by $\tau_a(\lambda)$ itself is not enough for complete climate studies. For instance, information about the microphysical properties of aerosols (i.e. size, shape and chemical composition) is needed to quantify the

aforementioned climate effect produced by atmospheric aerosols (Mishchenko et al., 2007). However, numerous studies have demonstrated that just a set of $\tau_a$ measurements could be used for deriving more detailed information about aerosol apart from a characterization of the quantity. For example, the spectral dependence, typically given by the Ångström exponent (Angstrom, 1961), is commonly used as an indicator ¨ of the predominant aerosol size (Reid et al., 1999; Eck et al., 1999, 2001; Kim et al., 2004; Toledano et al., 2007, 2009; Kaskaoutis et al., 2009; Kumar et al., 2014, etc.).

The pioneering study of King et al. (1978) showed the capacity of retrieving aerosol size distributions by modifying initial Junge distributions (Junge, 1955) through a numerical inversion. The algorithm has been used extensively for decades (for instance, Cachorro and De Frutos 1994; Liousse et al. 1995; Cachorro 1998; Martinez-Lozano et al. 1999; and Wang et al. 2006 applied the retrieval to $\tau_a$ data, while Elias et al. (1998, 2000), Vermeulen et al. (2000) combined the retrieval with radiance measurements).

O'Neill et al. (2003) developed the Spectral Deconvolution Algorithm (SDA) to discriminate the extinctions of the fine and coarse modes. That study established a mathematical relation between the ratio fine/coarse extinction with the Ångström exponent and its fine and coarse mode components. Then, by applying several approximations (fixing the coarse Ångström exponent to a low value and neglecting its derivatives) and using some empirical relations (between the fine Ångström exponent and its derivatives), the ratio fine/coarse extinction is derived (more information also in O'Neill et al., 2001a, b). The algorithm

is integrated as a part of the AERONET procedures: the value of the aerosol optical depth of fine and coarse mode at 500 nm is retrieved from every aerosol optical depth measurements and provided as a standard product of the network (full description in http://aeronet.gsfc.nasa.gov/new_web/PDF/tauf_tauc_technical_memo1.pdf).

Several recent studies retrieve the effective radius and total aerosol concentration from aerosol optical depth measurements by using linear estimation techniques (LET), which have been largely developed for the retrieval of lidar measurements

(Veselovskii et al., 2012, 2013). Kazadzis et al. (2014) applied LET retrievals to direct Sun measurements from GAW-PFR and




AERONET-CIMEL photometers. Pérez-Ramírez et al. (2015) extended the use of LET to star photometry measurements. They also applied LET technique to sun-photometer data from several AERONET sites influenced by different aerosol types. The retrieved effective radius and aerosol concentrations were afterwards compared to those retrieved by the operational AERONET code. The GRASP-AOD inversion, illustrated here, also allows the derivation of secondary products, such as the extinction

of fine/coarse mode and the effective radius of the retrieved size distribution. We include a special emphasis on $\tau_f(500)$ (fine mode extinction at 500 nm) in this study, since it is directly comparable to the SDA product. However, other products are systematically evaluated (such as the effective radius).

The analysis presented here is focused on $\tau_a$ measurements in the spectral range of the CIMEL sun-photometers that are used in the AERONET project. This approach allowed us to compare the results obtained by the GRASP-AOD inversion with

the standard AERONET inversion. Consequently, the simulation tests proposed in section 3 are done using the wavelengths in AERONET network, and the main analysis with real measurements in section 4 corresponds to AERONET data. However, we emphasize that the code is not restricted to AERONET measurements, and some example applications of the code to moon photometer data are also shown in section 4. Finally we present some retrievals from data obtained at different heights during the ChArMEx campaign (Mallet et al., 2016) with the new airborne sun-tracking photometer PLASMA (Karol et al., 2013)

fully designed at the LOA (Laboratoire d'Optique Atmosphérique, University of Lille) .

## 2    Inversion strategy (GRASP-AOD).

As commented in the introduction, we make use of the recently developed GRASP (Generalized Retrieval of Aerosol and Surface Properties, see Dubovik et al. 2014) algorithm and software (more information and a free version of the code can be obtained in http://www.grasp-open.com/) to assess the potential of spectral aerosol optical depth measurement for characteriz-

ing aerosol properties. The main concept of GRASP is the multi-term least square method (LSM) (for details of the inversion methodology see Dubovik and King 2000; and also Dubovik 2004) and is designed to be applied to a broad number of remote sensing measurements (e.g. sun-photometer, lidars, satellite images, etc).

The multiterm LSM method solves the following system of equations

$$\begin{cases} \mathbf{f}^* & = \mathbf{f}(\mathbf{a}) + \Delta_{\mathbf{f}} \\ \mathbf{0}^* & = \mathbf{G}\mathbf{a} + \Delta_{\mathbf{g}} \\ \mathbf{a}^* & = \mathbf{a} + \Delta_{\mathbf{a}} \end{cases} \tag{1}$$

The first equation in (1) describes the physical relation between the set of measurements $\mathbf{f}$ and the vector of unknowns $\mathbf{a}$. The symbols $\Delta_{\mathbf{f}}$ denote the uncertainty in the real measurements $\mathbf{f}^*$. Note that in our case, the set of measurements is only the





spectral aerosol optical depths (defined in Eq. (A2)). In GRASP-code the atmospheric aerosol is modeled as an ensemble of randomly oriented spheroids and the aerosol optical depth is modeled as follows:

$$\tau_a(\lambda) = \int\limits_{\ln \epsilon_{\min}}^{\ln \epsilon_{\max}} \int\limits_{\ln r_{\min}}^{\ln r_{\max}} \frac{C_{ext}(\lambda, n, k, r)}{\nu(r)}$$
$$\cdot \frac{dn(\epsilon)}{d\ln(\epsilon)} \frac{dV(r)}{d\ln(r)} d\ln(r) d\ln(\epsilon), \tag{2}$$

where $C_{ext}$ is the cross section of extinction, $\lambda$ - wavelength, n and k - real and imaginary parts of the refractive index. The

aerosols are approximated as spheroids (Mishchenko et al., 2002) with $\epsilon$ being the axis ratio ($\epsilon = a/b$, $a$ – axis of spheroid rotational symmetry, b – axis perpendicular to the axis of spheroid rotational symmetry) and $r$ – radius of the equivalent sphere. As discussed by Mishchenko et al. (1997), the usage of $r$ and $\epsilon$ is convenient for separating the effect of particle shape and size in analysis of aerosol mixture light scattering. Then the functions $\frac{dV(r)}{d\ln(r)}$ and $\frac{dn(\epsilon)}{d\ln(\epsilon)}$ denotes the volume distribution of the spheroids (for the total column) and the number particle shape (axis ratio) distributions accordingly.

Dubovik et al. (2006) have demonstrated that the particle shape distribution $\left(\frac{dn(\epsilon)}{d\ln(\epsilon)}\right)$ for the non-spherical fraction of any tropospheric aerosol can be approximated as constant. This assumption simplifies Eq. (2) and the aerosol extinction is calculated for the retrieval as a mixture of spherical and non-spherical fractions. Moreover, in order to perform fast and accurate calculations, the integrals are replaced by sums of pre-calculated kernels as follows:

$$\tau_a(\lambda) = \tau_{\mathrm{sph}}(\lambda) + \tau_{\mathrm{nons}}(\lambda) = \sum_{i=1,...,N_r} \left( C_{\mathrm{sph}} \mathbf{K}_\tau^{sph}(\lambda, k, n, r_i) \right.$$
$$\left. + (1 - C_{\mathrm{sph}}) \mathbf{K}_\tau^{nons}(\lambda, k, n, r_i) \right) \frac{dV(r_i)}{d\ln(r)} \tag{3}$$

where $C_{\mathrm{sph}}$ is the fraction of the spherical particles and $\mathbf{K}_\tau^{sph}$ and $\mathbf{K}_\tau^{nons}$ are the kernels for spherical and non-spherical particles respectively. The complete information about the forward model and the detailed calculation of the kernels can be gained in Dubovik et al. (2006, 2011).

The second equation in (1) represents a priori smoothness constraints on the retrieved characteristics. They are typically applied to eliminate unrealistic strongly oscillating dependencies in the retrieved characteristic. Specifically, the study of Dubovik

and King (2000) showed that assuming zeros ($\mathbf{0}^*$ – zero vector) for the derivatives of retrieved aerosol characteristics allows the elimination of strongly oscillating solutions with high derivatives. The matrix $\mathbf{G}$ is composed of coefficients allowing the numerical estimates of derivatives of function $y(x)$ using discrete values $a_i = y(x_i)$. These constraints are normally used to smooth the retrieved size distribution and the spectral dependencies of refractive index. The symbol $\Delta_{\mathbf{g}}$ accounts for the uncertainties in the a priori constraints.

However, here in the so-called GRASP-AOD application, the refractive index is assumed as known and the size distribution is characterized as bimodal log-normal being defined by six independent parameters: volume median radius ($r_{Vi}[\mu m]$), geometric standard deviation ($\sigma_{Vi}$) and particle volume concentration ($C_{Vi}[\mu m^3/\mu m^2]$), with $i = f, c$ for the each mode. Therefore, the





second equation is not used in this particular application of GRASP code. It should be commented here that we initially tested binned size distributions but we rapidly observed that strong smoothness constraints were required in order to assure realistic retrievals. Hence, we decided to model the size distribution in terms of log-normal functions. Note that log-normal approximations to represent size distributions are largely used by many physical models (e.g. see Whitby, 1978; Shettle and Fenn, 1979; Koepke et al., 1997; Hess et al., 1998, etc.) and this aerosol representation generally agrees well with observations (Tanre et al., 1988; Remer and Kaufman, 1998; Dubovik et al., 2002a). Therefore, applications with a need of representing aerosol by limited number of parameters naturally choose this concept for modeling size distribution. In particular, bimodal lognormal functions are fully sufficient to interpret variability of aerosol optical depth (to cite some Eck et al., 1999; O'Neill et al., 2001a, b; Schuster et al., 2006, etc).

The last equation in (1) shows the possibility of using a priori constraints on actual values of any retrieved parameter $\mathbf{a}_i$ and $\mathbf{a}^*$ is the vector of a priori estimates of $\mathbf{a}_i$. The symbol $\Delta_\mathbf{a}$ represents the uncertainty in the a priori constraints of $\mathbf{a}^*$. For optimized accounting of those uncertainties, the solution of the system defined in (1) is given by the minimization of the following quadratic form:

$$
2\Psi(\mathbf{a}) = (\mathbf{f}(\mathbf{a}) - \mathbf{f}^*)^\mathrm{T} \mathbf{W_f}^{-1} (\mathbf{f}(\mathbf{a}) - \mathbf{f}^*)
$$
$$
+ \gamma_g \mathbf{a}^\mathrm{T} \mathbf{G}^\mathrm{T} \mathbf{W_g}^{-1} \mathbf{G} \mathbf{a} + \gamma_a (\mathbf{a} - \mathbf{a}^*)^\mathrm{T} \mathbf{W_a}^{-1} (\mathbf{a} - \mathbf{a}^*)' \tag{4}
$$

where the weighting matrices $\mathbf{W}$ and the Lagrange parameters $\gamma$ are defined as follows:

$$
\mathbf{W_f} = \frac{1}{\epsilon_\mathbf{f}^2} \mathbf{C}_f; \qquad \mathbf{W_g} = \frac{1}{\epsilon_\mathbf{g}^2} \mathbf{C}_g; \qquad \mathbf{W_a} = \frac{1}{\epsilon_\mathbf{a}^2} \mathbf{C}_a;
$$
$$
\gamma_g = \frac{\epsilon_\mathbf{f}^2}{\epsilon_\mathbf{g}^2}; \qquad \gamma_a = \frac{\epsilon_\mathbf{f}^2}{\epsilon_\mathbf{a}^2} \tag{5}
$$

where $\epsilon_\mathbf{f}^2$ and $\epsilon_\mathbf{a}^2$ are the first diagonal elements of the corresponding covariance matrices $\mathbf{C_f}$ and $\mathbf{C_a}$ respectively. During the retrieval, we assume that all input data have log-normal error distribution (Dubovik and King, 2000; Dubovik, 2004), which means that the measurements and the retrieved parameters are used in logarithm space.

Finally, it should be indicated that one of the recent success of GRASP code has been the easy adaptation of the multi-pixel retrieval concept in the methodology proposed by multiterm LSM (Eq. (1)). The strategy was developed by Dubovik et al. (2011) in order to overcome the difficulties related to the limited information of the satellite observations over a single pixel. The multi-pixel retrieval regime takes advantage of known limitations on spatial and temporal evolution in both aerosol and surface properties. Specifically, a large group of pixels are inverted simultaneously, using a priori constraints on the temporal and spatial variability of the retrieved parameters. The concept has been expanded in the latest version of the GRASP-algorithm, where the measurements can also belong to different remote sensing instruments. The present study focuses on the potential of a single set of aerosol optical depth measurements, so that the multi-pixel inversion will not be used in this work. Nevertheless, in a number of recent promising applications the multi-pixel approach has been used for synergy retrievals when $\tau_a$ observations were combined with other co-located but not co-incident measurements (Lopatin et al., 2013, 2016).



## 3   Sensitivity studies.

The description of any algorithm must answer several inherent questions regarding the retrieval: stability of the inversion, confidence in the retrieved products, and dependence up on the a-priori assumptions, etc. In our particular case, the main challenges are to (a) identify the reliability of the six parameters which describe the bimodal log-normal size distribution (and

of the secondary products derived from them), (b) check the effects of possible errors in the aerosol optical depth measurements, and finally, (c) analyze the consequences of assuming the refractive index as "known" during the retrieval process.

Several simulation tests are considered in this section to address these points. First, a self-consistency analysis, including multiple variations of the initial guess, will be carried out to check the stability of the retrieval. Next, simulated errors in the aerosol optical depth measurements will be introduced to determine the ramifications on the retrieved properties. The last study

will include bias in the refractive index assumptions.

### 3.1   Aerosol models

Our database for carrying out the simulations is based upon the Dubovik et al. (2002a) climatology, which utilizes real aerosol retrievals from AERONET observations at several key sites. The climatological size distributions given in this study use bi-modal log-normal approximations (instead of 22-bin size distributions) which simplifies the analysis of our simulation tests.

Specifically, we selected aerosol properties from six different sites to carry out the simulations. Three of the sites are dominated by fine mode aerosols (from less to more absorbing): Goddard Space Flight Center (Maryland-USA), which represents urban non-absorbing aerosol, Mexico-City (Mexico) representing urban absorbing aerosol, and Mongu (Zambia) which corresponds to biomass burning aerosol. Additionally, we have selected Bahrain (Bahrain) and Solar Village (Saudi Arabia) as examples of mixed desert dust and pure desert dust, respectively. Finally, we have used the aerosol properties at Lanai (Hawaii-

USA) as an example of maritime aerosol. For each example, we have considered three different aerosol loads: $\tau_a(440) = 0.3$, 0.6 and 0.9. The $\tau_a(440) = 0.9$ case is omitted for Lanai, however, since this case would be excessively unrealistic (given the typical values observed at that site).

The aerosol properties of all the examples considered here are represented in table 1. The first parameter is the reference value of the aerosol optical depth. The rest of the parameters are derived from the expressions in Dubovik et al. (2002a). The

following parameters in table 1 are used to describe the bimodal log-normal size distribution: volume median radius ($r_{Vi}[\mu m]$), geometric standard deviation ($\sigma_{Vi}$) and particle volume concentration ($C_{Vi}[\mu m^3/\mu m^2]$), for fine and coarse mode. The rest of the inputs are the sphericity parameter and the refractive index. The former is taken as 0 for the two cases of desert dust (all the particles are considered to be non-spherical) and as 100 for the rest of the cases (considering all the particles as spheres). Note that henceforth, we will refer the six aerosol models as: GSFC (Goddard Space Flight Center), MEXI (Mexico-City), ZAMB

(Mongu), BAHR (Bahrain), SOLV (Solar Village) and LANA (Lanai). In the same way, we will use different index for the different aerosol loads: index "1", for $\tau_a(440) = 0.3$, index "2", for $\tau_a(440) = 0.6$ and finally index "3" for $\tau_a(440) = 0.9$.

Figure 1 illustrates the size distributions created from the parameters described in table 1. Each aerosol load for every case is represented by different lines: the cases with $\tau_a(440) = 0.3$ by dashed line, the cases with $\tau_a(440) = 0.6$ by solid line, and





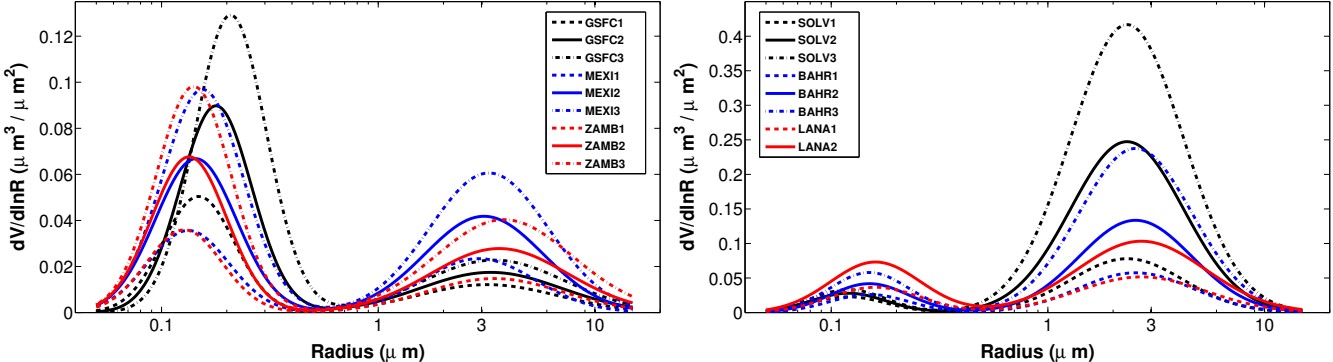

**Figure 1.** Representation of the aerosol size distributions used as examples in the simulation analysis (described in table 1). Aerosol models with a predominance of the fine mode are plotted in the left (subfigure a): GSFC (black lines), MEXI (blue lines) and ZAMB (red lines). The rest of the cases are plotted in the right (subfigure b): SOLV (black lines), BAHR (blue lines) and LANA (red lines). In both subfigures the cases with $\tau_a(440) = 0.3$ (index "1") are represented by dashed line, the cases with $\tau_a(440) = 0.6$ (index "2") by solid line, and cases with $\tau_a(440) = 0.9$ (index "3") by dashed-dotted line.

cases with $\tau_a(440) = 0.9$ by dashed-dotted line. The three cases with a predominance of fine mode are plotted in the left panel. The two desert dust examples and the case of maritime aerosol are represented in the right subfigure.

In the cases with a predominance of fine mode, left panel of figure 1, the mode radii vary with AOD for both modes. The feature was pointed in the climatology study from Dubovik et al. (2002a) and depicted in the summarizing formulas (table 1 in Dubovik et al. 2002a) from that study, which are the basis of our aerosol models here. The other three aerosol cases, which are represented in the right panel, do not present this property as can be gained from the figure (or directly from the values in table 1).

The simulated aerosol optical depth values obtained by running the GRASP-forward module with the values from table 1 are shown in table 2. We have also included the resulting Ångström exponent, which is computed as a linear regression of $\tau_a(440)$, $\tau_a(675)$ and $\tau_a(870)$. In the last two columns, the values of the fine mode aerosol optical depth at 500 nm and the effective radius are depicted since they will be analyzed in the different sensitivity studies (as derived secondary products).

In order to provide a graphical representation of the tendencies from the $\tau_a$ simulated values, they are represented for the cases with $\tau_a(440) = 0.3$ and 0.6, in figure 2 in logarithmic scale. In the figures, we observe a linear relationship with slight curvature for all of the sites. This curvature is negative for aerosol size distributions dominated by fine mode aerosols, and positive for the desert dust cases (especially for SOLV). This is consistent with previous works (Kaufman, 1993; Eck et al., 1999, 2001; Reid et al., 1999; Schuster et al., 2006).





**Figure 2.** Aerosol optical depth values of the six aerosol models represented in logarithmic scale for the cases with $\tau_a(440) = 0.3$ (on the left) and $\tau_a(440) = 0.6$ (on the right).

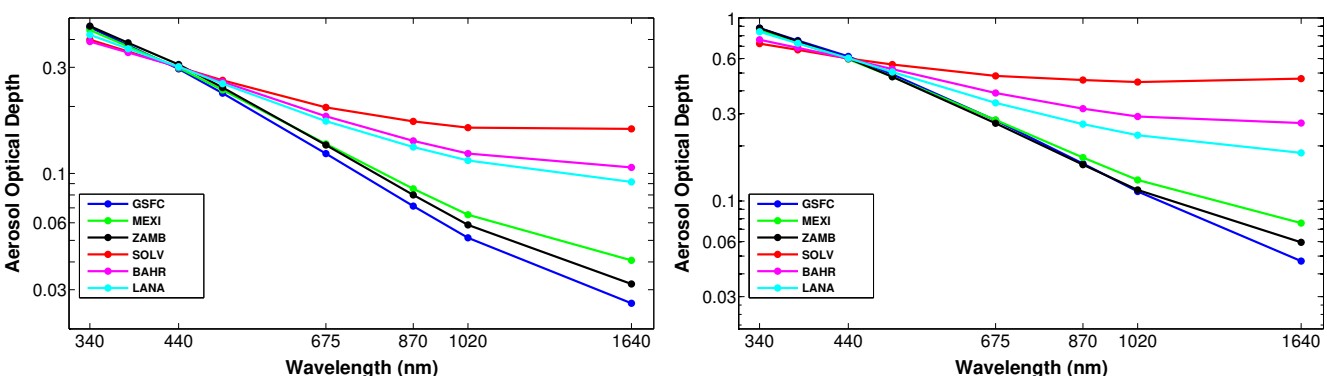

**Figure 3.** Methodology diagram followed to make different sensitivity "tests" on GRASP-AOD code.

## 3.2 Self-consistency analysis

Our strategy for the simulation tests has been adapted from previous work (Dubovik et al., 2000; Torres et al., 2014). A general scheme of the procedure is drawn in figure 3. The methodology consists basically of inverting the aerosol optical depths from table 2 (which were obtained by running the forward module using the aerosol properties described in table 1), but introducing





some modifications in order to "test" the sensitivity of the code regarding the different aspects already mentioned: multiple initial guesses, variation of the refractive index, and uncertainty in the aerosol optical depth data. Nevertheless, we first present a self-consistency study where we do not make any modification during the inversion process.

The first 6 columns of Table 3 represent the differences between the output of the self-consistency analysis (obtained from the inversion of the $\tau_a$ values from table 2 and considering the values of the refractive indices also from table 2) and the original values of the size distribution (table 1). In the last two columns we show the differences in the selected secondary products: $\tau_f(500)$ and $r_{eff}$ (theoretical values in table 2).

The capacity of the GRASP-AOD inversion to discriminate fine and coarse mode aerosol optical depths is one of the most important results that can be gained from table 3: differences between the reference and retrieved values of $\tau_f(500)$ are always less than $0.01$. This result is even better for those cases with a predominant fine mode, where a maximum difference of $0.002$ is obtained.

The retrieved parameters that characterized the fine mode also show a good agreement with the reference values (from table 1). Thus, the maximum difference observed for $r_{V_f}$ is $0.009 \mu m$ (LANA2). Once again the comparison is better for the cases with a prevailing fine mode with a maximum difference of $0.003 \ \mu m$. The differences in $\sigma_{V_f}$ do not exceed $0.02$ for the aerosol cases with a predominance of fine mode, while they are a bit higher in the rest of the cases: up to $0.03$ for desert dust cases and the extreme case of maritime aerosol with a maximum difference of $0.06$ (LANA2). Finally, the divergences in the concentration ($C_{V_f}$) are under $0.002 \ \mu m^3/\mu m^2$ in all the examples except in LANA2 where a maximum difference of $0.005 \ \mu m^3/\mu m^2$ is obtained.

The retrieval of the coarse mode is less accurate than the retrieval of the fine mode. We expected this result, since the wavelengths used in this study are less sensitive to the radii in the range of the coarse mode than to those in the range of the fine mode (see, for instance, the discussion about extinction efficiency and size parameter in Chapter 2 of Lenoble et al. 2013; or in Chapter 3 of Coulson 1988). Differences in $r_{V_c}$ are up to $0.4 \ \mu m$ for the cases with fine mode predominance and a bit smaller for desert dust and maritime aerosol, up to $0.3 \ \mu m$, since those cases have larger coarse mode aerosol optical depth and therefore more information. The discrepancies in the concentration ($C_{V_c}$) are at maximum $0.006 \ \mu m^3/\mu m^2$ (ZAMB3) for the cases with fine mode predominance and $0.04 \ \mu m^3/\mu m^2$ (SOLV2) for the cases with a prevailing coarse mode. For both cases these differences represent $10\%$.

We should point out here the strong connection between the retrievals of $r_{V_c}$ and $C_{V_c}$. Those aerosol examples with the best characterization of $r_{V_c}$ correspond as well to those with the best characterization of $C_{V_c}$: GSFC in the case of fine mode predominance and BAHR for the cases with a prevailing coarse mode. Those cases with an overestimation in the volume median radius also display an overestimation in the concentration: MEXI and SOLV. Note that for the radius range of the coarse mode, the extinction efficiency diminishes as the radius grows. Therefore an overestimation of $r_{V_c}$ needs to be optically compensated increasing the coarse mode concentration. In the same way, the cases that present an underestimation in the volume median radius show an underestimation in the concentration: ZAMB and LANA.

Finally, the effective radius is also computed and compared with the reference values in table 3. Note that this parameter is not directly derived from the inversion and is computed from the retrieved values of the bimodal log-normal size distribution.




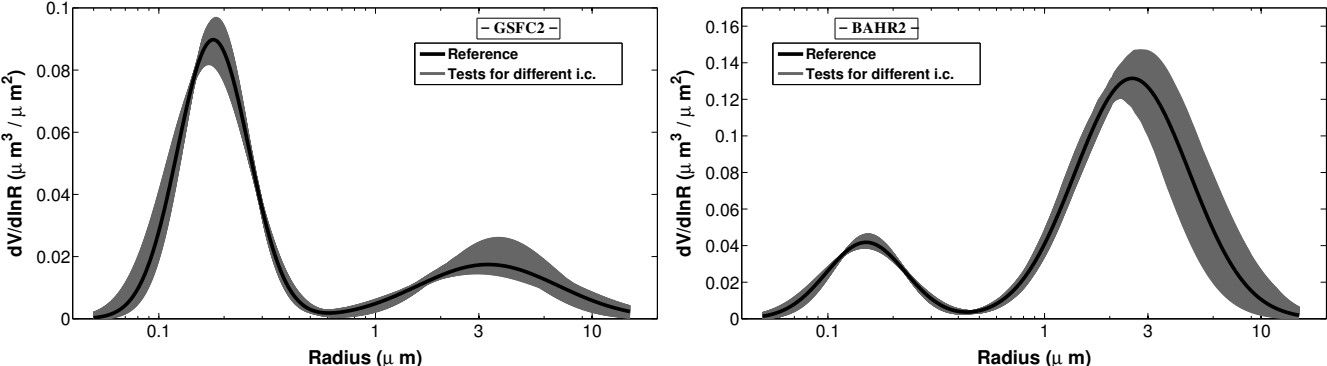

**Figure 4.** Size distribution retrieved in the multiple initial guess analyses for GSFC2 (left subfigure), and BAHR2 (right subfigure). Black solid lines are used to represent the reference values while the gray lines illustrates all the retrievals of the multiple initial guess analysis.

Its accuracy, therefore, will be conditioned by the accuracy of the retrieved parameters. The differences between the effective radii from the reference and those retrieved in the self-consistency are under $0.015\ \mu$m except for the extreme case of SOLV2 where a difference of $0.061\ \mu$m is obtained.

### 3.3 Initial guess variation

In this section, we determine the sensitivity of the inversion results to the initial guess. Specifically we test three different initial guesses for each of the six retrieved parameters, which results in a total of 729 inversions ($3^6$) for the 17 aerosol examples. Note that we need not verify that the initial guesses are consistent with the inversion results; rather, we analyze the stability of the code with respect to the initial guess. That is, a perfect retrieval code would give us the same correct results, independently of the initial guess.

Table 4 shows the default initial guesses used for the first iteration of the GRASP-AOD code, which depend upon the Ångström exponent and the aerosol optical depth at $440$ nm. (Note that these initial guesses can be modified if ancillary information for specific applications exists). Table 5 shows the variety of initial guesses chosen for this portion of our study. These values are computed from the expressions in table 4, with a modal variation of $\pm 25\%$ for $r_{V_i}$, an absolute variation of $\pm 0.1$ for $\sigma_{V_i}$ and $\pm 40\%$ for $C_{V_i}$. Here, we only show the results obtained for the examples with aerosol load $\tau_a(440){=}0.6$,

since they are similar to the results obtained with the other two aerosol loads.

  Before analyzing in detail all the cases, the 729 inversion retrievals for GSFC2 and BAHR2 are represented in figure 4 to get a visual idea of the results obtained in this analysis. Black solid lines are used to represent the reference values, while the gray shadow contains all the retrievals. It can be observed how the fine mode is well characterized whereas there is a larger uncertainty in the description of the coarse mode. The better performance of the fine mode compared to the coarse mode

was expected, since as commented, small particles have higher Mie extinction efficiencies at the selected wavelengths in the analysis.



In table 6, average values and standard deviations of the retrieved parameters calculated from the 729 inversions are shown for each aerosol type. The last two columns include the secondary products: $\tau_f(500)$ and the effective radius. We focus on the standard deviations in Table 6 for this second study, since they indicate the stability of the retrieved parameters. If we analyze $\tau_f(500)$, we see that the standard deviations are between 0.001-0.002 when the fine mode dominates, while they are between 0.003-0.004 for those cases when the coarse mode dominates. This result denotes that the separation of the modes is very stable regardless the initial guess values.

Similarly, we observe that the standard deviations for $r_{V_f}$ and $C_{V_f}$ are less than 0.01 in all cases (units $[\mu m]$ and $[\mu m^3/\mu m^2]$ respectively). The variation is even smaller for the cases with a fine mode predominance, with a maximum value of 0.003 in the two parameters. In relative terms this variation is less than 3% for both cases. These results indicate that the retrieved $r_{V_f}$ and $C_{V_f}$ have little sensitivity to the initial guess, and the variability is practically negligible for the cases with a dominant fine mode. The coarse modes in Table 6 indicate that the standard deviations are around 10% for both $r_{V_c}$ and $C_{V_c}$, regardless of whether the fine mode or coarse mode dominates.

The characterization of the geometric standard deviation of fine mode, $\sigma_{V_f}$, is more sensitive to the initial guess selection than the other fine mode parameters; the standard deviation is between 0.02 and 0.03 for the cases where the fine mode dominates, and up to 0.06 for the rest of the cases. This result denotes a low sensitivity of the retrieval to the width of the fine mode. The results are a bit worse for the standard deviation of $\sigma_{V_c}$: in those cases with a predominant coarse mode the values are between 0.03-0.04, while for the cases with a prevailing fine mode the standard deviations are around 0.08. Therefore, a good election of the initial guess of $\sigma_{V_f}$ and $\sigma_{V_c}$ is very important for a suited retrieval of both parameters.

The standard deviation of the effective radius is 5% in all the cases with a fine mode predominance, and it is between $12 - 13\%$ in the cases with a prevailing coarse mode. The largest influence of the predominant mode in the calculation of the effective radius explain the result obtained here and, as commented, the fine mode is better characterized than the coarse mode.

### 3.4  Simulation of aerosol optical depth errors

The purpose of this section is to analyze how aerosol optical depth errors affect the inversion and its products. Following the scheme in figure 3, variations of $\pm0.01$ are introduced in the aerosol optical depth values (from table 2) for each channel. Similar to the analysis of the initial guess variation, all the different combinations are considered resulting in $3^8 = 6561$ retrievals for each aerosol case: the case with no error is considered together with the variations $\pm0.01$ in the eight channels. In order to summarize, only the cases with the lowest and highest aerosol load for each type are presented here. In this case, there are some tendencies when the aerosol optical depth increases and the analysis of the results for two different aerosol load needs to be done. The error value of 0.01 is selected in the analysis, since it represents the maximum derived error (considering the Sun in the zenith, for more information see the Appendix A and in particular Eq. (A6)) from a miscalibration error of 1% on $V_0(\lambda)$, which is the expected value in AERONET-network .

So as to get a visual representation of the results, first we represent in figure 5 the retrievals for two aerosols examples, GSFC and BAHR and two aerosol loads $\tau_a(440) = 0.3$ and $0.9$. Black solid lines represent the reference values while black dashed lines depict the retrievals without errors. Note that both lines are almost identical as commented in the self-consistency study.





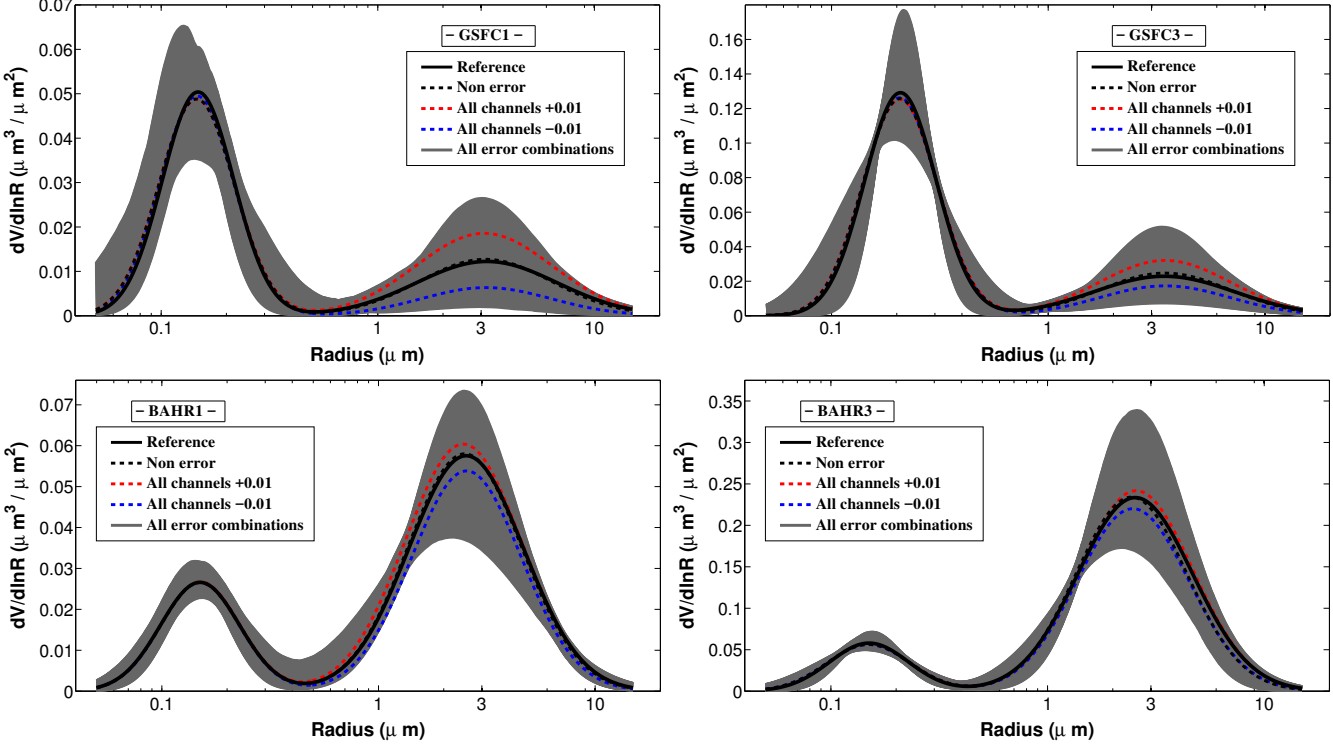

**Figure 5.** The effect of $\tau_a$ errors on size distributions retrieved with the GRASP-AOD code for GSFC (top panels) and BAHR (bottom). The subfigures in the left corresponds to the cases with $\tau_a(440) = 0.3$, and in the right, with $\tau_a(440) = 0.9$. Black solid lines are used to illustrate the reference values while gray dashed lines represent the retrievals without errors (self-consistency study). Red and blue lines are used to indicate the cases where all the wavelengths have the same error $+0.01$ and $-0.01$ respectively. Gray lines illustrates all the retrievals for the multiple combinations of errors.

Gray regions illustrate the retrievals for the rest of the combinations. The cases where all the wavelengths have the same errors of $+0.01$ and $-0.01$ are illustrated with red and blue lines respectively. The red line (all errors $+0.01$) gives larger values of the size distribution than the reference, whereas the opposite happens for the blue line. However, these retrievals are quite close to the reference values when compared to the envelope of the gray area. Note that for both, red and blue lines, the mode radii are practically the same as those obtained in the self consistency test, the discrepancies are only observed in the concentration values. On the other hand, discontinuities in consecutive wavelengths (e.g. $\Delta\tau_a(1020) = -\Delta\tau_a(1640)$) produce changes in the spectrum slope, and consequently variations, not only in the concentration, but also, in the retrieval of mode radii.

The uncertainties in $r_{V_f}$ and $C_{V_f}$ are significantly reduced for GSFC3 and BAHR3 with respect to GSFC1 and BAHR1. For GSFC3, even though there is still an extensive gray shadow, the uncertainty is associated with the large dispersion of $\sigma_{V_f}$ as we will see in the detailed analysis. In the coarse mode, there is a notable improvement of $r_{V_c}$ and $C_{V_c}$ for GSFC with the increasing of the aerosol load which is not observed for BAHR.



Table 7 presents the values and standard deviations of the retrieved parameters calculated for the 6561 inversions for each aerosol case. As in the study of the initial guess variation, standard deviation values will be used to check the stability of the retrieval, though in this case, with respect to the $\tau_a$-errors.

Average values of $\tau_f(500)$ obtained in the study do not differ by more than 0.005 from those retrieved in the self-consistency analysis. The standard deviations of $\tau_f(500)$ are between 0.010 and 0.015. This result is consistent with the variation of $\pm 0.01$ in the aerosol optical depth considered in the study.

The largest differences between the mean values of $r_{V_f}$ in table 7 and the values of the model (in table 1) is $0.008\ \mu m$, and it is obtained in the case of LANA2. The standard deviations of $r_{V_f}$ are between 0.008 and 0.014 without significant variations for the different aerosol models. Note that when the aerosol optical depth increases there is a slight diminution in the standard deviation for all aerosol cases. This effect is more visible in the cases with a prevailing fine mode, where the variations are about 10% for the cases with $\tau_a(440) = 0.3$ and they diminish to $5 - 6\%$ for the cases with $\tau_a(440) = 0.9$

The differences between the average values of $r_{V_c}$ and the input values are larger than for $r_{V_f}$. They are around $0.4\ \mu m$ for the aerosol cases dominated by the coarse mode and around $0.3\ \mu m$ for the cases with a prevailing fine mode. These differences are the same as in the self-consistency analysis. Actually, the average values here and those obtained in the self-consistency do not differ by more than $0.1\ \mu m$ for all the cases.

The standard deviation of $r_{V_c}$ are between 0.1-0.3, indicating that the retrieval of the coarse mode is less stable against possible errors in the aerosol optical depth measurement. These variations represent between the 5% and 12% in relative terms. No improvement is observed when the aerosol load grows except for the case of GSFC where the standard deviation is reduced to the half for GSFC3 respect to GSFC1. Note that the same absolute error of $\pm 0.01$ is considered for all the wavelength which is much higher in relative terms for long wavelengths (see table 2) than for short ones. The largest influence in the coarse mode of the longest wavelengths justifies this result compared to the one obtained for $r_{V_f}$.

The mean values of the fine mode concentration, $C_{V_f}$, are almost identical to those obtained in the self consistency analysis with maximum differences of $0.002\ \mu m^3/\mu m^2$ or in relative terms under 5%. The standard deviation are typically around 10% for the cases with fine mode predominance and around 5% for the aerosol cases with a prevailing coarse mode.

The analysis of $C_{V_c}$ gives differences of 5-10% between the average values in table 7 and those obtained in the self-consistency study for the cases with a prevailing fine mode. For the three cases, no tendencies regarding the total aerosol optical load are observed. However, a reduction in the differences with the increase in the aerosol load is observed for those cases dominated by the coarse mode. Thus, the differences are around 7% for SOLV1 and BAHR1, they get reduced under 3% for the cases of SOLV3 and BAHR3.

Back to the cases with a fine mode predominance, table 7 shows that the standard deviations of $C_{V_c}$ are relatively high with values between 0.01 and $0.02\ \mu m^3/\mu m^2$. Note that for the cases with the lowest aerosol load, sometimes these variation represents around 50% of the total value. Certainly, variations of $\pm 0.01$ for the longest wavelengths should have a large influence on the retrieval of the coarse mode considering the values of the aerosol optical depth at these wavelengths ($\tau_a(1640) \sim 0.03 - 0.04$; see table 2 for GSFC1, ZAMB1 and MEXI1). The differences, in relative terms, are considerably smaller for the cases with the highest aerosol load.





For the cases with a prevailing coarse mode, the standard deviations of $C_{V_c}$ are around $10\%$ for all the cases without a notable differences when the aerosol optical load increases.

For both, $\sigma_{V_f}$ and $\sigma_{V_c}$, the analysis is very similar. Differences between the averages in table 7 and the self-consistency analysis are typically around $0.01$, although there are some cases with differences up to $0.015$. Standard deviations for both parameters can be delimited between $0.04$ and $0.075$ without clear tendencies regarding the aerosol load or the predominant mode.

There is also a good agreement between the average values of the effective radius in table 7 and those retrieved in the self-consistency test. The differences are under $0.015\,\mu m$ for all the examples except for SOLV3, where the difference is $0.03\,\mu m$. It should also be mentioned that this case has a significantly larger value of the effective radius with respect to the rest of the cases, which explains the difference (i.e., the difference is only $3\%$ of the total value, which is similar to the rest of the cases). In the same way, SOLV3 presents the largest value of the standard deviation: $0.111$. In relative terms, this represents $10\%$ of the effective radius, and curiously, it is a bit lower than for the rest of the cases where the values are between $10-15\%$.

## 3.5 Sensitivity to the refractive index

As it was already discussed in the introduction, the information contained exclusively in the spectral aerosol optical depth measurements is not enough to retrieve the refractive indices. We have assumed this parameter as known thus far, and at this point we wish to determine how errors in this assumption affect the characterization of the size distribution. Thus, we use the same scheme presented in figure 3 to answer this question, but this time we modify the values of the refractive indices.

We only analyze the effects on three of the aerosol cases are here: GSFC and SOLV to represent the cases with fine or coarse mode predominance, and ZAMB is also included to review the effects in an absorbing aerosol.

### 3.5.1 Pre-analysis with the forward code

First, we use the forward code to check the variation in the aerosol optical depth generated by a modification in the refractive index. Although the retrievals for three aerosol loads were evaluated in the main study, we only evaluate variations for GSFC2, SOLV2 and ZAMB2 (cases with $\tau_a(440) = 0.6$) in this portion of the analysis.

Thus, in table 8, the change in the aerosol optical depth caused by a change of $\pm 2\sigma$ in the refractive index (in both parameters $n$ and $k$) can be determined for the three examples considered. The values of $2\sigma$ are obtained as well from the climatology study of Dubovik et al. (2002a) (see table 1 in that work). In the cases GSFC2 and ZAMB2, $2\sigma_n$ is equal to $0.02$ whereas it is $0.06$ in the case of SOLV2. For the imaginary part, $2\sigma_k$ has values of $0.008$, $0.006$ and $0.002$ for ZAMB2, GSFC2 and SOLV2 respectively. In all the cases, the values of $2\sigma$ are spectrally independent. Note that for SOLV2 and ZAMB2, the case with $k - 2\sigma_k$ give a negative value of the absorption in some wavelengths. In those cases, $k$ is fixed to zero and it is indicated with an asterisk in table 8.

An increase in the real part of the refractive index produces an increase in $\tau_a$ because more light is scattered, and consequently, the direct beam is reduced (Bohren and Huffman, 1983). This change is symmetric as it can be seen in table 8.





Moreover, $\Delta\tau_a$ is linear with the increments of $n$ considered in this study (the same results were obtained in Torres (2012) where a more detailed explanation can be found).

Another interesting result is the larger effect produced by an increment of $n$ for GSFC2 and ZAMB2 compared to SOLV2: although $2\sigma_n$ is three times larger for SOLV2, the increments of the aerosol optical depth are similar for the three examples (at

440 nm the variations are $0.046$ and $0.034$ for GSFC and ZAMB while it is $0.039$ for Solar Village). The explanation lies in the fact that most of the increment produced by the real part of the refractive index goes to increment the optical depth of the fine mode without practically changing the optical depth of the coarse mode. Only for SOLV2, there is a slight decrease in the optical depth of coarse mode for the largest wavelengths. This property also explains the strong spectral dependence in table 8: the fine mode optical depth, $\tau_f$, is much larger for short wavelengths respect to long wavelengths.

Finally, it can be observed in table 8, that the variation in the imaginary part has not a big influence in the aerosol optical depth. The maximum variations, always under $0.01$, were obtained for the case ZAMB which is the case with the largest $2\sigma_k$. As expected, positive increments of the imaginary refractive index increase the absorption, and therefore, the aerosol optical depth. In this case, the variation is not only allocated in the fine mode, but it is distributed in both modes. Nevertheless, the predominant mode gets the largest variations for each particular aerosol example.

### 3.5.2    Retrieval analysis.

Following the scheme from figure 3, GRASP-AOD code has been applied to the optical depth values for the aerosol cases GSFC, ZAMB and SOLV (table 2), considering variations of $\pm\sigma$ and $\pm2\sigma$ (see Dubovik et al., 2002a) from the original values of their refractive index (in table 1) during the inversion procedure.

Table 9 presents the retrieved values of the fine mode aerosol optical depth at $500\ nm$ for variations of $\pm2\sigma$ in the real,

and in the complex part of the refractive index. The maximum difference between the retrieved values with and without the variations is observed for the case SOLV3 and is equal to $0.005$; however, the differences are typically around $0.001 - 0.002$. A similar result is obtained for the rest of the wavelengths, although it is not presented here. Therefore, the discrimination between the extinction of fine and coarse modes does not depend upon the refractive index assumption.

Figure 6 shows the size distributions obtained by varying the real and the imaginary parts of the refractive index by $\pm\sigma$ and

$\pm2\sigma$ during the retrieval process, for the three aerosol cases (for each of them with the three different aerosol load $\tau_a(440) = 0.3, 0.6$ and $0.9$). In the figure, red lines are used to indicate the cases when the refractive index is overestimated and blue lines represent the cases when the parameter is underestimated. Variations of $\pm\sigma$ are depicted by solid lines and the ones of $\pm2\sigma$ by dashed lines. Finally, black solid lines illustrate the reference values while gray dashed lines represent the retrievals without variations in the refractive index (self-consistency study).

The retrieval behavior of the fine mode presents a similar pattern for the three analyzed cases towards the variations of the real refractive index: there is a significant decrease of the mean radius and the volume concentration of the fine mode when the real part of the refractive index increases. For negative variations, both parameters increase their values. Thus, the increments of $r_{V_f}$ are between $\mp4 - 5\%$ for GSFC and ZAMB while they are between $\mp8 - 9\%$ for SOLV for variations of $\pm2\sigma_n$ in the refractive index. In the case of $C_{V_f}$, the increments are between $\mp2 - 3\%$ for the examples with a predominant





fine mode and around $\mp 7 - 8\%$ for SOLV. For both parameters a linear behavior is been observed so the differences for $\pm\sigma_n$ are approximately half as large as for $\pm 2\sigma_n$. These results were expected after the pre-analysis with the forward code and the stability of $\tau_f$ observed in table 9. As mentioned earlier, an increase of the refractive index results in a rise of the aerosol optical depth. Since $\tau_f$ does not change, the increase of the real refractive index needs to be compensated. It is balanced by

the decrease of the particles but also with the reduction of $r_{V_f}$. Note that the extinction efficiency diminishes as the radius decreases in the fine mode, specially for the shortest wavelengths.

For the coarse mode, the variations of $r_{V_c}$ and $C_{V_c}$ do not show a clear tendency for GSFC and ZAMB, with values practically negligible (typically under $1\%$). In the case of SOLV, however, the same tendency as the fine mode can be observed: the mean radius and the volume concentration decreases between $3 - 4\%$ (for $+2\sigma_n$) when the real part of the refractive index increases.

Note that for the radius range of the coarse mode, the extinction efficiency diminishes as the radius grows. At the same time, there is a small increment in $\sigma_{V_c}$ (around $0.02$ in absolute value) which, as can be seen in figure 6, enlarges the presence of the particles in the radius range between $0.7 - 1\mu m$. Actually, the net balance of all effects is a slight increment of the optical depth in the coarse mode for the longest wavelengths. This increment compensates the reduction of the optical depth when the refractive index grows, as commented in the pre-analysis with the forward code for the case of SOLV.

The effective radius follows the same tendency as the mean radii of the modes. In the examples with a prevailing fine mode for the maximum increment of $\pm 2\sigma_n$, it varies as $\mp 0.01\mu m$ in the three aerosol loads of GSFC and $\mp 0.006\mu m$ in those of ZAMB. These changes represents the $\mp 5\%$ and the $\mp 3\%$, respectively, of the total value. In the examples of SOLV, the variations are of $\mp 3\%$ for the two lowest aerosol loads and approximatively $\mp 7\%$ for SOLV3.

The three sub-figures at the right in figure 6 depict the retrievals varying the imaginary part of the refractive index. The

size distributions show similar patterns for the three aerosol cases: mean radii of the two modes do not have large variations with difference under $2\%$ in the case of $r_{V_f}$, being a bit higher for $r_{V_c}$ where they are around $3 - 4\%$. For both modes, these differences do not present a clear behavior being randomly positive or negative regardless the sign in the variation of the refractive index. Analyzing the concentration, however, a definite tendency can be observed: they decrease for positive variations of the refractive index and they increase for negative increments. The differences are around $1\%$ for $C_{V_f}$ and between

$3 - 4\%$ for $C_{V_c}$. This result is expected, since when the absorption is increased there is an increase of the aerosol depth that is compensated by a reduction in the number of the particles. The behavior in the concentration, however, is not so clear in figure 6 where it seems that the opposite happens. The explanation is in the standard deviation of the modes which also diminishes for positive increments of the absorption and increases in the opposite case. Thus the maximum values in the size distribution are reached for the cases of $+2\sigma_k$ though the aerosol concentration are the lowest in these cases. The maximum variations of $\sigma_{V_f}$

and $\sigma_{V_c}$ are between $0.015 - 0.022$ (absolute values) independently of the aerosol type. The variations in the effective radius are negligible for GSFC and ZAMB (differences under $0.5\%$). In the case of SOLV, differences up to $2\%$ are observed though there is no a clear tendency in them.



## 4 Real cases

### 4.1 Comparison with AERONET

We present here a comparison between the aerosol products obtained from the new GRASP-AOD inversion and AERONET products, in order to validate the results achieved in previous sections. In particular, we compare our results with: (a) size distributions obtained from the AERONET inversion code (Dubovik and King, 2000), and (b) $\tau_f(500)$ obtained from the SDA (O'Neill et al., 2003). All of the data and products used in the comparison belong to AERONET level 2.0 (quality assured data) and can be found in the public AERONET database (http://aeronet.gsfc.nasa.gov).

### 4.1.1 Data selection

To homogenize the two different data sets (almucantar and spectral aerosol optical depth measurements), we propose a comparison of daily averages on days with stable aerosol conditions instead of comparing single inversions. This approach allows us to include a large number of $\tau_a$ inversions which will be useful to check the stability of the new inversion. We limit our analysis to days that fulfill three requirements. (1) There are a minimum of 15 AOD measurements per day; each AOD measurement is only eligible if there is a valid almucantar within a maximum of $\pm 30$ minute delay. This requirement also improves the quality of the $\tau_a$ data selected as there are only almucantars in level 2.0 with $\theta_s > 50°$. That is, a maximum delay of half an hour assures that all $\tau_a$ measurements are obtained when $\theta_s > 40$, which reduces the bias produced by errors in the calibration coefficients (see Eq. (A6)). (2) In order to guarantee stable aerosol conditions, the ratio between the standard deviations and the averages of the eligible aerosol optical depths are required to be smaller than $0.1$ throughout the day (evaluated for every spectral wavelengths used by the GRASP-AOD inversion). (3) There should be a minimum of 4 valid almucantars per day.

Following these conditions, 23 days have been selected from the same sites used as (climatological) inputs in the simulations. In total, 130 AERONET inversions (almucantar and AOD) and 592 GRASP-AOD inversions have been compared. Table 10 depicts the information regarding the daily conditions of the analyzed cases. The first columns in table 10 contains technical information about the day selected (date, photometer number and site). Unfortunately Mexico-City, Mongu and Lanai sites have not counted with a extended photometer for the measurements, and therefore, we have used only standard wavelengths for the analysis in those sites. This fact is indicated with the label "STD" (for the standard photometers) or "EXT" (for the extended photometers) together with the AERONET number of the instrument. The daily averaged value and the standard deviation of the aerosol optical depth at $440$ nm as well as the Ångström exponent (from $440$ nm, $675$ nm and $870$ nm) are indicated in columns four and five, so as to give an idea of the characteristics of the aerosol analyzed: load and predominance of fine or coarse mode. As can be seen from the table, we covered a broad aerosol load with values ranging from 0.2-0.3 up to 0.6-0.7 using four examples for each site. In the case of Lanai, however, due to the impossibility of finding stable days with $< \tau_a(440) >$ larger than 0.25 only 3 cases with values ranging between 0.1 and 0.25 have been selected.

The last two columns indicate the number of inversions: ALM indicates the number of almucantar used for the AERONET standard inversion, AOD the number of inversions used for both GRASP-AOD inversion and SDA algorithm. Note that for GRASP-AOD inversions we have considered the climatological values from table 1 for the refractive index and sphericity.





### 4.1.2 Results

Table 11 represents the daily average of the standard parameters for fine and coarse modes retrieved using AERONET-standard and GRASP-AOD inversion for the chosen days shown in table 10. While both $r_{V_f}$ and $r_{V_c}$ are direct outputs of the GRASP inversions, the standard AERONET inversions gives a 22-bin retrieved size distribution. In the latter situation, values of $r_{V_f}$ and $r_{V_c}$ are estimated using the standard AERONET procedure described in http://aeronet.gsfc.nasa.gov/new_web/Documents/Inversion_products_V2.pdf.

Figure 7 illustrates the absolute differences of the mean volume radii by using GRASP-AOD and AERONET standard inversion calculated from the values in table 11. In the left panel, the differences for fine mode are represented, while those of the coarse mode are depicted in the right panel. Analyzing the differences of $r_{V_f}$ at sites with fine mode predominance, we observe an excellent agreement between both inversions, with differences under $0.01 \, \mu$m except for the cases of GSFC -A- (maximum difference of $0.028 \, \mu$m) and MEXI -A- ($0.013 \, \mu$m). For the sites with a prevailing coarse mode, the differences are a bit higher, generally, ranging from $0.010 \, \mu$m to $0.020 \, \mu$m. Note that the two cases with the largest differences, GSFC -A- and LANA-C, also present the largest $r_{V_f}$ (not shown in figure) — $0.27 \, \mu$m for GSFC -A- and $0.26 \, \mu$m for LANA -C-, while the values for the other cases range between $0.14$-$0.19 \, \mu$m. In relative terms, the maximum difference is 15 % (SOLV -C-), thought the differences are under 10 % in most of the cases.

The differences in $r_{V_c}$ are under $0.3 \, \mu$m for the cases with $\tau_a(440) > 0.3$. The maximum differences are reached for the cases with a prevailing fine mode and $\tau_a(440) < 0.3$: MEXI -A- and ZAMB -A- with values of $0.49 \, \mu$m and $0.47 \, \mu$m, respectively. The analysis in relative terms give similar results as in the fine mode, with differences of around 15 % in the extreme cases, although they do not exceed 10 % in the rest of the cases.

In the case of $C_{V_f}$, the largest differences are obtained for the cases with a predominant fine mode with a maximum value of $0.018 \, \mu$m$^3/\mu$m$^2$ found for MEXI -D- (see table 11). For $C_{V_c}$, the largest differences are found for the desert dust cases, with a maximum value of $0.059 \, \mu$m$^3/\mu$m$^2$ (SOLV -D-). The high variability of the concentration for both modes makes the analysis more suitable in terms of relative differences. Figure 8 shows the relative differences (in modulus) of the daily volume concentration retrieved by GRASP-AOD and AERONET standard inversion for the different examples. As a general result, we observe that for $C_{V_f}$, the differences are the smallest for the cases dominated by fine mode, while for $C_{V_c}$ the smallest differences are obtained for the cases with a prevailing coarse mode. Thus, the differences of the volume concentration in the predominant mode are always under 20 % for all the examples. In general terms, once again the characterization is a bit better for the fine mode (average of the differences of $C_{V_f}$ around 15%) than for the coarse mode (average of the differences of $C_{V_c}$ around 18%).

In figure 9, the absolute differences for the daily means of $\sigma_{V_f}$ and $\sigma_{V_c}$, retrieved by GRASP-AOD and AERONET standard inversions (table 11), are shown. The average of the differences is $0.065$ for $\sigma_{V_f}$ and $0.069$ for $\sigma_{V_c}$, although differences up to $0.2$ are observed in both cases. These results confirm one of the outcomes from the sensitivity analysis, where we pointed out that the retrieval of the standard deviations (mode widths) is less accurate than the retrieval of the radii and the concentrations.





Values of $\tau_f(500)$ are computed using the retrieved parameters from AERONET standard[2] and GRASP-AOD inversion. Both values are presented together with those obtained by the SDA algorithm in table 12. It can be observed that, in general, the values of $\tau_f(500)$ retrieved by GRASP-AOD are ranging between those retrieved by the standard AERONET inversion

and the SDA algorithm. Analyzing the differences for the cases with a prevailing fine mode, we see that a maximum value of $0.015$ is found for the case ZAMB -C- between GRASP-AOD and the standard AERONET inversion. On the other hand, a maximum difference of $0.021$ is observed between GRASP-AOD and SDA algorithm for the case ZAMB -A-. The analysis of the cases dominated by the coarse mode shows difference up to $0.035$ (SOLV -D-) between GRASP-AOD and the standard AERONET inversion. The maximum difference is larger for the comparison GRASP-AOD and SDA algorithm with a value of

$0.048$ (BAHR -D-)

Note that computing $\tau_f(500)$ also allows us to compare the results obtained by the SDA and AERONET standard inversions for the examples presented here. This comparison was already done in O'Neill et al. (2003) (and in the technical memo of the Spectral Deconvolution Algorithm; http://aeronet.gsfc.nasa.gov/new_web/PDF/tauf_tauc_technical_memo.pdf) and in other studies like in Eck et al. (2010). The previous studies have pointed out a negative bias for $\tau_f(500)$ calculated from SDA with

respect to the AERONET standard inversion. The same general tendency is observed here for all the cases presented. For the examples with a predominance of fine mode, absolute values of the differences are a bit smaller here than those found in O'Neill et al. (2003) Thus, a maximum difference of 0.02 is observed here (case MEXI -B-) while in O'Neill et al. (2003) the differences were ranging between 0.015-0.03, for examples of similar characteristics[3]. The comparison for the cases dominated by the coarse mode shows differences between 0.01-0.05. The same range is found in the study by Eck et al. (2010), which

contains examples with large coarse mode (see section 3.2: $\sim 0.03$ at Ilorin site and $\sim 0.05$ at Kanpur site).

Table 12 also contains the values of $r_{eff}$ retrieved by GRASP-AOD and AERONET standard inversion. For the cases with a prevailing fine mode, the largest differences are obtained for the cases with the lowest aerosol load (cases with index -A-). Among them, the largest value is $0.05$ $\mu$m reached for the case ZAMB -A-. In the other cases with $\tau_a(440) > 0.3$, the differences are between $0.01$ $\mu$m and $0.02$ $\mu$m which is equivalent to an uncertainty of $5-10\%$ registered in the different

sensitivity tests. The differences are higher in the cases with a predomination of the coarse mode. The extreme case, both in absolute and relative terms, is found for BAHR -B- with an absolute difference of $0.153$ $\mu$m which is around the $30\%$ in relative terms. For the rest of the cases, the differences are under $0.1$ $\mu$m, or in relative terms, under the $15\%$.





## 4.2 Night AOD inversion from Moon photometers.

The recent advances in Moon photometry (in terms of instrumentation, calibration and data treatment) have enabled the set-up of a few sites with night measurements. One of the pioneer sites at Lille University has been providing regular night AOD

measurements since 2013 from a moon-photometer (Cimel model CE318-U ). The typical cloudy conditions of the site and the constraints due to the moon cycle (only 14 days out of 28 valuable for night measurements) limit the study cases to a relatively small number of nights.

The left panel of figure 10 shows the aerosol optical depth measured during one of those nights, specifically on 9th April 2015. In the same figure, we have added the AOD values measured by the sun-photometer #741 (AERONET number) during

previous evening and the following morning. Even though there is only one hour of valid measurements (a total of six measurements), the case represents an interesting example with relatively high AOD values due to the pollution episode that occurs in the North of France in the spring 2015 on 7th-9th of April. During those days, the values of $\tau_a(440)$ were typically ranging between $0.4$ and $0.6$ (with peaks up to $0.9$ on 8th April) while the climatological average at the Lille site is $< \tau_a(440) >= 0.22$.

In the right panel of figure 10, size distributions retrieved by GRASP-AOD inversion for the night AOD-measurements are

represented (six inversions in red solid lines). In the same figure, we have added the GRASP-AOD retrievals from the two AOD measurements of the previous evening (blue lines) and from one of the AOD measurement on the next morning (dashed black line). The AOD measurement selected is the one coincident with the almucantar measure and together they form the input of the first AERONET-standard inversion of the day and the closest to the night measurement. Note that the refractive indices retrieved from this inversion have been used in the GRASP-AOD retrievals shown in figure 10 (day and night). The 22 bins

size distribution from this inversion is also illustrated in black solid line.

Observing the AERONET retrieval, we can see that the episode is characterized by a predominant fine mode ($C_{v_f}/C_{v_c} = 4$) with relatively high values of $r_{v_f} = 0.320$ $\mu$m and $r_{eff} = 0.317$ $\mu$m compared to the typical urban aerosols in Lille ($r_{eff} < 0.2$ $\mu$m see Mortier, 2013). The average of the retrieved values during the night show similar tendencies in the fine mode with $r_{v_f} = 0.304 \pm 0.01$ $\mu$m and $r_{eff} = 0.296 \pm 0.012$ $\mu$m. The ratio between the concentrations, however, is a bit higher

$C_{v_f}/C_{v_c} = 6.94 \pm 1.45$ than the one found in the AERONET retrieval. This difference is mainly due to the discrepancies in the coarse mode between both retrievals: the $C_{v_c}$ retrieved during the night is $0.009 \pm 0.002$ while in the AERONET retrieval is $0.015$. The concentration values of the fine mode are quite similar $C_{v_f} = 0.057 \pm 0.003$ (GRASP-AOD) and $C_{v_f} = 0.06$ (AERONET)

---

[2]The computation from the retrieved parameters of the standard AERONET inversion requires running the forward model with extra information about the refractive index at 500 nm. In our case, the value of the refractive index is interpolated from 440 nm and 675 nm for every single inversion.

[3]This slight improvement can be justified since the AERONET Version 2 is used in the present work instead of Version 1 employed in O'Neill et al. (2003). In Version 2, the fine-coarse mode separation is done by finding the minimum value in the size distribution within the size interval from 0.439 to 0.992 $\mu$m, while in Version 1 the separation was done universally at 0.6 $\mu$m. The universal cut off at 0.6 $\mu$m was the main source of discrepancies between the algorithms in O'Neill et al. (2003), and justified using Mie calculations in the technical memo of SDA. The results of this comparison certainly improves with the new fine-coarse mode separation in Version 2.



Finally, the $\tau_f(500) = 0.360 \pm 0.004$ retrieved during the night from GRASP-AOD is in an excellent agreement with the value retrieved during the next morning by SDA, $\tau_f(500) = 0.337$, and the one derived by the almucantar inversion $\tau_f(500) = 0.349$.

## 4.3 Application of the method to airborne PLASMA.

The spectral optical depth measurements obtained by airborne sun-photometers represent another interesting application example of the GRASP-AOD inversion. Typically the use of airborne sun-photometers is limited to field campaigns developed in singular areas with specific research objectives such as validation of models, characterization of specific aerosol types or interaction between aerosols and other atmospheric components like clouds or gases. From the AOD measurements taken at different heights, the derivation of the spectral aerosol extinction profile is immediate. The extinction profiles are normally available for every landing/taking off and during pre-scheduled vertical profiles carried out by the airplanes and they give basic information for the characterization of the aerosol vertical distribution.

In the example presented here, we use the data from PLASMA airborne sun-photometer during the Chemistry-Aerosol Mediterranean Experiment/Aerosol Direct Radiative Forcing on the Mediterranean Climate (ChArMEx/ADRIMED) summer campaign that took place from 11 June to 5 July 2013 over the western Mediterranean (Mallet et al., 2016). PLASMA airborne sun-photometer was installed in the ATR-42 French research aircraft operated from Sardinia (Italy) and participated in the 18 flights of the campaign. The experimental setup also involved several ground-based measurement sites on different Mediterranean islands (Corsica, Lampedusa, Minorca and Sicily) and additional measurements from lidar and sun-photometers performed on alert during aircraft operations (at Granada and Barcelona). The meteorological conditions observed during the campaign (moderate temperatures and southern flows) were not favorable to produce large concentrations of local polluted smoke particles. However, several moderate dust plumes were observed during the campaign, with the main sources located in the north-west of Sahara desert. Though peaks of AOD values up to 0.6 (at 440 nm) were registered by ground-based sun-photometers during the campaign, the maximum values measured by collocated sun-photometers at the time of ATR-42 vertical profiles were around 0.3 (at 440 nm).

The AOD values represented in the left part of figure 11 correspond to PLASMA measurements at different heights during one of the aforementioned plumes, specifically, in the taking off from Minorca airport on the 17 June 2013 between 11:45 and 12:00 UTC. The average values of the three AOD-data measured during this period by the ground based AERONET sun-photometer at Minorca (AERONET site Cap d'en Font see Chazette et al. 2016) are also plotted as reference. Note that the ground based sun-photometer is standard (see description in table A1) and does not have the 1640 nm channel. At the same time, the 340 nm channel of PLASMA suffered from instabilities during the campaign and the derived AOD-data at this channel is not represented.

One of the first conclusion obtained from figure 11 is the vertical homogeneity of the aerosol characteristics in the whole column through 4000 m. The same property was observed at analyzing the coincident lidar data (figure 6 in the study by Chazette et al. (2016) on the 17 June 2013. The AOD diminishes with the height but the spectral shape of the measurements does not change considerably: the Ångström exponent (calculated like in AERONET $\tau_a(440)$, $\tau_a(675)$ and $\tau_a(870)$) varies





between 0.46 at 500 m to 0.61 at 4000 m. The $\alpha$ from AERONET sun-photometer reference at ground level is 0.48. This same property is visible in the size distribution obtained by GRASP-AOD inversions and represented in the right part of figure 11: neither the mean volume radius nor the standard deviation of the modes change significantly for the retrievals at different

heights. For $r_{V_f}$, the values go from 0.120 $\mu$m at ground level and 500 m to 0.135 $\mu$m at 4000 m. In the case of $r_{V_c}$, the variation goes from 2.06 $\mu$m at 4000 m to 2.21 $\mu$m at ground level. The ratio between the concentrations ($C_{V_f}/C_{V_c}$) also maintains from 0.19 to 0.25 until 2500 m. For the last three levels, it increases up to 0.4 though the concentrations are very low for both modes. Note that the volume size distribution from the closest AERONET standard inversion (9:00 AM) is also represented in the figure as a reference. The refractive index of this inversion are used for the different GRASP-AOD inversions

of the size distributions in figure 11.

## 5   Conclusions

The main goal of the present work is to show the potential of retrieving the total column aerosol size distributions from spectral optical depth measurements without the aid of coincident radiance measurements, and estimating a set of secondary aerosol properties (e.g. effective radius or fine mode fraction of aerosol optical depth) derived from it. The limited information content

in spectral AOD measurements results in the necessity of using a priori constraints. The utilization of GRASP approach and public software allowed us to test and evaluate different aerosol model descriptions and a priori constraints. The current analysis indicates that bimodal log-normal size distributions and a priori estimates of refractive indices provide a practically efficient retrieval setup. The validation of the retrieval has been done through a) a sensitivity analysis using 6 different aerosol models with three aerosol loads in each of them (section 3), and b) comparing the aerosol properties obtained from 600 AERONET

observations using GRASP-AOD inversion to those obtained through 130 almucantar AERONET standard inversion at 6 different AERONET sites.

    The simulated tests have shown that spectral AOD measurements are sufficient for a precise discrimination between the extinction of fine and coarse modes, independently from any assumption, with maximum differences in $\tau_f(500)$ under 0.01 from the input values found in all the sensitivity tests. Specifically, the characterizations of aerosol fine mode optical properties

are accurate, although they depend on reliable a-priori information about the real refractive indices and accurate measurement of aerosol optical depths. The uncertainty observed during the sensitivity tests for the fine mode $r_{V_f}$ and $C_{V_f}$ is between 5%, for the cases with a fine mode predominance, and 10% for the cases with a prevailing coarse mode. The characterization of the optical properties of the coarse mode using AOD measurements is less accurate, but can be significantly improved using moderate a priori information about coarse mode parameters (for example, using as initial guess data retrieved from near

almucantar inversions or climatological values of the site). The study showed that a good calibration of the long wavelengths (1020 nm and 1640 nm) is essential due to its influence on the coarse mode and the typical low aerosol optical depth values presented in this spectral range. Nevertheless, in the cases where the coarse mode is predominant, the uncertainty observed in sensitivity studies is 10% for $r_{V_c}$, and around 20% for $C_{V_c}$. The effective radius has been revealed as a quite stable parameter





towards calibration errors (uncertainties around 15%) and bias in the assumed refractive index (differences under 10% for the analyzed cases).

Daily averaged aerosol properties obtained applying GRASP-AOD inversion to 600 AERONET observations were com-
pared to those retrieved by 130 almucantar AERONET standard inversion for the 6 selected AERONET sites. The retrieved values of $\tau_f(500)$ using GRASP-AOD are generally between those obtained by AERONET standard inversion and the Spectral Deconvolution Algorithm, with differences typically lower than 0.02 between GRASP-AOD and both algorithms. In some cases with large aerosol load in the coarse mode, maximum discrepancies up to 0.05 are observed between the three retrieval methods. The study on real cases confirms the retrieval capabilities in the fine mode: maximum differences of 0.028 $\mu$m in
$r_{V_f}$ between GRASP-AOD and AERONET inversion are observed though they are generally lower than 0.015 $\mu$m (10% of the values) if the aerosol optical depth at 440 nm is larger than 0.3. In these last cases, the differences in $C_{V_f}$ are typically under 0.01 $\mu$m$^3$/$\mu$m$^2$. The differences are between the 10% and 15% in relative terms, which is in agreement with the results obtained in the sensitivity study. The comparison of coarse mode parameters shows larger differences, however, especially at the sites with a prevailing fine mode. In these cases differences of 0.2-0.3 $\mu$m in $r_{V_c}$ are normally obtained, although some
extreme differences of up to 0.5 $\mu$m are also observed. The comparison improves significantly when limiting the study to those sites characterized by the predominance of desert dust aerosol, with differences typically under 0.2 $\mu$m between GRASP-AOD and AERONET. In relative terms, these differences are under 20% for all the dust cases, which agrees with the uncertainty estimated from the sensitivity tests. That is, the characterization of the effective radius shows typical differences between $5-10\%$ for the cases with fine mode predominance, and around 15% for the desert dust cases.

*Acknowledgements.* The research has been supported by the Labex CaPPA: the CaPPA project (Chemical and Physical Properties of the Atmosphere) is funded by the French National Research Agency (ANR) through the PIA (Programme d'Investissement d'Avenir) under contract "ANR-11-LABX-0005-01" and by the Regional Council "Nord Pas de Calais - Picardie" and the European Funds for Regional Economic Development (FEDER)

Acknowledgement to ESA/ESRIN for funding through the IDEAS+ (Instrument Data quality Evaluation and Analysis Service) contract.

This research has also received funding from the French National Research Agency (ANR) project ADRIMED (contract ANR-11-BS56-0006). This work is part of the ChArMEx project supported by ADEME, CEA, CNRS-INSU and Météo- France through the multidisciplinary programme MISTRALS (Mediterranean Integrated Studies aT Regional And Local Scales).

The authors acknowledge the funding provided by the European Union (H2020-INFRAIA-2014-2015) under Grant Agreement No. 654109 (ACTRIS-2). Financial support was also provided by MINECO (CTM2015-66742-R).

We thank the AERONET principal investigators and site managers at sites Solar Village (Brent Holben and Steve Wilcox), GSFC (Brent Holben and Jon Rodriguez), Bahrain (Brent Holben and Andreas Goroch), Mexico-City (Amando Leyva Contreras and Hector R. Estevez), Lanai (Brent Holben, Chuck McClain and Robert Frouin), and Mongu (Brent Holben and Mukufute Mukulabai) We thank the AERONET, Service National d'Observation PHOTONS/AERONET, INSU/CNRS, RIMA and WRC staff for their scientific and technical support. We acknowledge AERONET team members for calibrating and maintaining instrumentation and processing data.





## Appendix A: Aerosol optical depth measurements.

### A1    General background

Total optical depth of the atmosphere ($\tau$) can be understood as the attenuation of light passing through the atmosphere contain-
ing aerosol particles, molecules, and absorbing gases, and it is described by the well-known Beer-Bouguer-Lambert Law:

$$F(\lambda) = F_0(\lambda)e^{-m_s\tau}, \tag{A1}$$

where $F(\lambda)$ $(Wm^{-2}\mu m^{-1})$ and $F_0$ are the monochromatic direct flux densities at the Earth's surface and at the upper limit
of the atmosphere, respectively. For atmospheric applications where the Sun is the radiation source, $F(\lambda)$ and $F_0$ are defined
as solar flux densities. We intentionally avoid the adjective 'solar' here, since the flux density comes from the Moon for the
night observations presented in this work. Additionally, optical depth can also be derived from star radiation as well. Finally,
the optical air mass ($m_s$) accounts for the light path in the atmosphere and can be approximated[4] by $m_s = 1/\cos\theta_s$, where $\theta_s$
the zenith angle of the radiation source in the sky.

Under cloud-free conditions, the total optical depth can be separated into the gaseous absorption $\tau_g$, the molecular scattering
or Rayleigh scattering $\tau_R$, and the aerosol scattering and absorption $\tau_a$, which is known as the aerosol optical depth. The latter
can be derived, therefore, from the following expression:

$$\tau_a = \tau - \tau_R - \tau_g \tag{A2}$$

Instruments designed to derive the aerosol optical depth typically utilize wavelengths that lack significant gas absorption
in the region UV-NIR. This spectral region presents the highest sensitivity regarding scattering and extinction by aerosols
according to the typical sizes of the natural occurring aerosols (Shaw et al., 1973; Shaw, 1983; Dutton et al., 1994).

The instruments and wavelengths used in the real data tests of GRASP-AOD inversion (section 4) are depicted in table A1.
The main analysis of those tests has been done using data from the two main groups of instruments in AERONET: Cimel-318
standard version with a spectral range from 340 to 1020 nm, and Cimel-318 extended version that includes the 1640 nm chan-
nel[5]. The example of application of GRASP-AOD inversion on night measurements has been done with the moon photometer
Cimel-318U. Given the low signal of Moon radiation in the ultraviolet spectral region, the moon photometer Cimel-318U
does not include the 340 nm and 380 nm channels. The last example, the airborne sun-tracking photometer PLASMA (Pho-
tomètre Léger Aeroporté pour la Surveillance des Masses d'Air), has a wider spectral range (340-2250 nm) with respect to
AERONET-extended instruments. Nevertheless, the calibration of PLASMA has been done using the same protocols and tools
of AERONET, and therefore, only those channels of extended photometers were used in PLASMA retrievals.

---

[4]Only valid if $\theta_s \leq 75$. The exact formulation can be found in Kasten and Young (1989).

[5]For both instruments, there is an extra spectral band centered at 936 nm which is not used for the description of the aerosol properties. The channel is
intentionally selected in an absorption band of water vapor in order to determine the column abundance of this gas.





In order to summarize, the sensitivity study presented in section 3 has been done only for the spectral range from 340 to 1640 nm and using the channels of AERONET-Extended sun-photometer presented in table A1.

## A2  Calibration and data treatment

Calibration of the instruments is carried out by the Langley plot method (Shaw et al., 1973). The method is based on the Beer-Bouguer-Lambert law and it is directly applicable if the instrument has a linear response. In this case, Eq. (A1) can be transformed to:

$$\ln V(\lambda) = \ln V_0(\lambda) - \tau m_s, \tag{A3}$$

where $V(\lambda)$ are the digital counts measured by the instrument and $V_0(\lambda)$ is the calibration constant or the extraterrestrial signal
of the instrument.

The Langley method is used to derive $V_0(\lambda)$ by means of a set of direct Sun observations performed over a range of air masses (typically from 7 to 2). The measurements provide a straight line ($\ln V(\lambda)$ vs. $m_s$), from whose intercept ($\ln V_0(\lambda)$) the calibration constant ($V(\lambda)_0$) can be extracted.

In moon photometry, calibration presents a greater complexity than the common Langley approach for sun-photometry. The
main difficulty is that the moon is a highly variable source, and the illumination changes continuously with the lunar viewing geometry. Consequently, a lunar irradiance model needs to be considered. Barreto et al. (2013) developed the Lunar-Langley Method (LLM), which is a modification of the usual Langley technique that can be applied to cases with variable illumination conditions. The calibration coefficient $V_0(\lambda)$, is variable in time with the LLM, and is expressed as a function of the moon's extraterrestrial irradiance and the new instrument calibration constant ($\kappa(\lambda)$):

$$V_0(\lambda) = I_0(\lambda, t)\kappa(\lambda). \tag{A4}$$

Here, $I_0(\lambda, t)$ is the extraterrestrial irradiance from the moon, and is taken from the Robotic Lunar Observatory (ROLO) model, developed by (Kieffer and Stone, 2005). The ROLO model presents a relatively precise $I_0(\lambda, t)$, with maximum errors around 1% at any time and for all of the wavelengths. More information about the calibration process and instrument deployment can be obtained from Barreto et al. (2013) and Barreto et al. (2016).

In AERONET, the inter-calibration procedure is used for the calibration of field instruments. The method is based on the realization of simultaneous co-located measurements of the so-called master instrument (calibrated by Langley method) and the field instruments under certain atmospheric conditions. The field instrument can be calibrated just by a ratio of raw signals of each channel ($V_{field}(\lambda)$) with the master raw signal ($V_{master}(\lambda)$):

$$V_{0_{field}}(\lambda) = V_{0_{master}}(\lambda)\frac{V_{field}(\lambda)}{V_{master}(\lambda)} \tag{A5}$$





The accuracy of the master calibration is about $0.5\%$, whereas for field instruments the calibration uncertainty is $1 - 2\%$ (larger for shorter wavelengths) due to uncertainty in the calibration transfer (Holben et al., 2006). If we derive from Eq. (A3), we obtain:

$$d\tau = \frac{1}{m_s}\frac{dV_0}{V_0} \tag{A6}$$

which means, that for instance, an uncertainty of $1\%$ in $V_0$ represents a maximum error of 0.01 in the total optical depth in the extreme case of $m_s = 1$. This result will be taken into the account in the sensitivity tests of section 3.

Finally, all the data used in section 4 are part of the level 2.0 quality assured data set (http://aeronet.gsfc.nasa.gov/new_web/ PDF/AERONETcriteria_final1.pdf). The measurements were done following the standard sequences of AERONET and the

10 final values of $\tau_a$ obtained through the official data treatment of the network http://aeronet.gsfc.nasa.gov/new_web/Documents/ version2_table.pdf.





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





**Table 1.** Description of aerosol properties used for simulating the aerosol optical depth measurements which are based on the climatology study of Dubovik et al. (2002a). First rows specifies the parameters describing the aerosol load, the size distribution (modeled as a bimodal log-normal function: $r_{Vi}[\mu m]$, $\sigma_{Vi}$ and $C_{Vi}[\mu m^3/\mu m^2]$) and the sphericity parameter (Sph). The second part of the table contains the refractive index ($n(\lambda)$ and $k(\lambda)$) inputs for each aerosol model. Note that the values of the refractive index are constant for all the aerosol examples regardless the aerosol load except for the case GSFC where they depend on the aerosol optical depth at 440 nm represented as $\tau_a$.

| | $\tau_{a_{ref}}(440)$ | $r_{V_f}$ | $\sigma_{V_f}$ | $C_{V_f}$ | $r_{V_c}$ | $\sigma_{V_c}$ | $C_{V_c}$ | $Sph$ |
|---|---|---|---|---|---|---|---|---|
| - GSFC1 - | 0.300 | 0.148 | 0.380 | 0.048 | 3.187 | 0.750 | 0.023 | 100 |
| - GSFC2 - | 0.600 | 0.178 | 0.380 | 0.086 | 3.309 | 0.750 | 0.033 | 100 |
| - GSFC3 - | 0.900 | 0.208 | 0.380 | 0.123 | 3.432 | 0.750 | 0.043 | 100 |
| - MEXI1 - | 0.300 | 0.133 | 0.430 | 0.038 | 2.912 | 0.630 | 0.035 | 100 |
| - MEXI2 - | 0.600 | 0.144 | 0.430 | 0.072 | 3.080 | 0.630 | 0.066 | 100 |
| - MEXI3 - | 0.900 | 0.155 | 0.430 | 0.104 | 3.242 | 0.630 | 0.096 | 100 |
| - ZAMB1 - | 0.300 | 0.128 | 0.400 | 0.036 | 3.433 | 0.730 | 0.027 | 100 |
| - ZAMB2 - | 0.600 | 0.134 | 0.400 | 0.068 | 3.621 | 0.730 | 0.051 | 100 |
| - ZAMB3 - | 0.900 | 0.141 | 0.400 | 0.098 | 3.802 | 0.730 | 0.074 | 100 |
| - SOLV1 - | 0.300 | 0.120 | 0.400 | 0.022 | 2.320 | 0.600 | 0.125 | 0 |
| - SOLV2 - | 0.600 | 0.120 | 0.400 | 0.028 | 2.320 | 0.600 | 0.372 | 0 |
| - SOLV3 - | 0.900 | 0.120 | 0.400 | 0.033 | 2.320 | 0.600 | 0.629 | 0 |
| - BAHR1 - | 0.300 | 0.150 | 0.420 | 0.024 | 2.540 | 0.610 | 0.088 | 0 |
| - BAHR2 - | 0.600 | 0.150 | 0.420 | 0.044 | 2.540 | 0.610 | 0.201 | 0 |
| - BAHR3 - | 0.900 | 0.150 | 0.420 | 0.061 | 2.540 | 0.610 | 0.357 | 0 |
| - LANA1 - | 0.300 | 0.160 | 0.480 | 0.044 | 2.700 | 0.680 | 0.088 | 100 |
| - LANA2 - | 0.600 | 0.160 | 0.480 | 0.088 | 2.700 | 0.680 | 0.176 | 100 |
| | $n(340)$ | $n(380)$ | $n(440)$ | $n(500)$ | $n(675)$ | $n(870)$ | $n(1020)$ | $n(1640)$ |
| - GSFC - | $1.41-0.03\tau_a$ | $1.41-0.03\tau_a$ | $1.41-0.03\tau_a$ | $1.41-0.03\tau_a$ | $1.41-0.03\tau_a$ | $1.41-0.03\tau_a$ | $1.41-0.03\tau_a$ | $1.41-0.03\tau_a$ |
| - MEXI - | 1.47 | 1.47 | 1.47 | 1.47 | 1.47 | 1.47 | 1.47 | 1.47 |
| - ZAMB - | 1.51 | 1.51 | 1.51 | 1.51 | 1.51 | 1.51 | 1.51 | 1.51 |
| - SOLV - | 1.56 | 1.56 | 1.56 | 1.56 | 1.56 | 1.56 | 1.56 | 1.56 |
| - BAHR - | 1.55 | 1.55 | 1.55 | 1.55 | 1.55 | 1.55 | 1.55 | 1.55 |
| - LANA - | 1.36 | 1.36 | 1.36 | 1.36 | 1.36 | 1.36 | 1.36 | 1.36 |
| | $k(340)$ | $k(380)$ | $k(440)$ | $k(500)$ | $k(675)$ | $k(870)$ | $k(1020)$ | $k(1640)$ |
| - GSFC - | 0.003 | 0.003 | 0.003 | 0.003 | 0.003 | 0.003 | 0.003 | 0.003 |
| - MEXI - | 0.014 | 0.014 | 0.014 | 0.014 | 0.014 | 0.014 | 0.014 | 0.014 |
| - ZAMB - | 0.021 | 0.021 | 0.021 | 0.021 | 0.021 | 0.021 | 0.021 | 0.021 |
| - SOLV - | 0.0037 | 0.0032 | 0.0029 | 0.0022 | 0.0013 | 0.0010 | 0.0010 | 0.0005 |
| - BAHR - | 0.0035 | 0.0030 | 0.0025 | 0.0022 | 0.0014 | 0.0010 | 0.0010 | 0.0005 |
| - LANA - | 0.0015 | 0.0015 | 0.0015 | 0.0015 | 0.0015 | 0.0015 | 0.0015 | 0.0015 |





**Table 2.** Simulated aerosol optical depth values obtained from the aerosol examples described in table 1. The Ångström exponent, calculated from the wavelengths $\tau_a(440)$, $\tau_a(675)$ and $\tau_a(870)$, is also represented for each aerosol example. The last two columns include the values of the fine mode aerosol optical depth at 500 nm and the effective radius.

| | $\tau_a(340)$ | $\tau_a(380)$ | $\tau_a(440)$ | $\tau_a(500)$ | $\tau_a(675)$ | $\tau_a(870)$ | $\tau_a(1020)$ | $\tau_a(1640)$ | $\alpha$ | $\tau_f(500)$ | $r_{eff}$ |
|---|---|---|---|---|---|---|---|---|---|---|---|
| | | | | | | OUTPUT | | | | | |
| - GSFC1 - | 0.472 | 0.391 | 0.307 | 0.239 | 0.128 | 0.075 | 0.054 | 0.027 | 2.079 | 0.222 | 0.193 |
| - GSFC2 - | 0.867 | 0.739 | 0.602 | 0.480 | 0.269 | 0.157 | 0.111 | 0.047 | 1.967 | 0.457 | 0.223 |
| - GSFC3 - | 1.228 | 1.076 | 0.906 | 0.740 | 0.436 | 0.260 | 0.185 | 0.073 | 1.830 | 0.711 | 0.254 |
| - MEXI1 - | 0.442 | 0.372 | 0.298 | 0.237 | 0.136 | 0.085 | 0.065 | 0.041 | 1.837 | 0.212 | 0.211 |
| - MEXI2 - | 0.856 | 0.730 | 0.595 | 0.478 | 0.277 | 0.173 | 0.130 | 0.076 | 1.812 | 0.432 | 0.242 |
| - MEXI3 - | 1.263 | 1.091 | 0.903 | 0.733 | 0.432 | 0.269 | 0.201 | 0.110 | 1.770 | 0.671 | 0.260 |
| - ZAMB1 - | 0.460 | 0.387 | 0.309 | 0.243 | 0.134 | 0.080 | 0.059 | 0.032 | 1.977 | 0.225 | 0.200 |
| - ZAMB2 - | 0.880 | 0.745 | 0.602 | 0.477 | 0.266 | 0.158 | 0.115 | 0.060 | 1.958 | 0.445 | 0.209 |
| - ZAMB3 - | 1.286 | 1.100 | 0.899 | 0.719 | 0.406 | 0.241 | 0.174 | 0.086 | 1.924 | 0.675 | 0.220 |
| - SOLV1 - | 0.400 | 0.353 | 0.301 | 0.262 | 0.198 | 0.171 | 0.161 | 0.159 | 0.841 | 0.141 | 0.528 |
| - SOLV2 - | 0.724 | 0.670 | 0.601 | 0.557 | 0.482 | 0.458 | 0.446 | 0.466 | 0.410 | 0.174 | 0.907 |
| - SOLV3 - | 1.037 | 0.979 | 0.896 | 0.851 | 0.771 | 0.751 | 0.740 | 0.783 | 0.269 | 0.206 | 1.071 |
| - BAHR1 - | 0.392 | 0.350 | 0.301 | 0.258 | 0.181 | 0.140 | 0.123 | 0.106 | 1.130 | 0.209 | 0.340 |
| - BAHR2 - | 0.760 | 0.687 | 0.600 | 0.525 | 0.389 | 0.319 | 0.289 | 0.267 | 0.934 | 0.328 | 0.591 |
| - BAHR3 - | 1.134 | 1.028 | 0.902 | 0.794 | 0.598 | 0.498 | 0.454 | 0.425 | 0.881 | 0.455 | 0.683 |
| - LANA1 - | 0.420 | 0.364 | 0.302 | 0.253 | 0.172 | 0.132 | 0.114 | 0.092 | 1.228 | 0.179 | 0.380 |
| - LANA2 - | 0.840 | 0.727 | 0.603 | 0.507 | 0.344 | 0.263 | 0.229 | 0.183 | 1.228 | 0.358 | 0.380 |





**Table 3.** Differences obtained from the self-consistency test on GRASP-AOD code. In the first six columns the parameters that represent the bimodal log-normal size distribution (volume median radius ($r_{V_f}$, $r_{V_c}$ [$\mu$m]), geometric standard deviation ($\sigma_{V_f}$ and $\sigma_{V_c}$) and volume concentration ($C_{V_f}$, $C_{V_c}$ [$\mu$m$^3$/$\mu$m$^2$])) are represented. The following columns contain two secondary products derived from GRASP-AOD inversion and its corresponding reference values: fine mode aerosol optical depth at 500 nm ($\tau_f(500)$) and the effective radius ($r_{eff}$ [$\mu$m]).

|  | $r_{V_f}$ | $\sigma_{V_f}$ | $C_{V_f}$ | $r_{V_c}$ | $\sigma_{V_c}$ | $C_{V_c}$ | $\tau_f(500)$ | $r_{eff}$ |
|---|---|---|---|---|---|---|---|---|
| - GSFC1 - | 0.000 | 0.001 | 0.000 | -0.040 | -0.016 | 0.000 | 0.000 | 0.007 |
| - GSFC2 - | -0.002 | 0.008 | 0.001 | -0.008 | -0.050 | 0.000 | -0.001 | 0.004 |
| - GSFC3 - | -0.001 | 0.005 | 0.000 | -0.053 | -0.008 | 0.000 | 0.001 | 0.003 |
| - MEXI1 - | 0.003 | -0.019 | 0.000 | 0.240 | 0.068 | 0.002 | 0.001 | -0.011 |
| - MEXI2 - | 0.002 | -0.016 | -0.001 | 0.220 | 0.070 | 0.003 | 0.000 | -0.014 |
| - MEXI3 - | 0.001 | -0.016 | -0.001 | 0.246 | 0.070 | 0.003 | 0.002 | -0.007 |
| - ZAMB1 - | 0.000 | 0.001 | 0.000 | -0.178 | -0.021 | -0.001 | -0.001 | 0.002 |
| - ZAMB2 - | 0.000 | -0.002 | 0.000 | -0.323 | -0.029 | -0.004 | 0.000 | 0.000 |
| - ZAMB3 - | 0.000 | -0.002 | 0.000 | -0.362 | -0.029 | -0.006 | 0.000 | 0.007 |
| - SOLV1 - | -0.003 | 0.031 | 0.000 | 0.085 | 0.006 | 0.004 | 0.000 | 0.009 |
| - SOLV2 - | -0.003 | 0.021 | 0.001 | 0.280 | 0.039 | 0.044 | 0.003 | -0.061 |
| - SOLV3 - | 0.002 | -0.018 | -0.001 | -0.016 | -0.002 | -0.006 | -0.010 | -0.015 |
| - BAHR1 - | 0.000 | -0.004 | 0.000 | -0.039 | -0.005 | 0.000 | 0.000 | 0.000 |
| - BAHR2 - | 0.001 | -0.003 | -0.002 | -0.021 | -0.009 | 0.023 | -0.001 | 0.000 |
| - BAHR3 - | 0.000 | 0.015 | 0.001 | -0.123 | -0.026 | -0.012 | -0.005 | 0.027 |
| - LANA1 - | 0.006 | -0.028 | -0.001 | -0.152 | -0.037 | -0.004 | -0.001 | -0.009 |
| - LANA2 - | 0.009 | -0.059 | -0.005 | -0.278 | -0.029 | -0.016 | 0.000 | -0.014 |

**Table 4.** Initial guesses used as defaults for the GRASP-AOD retrieval. The values, as can be seen in the table, are given as a function of Ångström exponent and the aerosol optical depth at 440 nm.

| $\alpha$ | $r_{V_f}$ | $\sigma_{V_f}$ | $C_{V_f}$ | $r_{V_c}$ | $\sigma_{V_c}$ | $C_{V_c}$ |
|---|---|---|---|---|---|---|
| >1.5 | $0.13 + 0.05\tau_a(440)$ | 0.4 | $0.12\tau_a(440)$ | $3.0 + 0.5\tau_a(440)$ | 0.7 | $(0.48 - 0.2\,\alpha)\,\tau_a(440)$ |
| 1.0 – 1.5 | $0.13 + 0.05\tau_a(440)$ | 0.4 | $0.08\,\alpha\,\tau_a(440)$ | $\alpha + 1.5$ | 0.6 | $(0.78 - 0.4\,\alpha)\,\tau_a(440)$ |
| <1.0 | 0.12 | 0.4 | $(0.02 + 0.06\,\alpha)\,\tau_a(440)$ | 2.3 | 0.6 | $(0.78 - 0.4\,\alpha)\,\tau_a(440)$ |



**Table 5.** Values used for the initial guess tests. For each aerosol example, there are three possible initial guess value for the six retrieved parameters (volume median radius ($r_{V_f}$, $r_{V_c}$ [$\mu$m]), standard deviation ($\sigma_{V_f}$ and $\sigma_{V_c}$) and volume concentration ($C_{V_f}$, $C_{V_c}$ [$\mu$m$^3$/$\mu$m$^2$])).

| | $r_{V_f}$ | $\sigma_{V_f}$ | $C_{V_f}$ | $r_{V_c}$ | $\sigma_{V_c}$ | $C_{V_c}$ |
|---|---|---|---|---|---|---|
| - GSFC2 - | 0.12 - 0.16 - 0.20 | 0.3 - 0.4 - 0.5 | 0.044 - 0.072 - 1.008 | 2.50 - 3.30 - 4.10 | 0.6 - 0.7 - 0.8 | 0.030 - 0.050 - 0.070 |
| - MEXI2 - | 0.12 - 0.16 - 0.20 | 0.3 - 0.4 - 0.5 | 0.044 - 0.072 - 1.008 | 2.50 - 3.30 - 4.10 | 0.6 - 0.7 - 0.8 | 0.043 - 0.070 - 0.097 |
| - ZAMB2 - | 0.12 - 0.16 - 0.20 | 0.3 - 0.4 - 0.5 | 0.044 - 0.072 - 1.008 | 2.50 - 3.30 - 4.10 | 0.6 - 0.7 - 0.8 | 0.030 - 0.050 - 0.070 |
| - SOLV2 - | 0.09 - 0.12 - 0.15 | 0.3 - 0.4 - 0.5 | 0.016 - 0.027 - 0.038 | 1.80 - 2.30 - 2.80 | 0.5 - 0.6 - 0.7 | 0.220 - 0.370 - 0.520 |
| - BAHR2 - | 0.12 - 0.16 - 0.20 | 0.3 - 0.4 - 0.5 | 0.035 - 0.046 - 0.057 | 1.80 - 2.30 - 2.80 | 0.5 - 0.6 - 0.7 | 0.145 - 0.240 - 0.335 |
| - LANA2 - | 0.12 - 0.16 - 0.20 | 0.3 - 0.4 - 0.5 | 0.035 - 0.060 - 0.085 | 2.10 - 2.70 - 3.30 | 0.5 - 0.6 - 0.7 | 0.105 - 0.175 - 0.245 |

**Table 6.** Result of the multiple initial guess test on GRASP-AOD code. The averages and the standard deviations of the retrievals obtained from the study of the six primary parameters (volume median radius ($r_{V_f}$, $r_{V_c}$ [$\mu$m]), geometric standard deviation ($\sigma_{V_f}$ and $\sigma_{V_c}$) and volume concentration ($C_{V_f}$, $C_{V_c}$ [$\mu$m$^3$/$\mu$m$^2$])) are shown in the first columns (reference values in table 1). In the last two columns, the averages and the standard deviations of the two main secondary products (fine mode aerosol optical depth at 500 nm ($\tau_f(500)$) and the effective radius ($r_{eff}$ [$\mu$m]) are presented.

| | $r_{V_f}$ | $\sigma_{V_f}$ | $C_{V_f}$ | $r_{V_c}$ | $\sigma_{V_c}$ | $C_{V_c}$ | $\tau_f(500)$ | $r_{eff}$ |
|---|---|---|---|---|---|---|---|---|
| - GSFC2 - | $0.177 \pm 0.003$ | $0.393 \pm 0.031$ | $0.087 \pm 0.003$ | $3.300 \pm 0.326$ | $0.700 \pm 0.081$ | $0.033 \pm 0.003$ | $0.457 \pm 0.001$ | $0.220 \pm 0.011$ |
| - MEXI2 - | $0.146 \pm 0.003$ | $0.402 \pm 0.028$ | $0.070 \pm 0.002$ | $3.304 \pm 0.325$ | $0.698 \pm 0.078$ | $0.070 \pm 0.008$ | $0.431 \pm 0.002$ | $0.256 \pm 0.014$ |
| - ZAMB2 - | $0.135 \pm 0.001$ | $0.394 \pm 0.018$ | $0.068 \pm 0.001$ | $3.301 \pm 0.327$ | $0.700 \pm 0.081$ | $0.048 \pm 0.006$ | $0.445 \pm 0.001$ | $0.206 \pm 0.011$ |
| - SOLV2 - | $0.117 \pm 0.004$ | $0.422 \pm 0.058$ | $0.029 \pm 0.002$ | $2.589 \pm 0.241$ | $0.631 \pm 0.027$ | $0.415 \pm 0.037$ | $0.171 \pm 0.004$ | $0.978 \pm 0.123$ |
| - BAHR2 - | $0.151 \pm 0.002$ | $0.416 \pm 0.035$ | $0.043 \pm 0.001$ | $2.524 \pm 0.279$ | $0.597 \pm 0.041$ | $0.220 \pm 0.028$ | $0.329 \pm 0.003$ | $0.596 \pm 0.070$ |
| - LANA2 - | $0.168 \pm 0.006$ | $0.420 \pm 0.061$ | $0.083 \pm 0.006$ | $2.420 \pm 0.211$ | $0.647 \pm 0.042$ | $0.160 \pm 0.013$ | $0.358 \pm 0.004$ | $0.394 \pm 0.048$ |





**Table 7.** Result obtained from the simulation of errors in $\tau_a$ on GRASP-AOD code. The parameters representing the bimodal log-normal size distribution (volume median radius ($r_{V_f}$, $r_{V_c}$ [$\mu$m]), geometric standard deviation ($\sigma_{V_f}$ and $\sigma_{V_c}$) and volume concentration ($C_{V_f}$, $C_{V_c}$ [$\mu$m$^3$/$\mu$m$^2$])) are shown in the first six columns (reference values in table 1). The following columns contain two secondary products derived from GRASP-AOD inversion: fine mode aerosol optical depth at 500 nm ($\tau_f(500)$) and the effective radius ($r_{eff}$ [$\mu$m])(reference values in table 3).

| | $r_{V_f}$ | $\sigma_{V_f}$ | $C_{V_f}$ | $r_{V_c}$ | $\sigma_{V_c}$ | $C_{V_c}$ | $\tau_f(500)$ | $r_{eff}$ |
|---|---|---|---|---|---|---|---|---|
| - GSFC1 - | $0.142 \pm 0.013$ | $0.390 \pm 0.052$ | $0.051 \pm 0.005$ | $3.063 \pm 0.208$ | $0.696 \pm 0.056$ | $0.021 \pm 0.012$ | $0.222 \pm 0.014$ | $0.185 \pm 0.021$ |
| - GSFC3 - | $0.206 \pm 0.008$ | $0.398 \pm 0.045$ | $0.126 \pm 0.009$ | $3.430 \pm 0.108$ | $0.677 \pm 0.045$ | $0.043 \pm 0.018$ | $0.713 \pm 0.013$ | $0.252 \pm 0.032$ |
| - MEXI1 - | $0.134 \pm 0.011$ | $0.419 \pm 0.052$ | $0.038 \pm 0.004$ | $3.148 \pm 0.194$ | $0.687 \pm 0.044$ | $0.036 \pm 0.012$ | $0.211 \pm 0.013$ | $0.232 \pm 0.031$ |
| - MEXI3 - | $0.157 \pm 0.008$ | $0.405 \pm 0.041$ | $0.102 \pm 0.008$ | $3.173 \pm 0.226$ | $0.690 \pm 0.061$ | $0.092 \pm 0.019$ | $0.666 \pm 0.013$ | $0.264 \pm 0.034$ |
| - ZAMB1 - | $0.127 \pm 0.010$ | $0.412 \pm 0.054$ | $0.037 \pm 0.003$ | $3.248 \pm 0.251$ | $0.698 \pm 0.040$ | $0.024 \pm 0.012$ | $0.226 \pm 0.013$ | $0.190 \pm 0.025$ |
| - ZAMB3 - | $0.141 \pm 0.008$ | $0.391 \pm 0.047$ | $0.098 \pm 0.009$ | $3.226 \pm 0.254$ | $0.712 \pm 0.063$ | $0.063 \pm 0.017$ | $0.674 \pm 0.013$ | $0.212 \pm 0.028$ |
| - SOLV1 - | $0.119 \pm 0.009$ | $0.412 \pm 0.043$ | $0.023 \pm 0.001$ | $2.234 \pm 0.194$ | $0.601 \pm 0.052$ | $0.121 \pm 0.013$ | $0.139 \pm 0.011$ | $0.513 \pm 0.066$ |
| - SOLV3 - | $0.120 \pm 0.008$ | $0.395 \pm 0.042$ | $0.033 \pm 0.001$ | $2.280 \pm 0.216$ | $0.588 \pm 0.040$ | $0.623 \pm 0.075$ | $0.208 \pm 0.011$ | $1.103 \pm 0.111$ |
| - BAHR1 - | $0.150 \pm 0.009$ | $0.409 \pm 0.054$ | $0.026 \pm 0.002$ | $2.477 \pm 0.159$ | $0.615 \pm 0.057$ | $0.083 \pm 0.008$ | $0.208 \pm 0.011$ | $0.338 \pm 0.040$ |
| - BAHR3 - | $0.149 \pm 0.006$ | $0.422 \pm 0.060$ | $0.061 \pm 0.003$ | $2.474 \pm 0.168$ | $0.590 \pm 0.059$ | $0.357 \pm 0.037$ | $0.453 \pm 0.012$ | $0.681 \pm 0.058$ |
| - LANA1 - | $0.164 \pm 0.014$ | $0.437 \pm 0.042$ | $0.042 \pm 0.002$ | $2.476 \pm 0.278$ | $0.632 \pm 0.072$ | $0.083 \pm 0.010$ | $0.175 \pm 0.012$ | $0.386 \pm 0.055$ |
| - LANA2 - | $0.168 \pm 0.010$ | $0.425 \pm 0.041$ | $0.083 \pm 0.003$ | $2.472 \pm 0.272$ | $0.637 \pm 0.075$ | $0.161 \pm 0.018$ | $0.356 \pm 0.015$ | $0.395 \pm 0.045$ |

**Table 8.** Increment in the aerosol optical depth values (references in table 2) originated by changes of $\pm 2\sigma$ in the index n and k for the cases GSFC2, SOLV2 and ZAMB2 (reference values in table 1). The value of $2\sigma_n$ is equal to 0.02 for the cases GSFC and ZAMB while is 0.06 for SOLV. For the imaginary part, $2\sigma_k$ has values of 0.008, 0.006 and 0.002 for ZAMB, GSFC and SOLV respectively. Note that for both SOLV and GSFC, the case with $k - 2\sigma_k$ would be negative in some wavelengths. For those cases, k is fixed as zero and it is indicated by an asterisk in the table.

| | GSFC2 | | ZAMB2 | | SOLV2 | | GSFC2 | | ZAMB2 | | SOLV2 | |
|---|---|---|---|---|---|---|---|---|---|---|---|---|
| $\lambda$ [$nm$] | $-2\sigma_n$ | $+2\sigma_n$ | $-2\sigma_n$ | $+2\sigma_n$ | $-2\sigma_n$ | $+2\sigma_n$ | $-2\sigma_k^*$ | $+2\sigma_k$ | $-2\sigma_k$ | $+2\sigma_k$ | $-2\sigma_k^*$ | $+2\sigma_k$ |
| 340 | -0.054 | 0.052 | -0.042 | 0.040 | -0.053 | 0.052 | 0.000 | 0.000 | 0.000 | 0.000 | 0.000 | 0.000 |
| 380 | -0.051 | 0.050 | -0.038 | 0.038 | -0.046 | 0.046 | 0.000 | 0.001 | -0.003 | 0.003 | 0.000 | 0.000 |
| 440 | -0.046 | 0.046 | -0.034 | 0.034 | -0.038 | 0.039 | -0.002 | 0.004 | -0.006 | 0.006 | -0.001 | 0.000 |
| 500 | -0.039 | 0.039 | -0.027 | 0.028 | -0.029 | 0.030 | -0.003 | 0.006 | -0.007 | 0.007 | -0.001 | 0.000 |
| 675 | -0.023 | 0.024 | -0.015 | 0.015 | -0.012 | 0.012 | -0.004 | 0.007 | -0.008 | 0.008 | -0.005 | 0.004 |
| 870 | -0.013 | 0.013 | -0.008 | 0.008 | -0.005 | 0.005 | -0.004 | 0.006 | -0.007 | 0.007 | -0.005 | 0.004 |
| 1020 | -0.008 | 0.009 | -0.005 | 0.005 | -0.004 | 0.004 | -0.003 | 0.007 | -0.006 | 0.006 | -0.005 | 0.004 |
| 1640 | -0.002 | 0.002 | -0.001 | 0.001 | -0.002 | 0.002 | -0.002 | 0.003 | -0.004 | 0.003 | -0.004 | 0.004 |



**Table 9.** Fine mode aerosol optical depth at 500 $nm$ ($\tau_f(500)$) obtained varying by $\pm 2\sigma$ the refractive index during the retrieval process. The value of $2\sigma_n$ is spectrally independent and equal to 0.02 for the cases GSFC and ZAMB while is 0.06 for SOLV. For the imaginary part, $2\sigma_k$ is also spectrally independent and has values of 0.008, 0.006 and 0.002 for ZAMB, GSFC and SOLV respectively. Note that the case $-2\sigma_k$ produces negative values of $k$ for all the wavelengths in the aerosol example GSFC and for some in SOLV. In those cases, k is fixed as zero in the retrieval.

|  | GSFC1 | GSFC2 | GSFC3 | ZAMB1 | ZAMB2 | ZAMB3 | SOLV1 | SOLV2 | SOLV3 |
|---|---|---|---|---|---|---|---|---|---|
| $\boldsymbol{\Delta n, k = 0}$ | 0.222 | 0.457 | 0.711 | 0.225 | 0.445 | 0.675 | 0.141 | 0.174 | 0.206 |
| $\boldsymbol{+2\sigma_n}$ | 0.221 | 0.456 | 0.709 | 0.225 | 0.445 | 0.674 | 0.140 | 0.171 | 0.202 |
| $\boldsymbol{-2\sigma_n}$ | 0.222 | 0.458 | 0.713 | 0.226 | 0.446 | 0.676 | 0.142 | 0.175 | 0.211 |
| $\boldsymbol{+2\sigma_k}$ | 0.224 | 0.460 | 0.715 | 0.227 | 0.448 | 0.678 | 0.142 | 0.174 | 0.207 |
| $\boldsymbol{-2\sigma_k^*}$ | 0.221 | 0.456 | 0.710 | 0.224 | 0.443 | 0.671 | 0.142 | 0.174 | 0.206 |





**Figure 6.** Sensitivity of size distribution to refractive index for three of the aerosol examples: GSFC (top subfigures), ZAMB (middle subfigures) and SOLV (bottom subfigures). The subfigures on the left corresponds to the sensitivity analysis to the real part of refractive index, and on the right, to the imaginary part. Black solid lines are used to illustrate the reference values while gray dashed lines represent the retrievals without variations in the refractive index (self-consistency study). Red and blue lines are used to indicate the cases when refractive index is overestimated and underestimated respectively. In all the sub-figures, the variation of $\sigma$ is represented by solid lines and the variation of $2\sigma$ by dashed lines. The value of $2\sigma_n$ is spectrally independent and equal to 0.02 for the cases GSFC and ZAMB while is 0.06 for SOLV. For the imaginary part, $2\sigma_k$ has values of 0.008, 0.006 and 0.002 for ZAMB, GSFC and SOLV respectively, being constant for the different wavelengths as well. Note that the case $-2\sigma_k$ produces negative values of $k$ for all the wavelengths in the aerosol example GSFC and for some in SOLV. In those cases, k is fixed as zero in the retrieval. In the three aerosol cases we have considered the three aerosol loads $\tau_a(440) = 0.3, 0.6$ and $0.9$



**Table 10.** Summary of the 23 chosen days for the comparison between the aerosol products obtained from the new GRASP-AOD inversion and AERONET products. The first columns depict information about the day selected: site, date, photometer, AOD at 440 nm and Ångström exponent. The last two columns contain the number of piece of data used for each inversion type: ALM or almucantar used for AERONET inversion, and AOD used for both GRASP-AOD inversion and SDA algorithm.

| Cases | Photometer | Date | $< \tau_a(440) >$ | $< \alpha >$ | No. ALM | No. AOD |
|---|---|---|---|---|---|---|
| GSFC -A- | | 22/11/2009 | $0.166 \pm 0.015$ | $1.639 \pm 0.029$ | 8 | 39 |
| GSFC -B- | #451 (EXT) | 01/09/2010 | $0.323 \pm 0.027$ | $1.817 \pm 0.025$ | 8 | 30 |
| GSFC -C- | | 27/08/2009 | $0.493 \pm 0.042$ | $1.913 \pm 0.025$ | 5 | 19 |
| GSFC -D- | | 06/07/2010 | $0.590 \pm 0.049$ | $1.968 \pm 0.019$ | 4 | 24 |
| MEXI -A- | | 05/04/2003 | $0.246 \pm 0.009$ | $1.556 \pm 0.014$ | 4 | 16 |
| MEXI -B- | #10 (STD) | 07/12/2003 | $0.384 \pm 0.016$ | $1.728 \pm 0.029$ | 4 | 16 |
| MEXI -C- | | 28/04/2003 | $0.475 \pm 0.015$ | $1.798 \pm 0.026$ | 4 | 18 |
| MEXI -D- | | 09/05/2003 | $0.686 \pm 0.060$ | $1.603 \pm 0.011$ | 4 | 17 |
| ZAMB -A- | | 18/06/2007 | $0.249 \pm 0.025$ | $1.837 \pm 0.055$ | 9 | 43 |
| ZAMB -B- | #155 (STD) | 02/08/2006 | $0.422 \pm 0.029$ | $1.783 \pm 0.027$ | 10 | 46 |
| ZAMB -C- | | 25/07/2006 | $0.487 \pm 0.010$ | $1.917 \pm 0.019$ | 5 | 23 |
| ZAMB -D- | | 18/08/2006 | $0.733 \pm 0.026$ | $1.874 \pm 0.016$ | 9 | 45 |
| SOLV -A- | | 27/05/2007 | $0.249 \pm 0.006$ | $0.298 \pm 0.006$ | 5 | 20 |
| SOLV -B- | #125 (EXT) | 19/05/2007 | $0.422 \pm 0.020$ | $0.208 \pm 0.011$ | 7 | 38 |
| SOLV -C- | | 06/04/2007 | $0.532 \pm 0.051$ | $0.273 \pm 0.027$ | 8 | 35 |
| SOLV -D- | | 06/06/2007 | $0.595 \pm 0.020$ | $0.089 \pm 0.010$ | 6 | 35 |
| BAHR -A- | | 20/04/2005 | $0.301 \pm 0.027$ | $0.694 \pm 0.049$ | 4 | 17 |
| BAHR -B- | #130 (EXT) | 01/09/2005 | $0.423 \pm 0.030$ | $0.752 \pm 0.123$ | 4 | 17 |
| BAHR -C- | | 12/07/2005 | $0.579 \pm 0.014$ | $0.512 \pm 0.089$ | 4 | 18 |
| BAHR -D- | | 20/10/2005 | $0.722 \pm 0.029$ | $0.880 \pm 0.050$ | 6 | 24 |
| LANA -A- | | 21/03/2002 | $0.110 \pm 0.003$ | $0.819 \pm 0.048$ | 4 | 18 |
| LANA -B- | #107 (STD) | 23/03/2002 | $0.147 \pm 0.014$ | $0.784 \pm 0.108$ | 4 | 17 |
| LANA -C- | | 19/04/2002 | $0.232 \pm 0.022$ | $1.208 \pm 0.054$ | 4 | 17 |



**Table 11.** Daily average of standard parameters for fine and coarse modes retrieved using AERONET-standard and GRASP-AOD inversion: volume median radius ($r_{V_f}$, $r_{V_c}$ [$\mu$m]), standard deviation ($\sigma_{V_f}$ and $\sigma_{V_c}$) and volume concentration ($C_{V_f}$, $C_{V_c}$ [$\mu$m$^3$/$\mu$m$^2$]). Aerosol information about the 23 chosen days can be gained in table 10.

| | Type | $r_{V_f}$ | $\sigma_{V_f}$ | $C_{V_f}$ | $r_{V_c}$ | $\sigma_{V_c}$ | $C_{V_c}$ |
|---|---|---|---|---|---|---|---|
| GSFC -A- | - ALM - | 0.230 ± 0.016 | 0.503 ± 0.028 | 0.021 ± 0.002 | 2.905 ± 0.159 | 0.677 ± 0.019 | 0.004 ± 0.001 |
| | GRASP (EXT) | 0.203 ± 0.012 | 0.617 ± 0.018 | 0.023 ± 0.002 | 2.856 ± 0.034 | 0.751 ± 0.014 | 0.006 ± 0.001 |
| GSFC -B- | - ALM - | 0.180 ± 0.008 | 0.446 ± 0.019 | 0.043 ± 0.014 | 3.223 ± 0.075 | 0.608 ± 0.014 | 0.025 ± 0.003 |
| | GRASP (EXT) | 0.176 ± 0.007 | 0.430 ± 0.020 | 0.039 ± 0.008 | 3.174 ± 0.033 | 0.680 ± 0.018 | 0.027 ± 0.002 |
| GSFC -C- | - ALM - | 0.186 ± 0.012 | 0.433 ± 0.018 | 0.072 ± 0.022 | 3.256 ± 0.107 | 0.668 ± 0.012 | 0.020 ± 0.001 |
| | GRASP (EXT) | 0.187 ± 0.006 | 0.412 ± 0.008 | 0.066 ± 0.008 | 3.260 ± 0.039 | 0.686 ± 0.010 | 0.024 ± 0.003 |
| GSFC -D- | - ALM - | 0.193 ± 0.001 | 0.396 ± 0.015 | 0.070 ± 0.010 | 3.619 ± 0.261 | 0.658 ± 0.035 | 0.015 ± 0.002 |
| | GRASP (EXT) | 0.197 ± 0.004 | 0.356 ± 0.007 | 0.069 ± 0.008 | 3.390 ± 0.056 | 0.680 ± 0.024 | 0.018 ± 0.001 |
| MEXI -A- | - ALM - | 0.145 ± 0.002 | 0.363 ± 0.006 | 0.052 ± 0.002 | 3.671 ± 0.074 | 0.567 ± 0.008 | 0.057 ± 0.003 |
| | GRASP (STD) | 0.148 ± 0.005 | 0.393 ± 0.008 | 0.049 ± 0.002 | 3.181 ± 0.002 | 0.669 ± 0.005 | 0.039 ± 0.007 |
| MEXI -B- | - ALM - | 0.163 ± 0.005 | 0.435 ± 0.010 | 0.079 ± 0.003 | 3.425 ± 0.117 | 0.539 ± 0.011 | 0.033 ± 0.004 |
| | GRASP (STD) | 0.169 ± 0.005 | 0.407 ± 0.005 | 0.068 ± 0.003 | 3.270 ± 0.005 | 0.663 ± 0.003 | 0.035 ± 0.006 |
| MEXI -C- | - ALM - | 0.141 ± 0.004 | 0.376 ± 0.026 | 0.091 ± 0.013 | 3.493 ± 0.136 | 0.583 ± 0.015 | 0.062 ± 0.002 |
| | GRASP (STD) | 0.154 ± 0.003 | 0.384 ± 0.008 | 0.075 ± 0.002 | 3.255 ± 0.031 | 0.673 ± 0.006 | 0.038 ± 0.014 |
| MEXI -D- | - ALM - | 0.173 ± 0.008 | 0.440 ± 0.006 | 0.119 ± 0.014 | 3.401 ± 0.172 | 0.544 ± 0.028 | 0.063 ± 0.003 |
| | GRASP (STD) | 0.181 ± 0.003 | 0.388 ± 0.005 | 0.101 ± 0.007 | 3.444 ± 0.031 | 0.665 ± 0.021 | 0.050 ± 0.008 |
| ZAMB -A- | - ALM - | 0.148 ± 0.003 | 0.393 ± 0.027 | 0.029 ± 0.005 | 3.631 ± 0.240 | 0.654 ± 0.051 | 0.020 ± 0.002 |
| | GRASP (STD) | 0.138 ± 0.005 | 0.479 ± 0.055 | 0.032 ± 0.004 | 3.164 ± 0.029 | 0.698 ± 0.017 | 0.011 ± 0.005 |
| ZAMB -B- | - ALM - | 0.149 ± 0.003 | 0.367 ± 0.018 | 0.041 ± 0.007 | 3.563 ± 0.114 | 0.692 ± 0.024 | 0.035 ± 0.002 |
| | GRASP (STD) | 0.144 ± 0.004 | 0.368 ± 0.018 | 0.042 ± 0.002 | 3.286 ± 0.003 | 0.698 ± 0.001 | 0.033 ± 0.007 |
| ZAMB -C- | - ALM - | 0.141 ± 0.004 | 0.374 ± 0.022 | 0.053 ± 0.010 | 3.519 ± 0.074 | 0.734 ± 0.036 | 0.021 ± 0.001 |
| | GRASP (STD) | 0.144 ± 0.001 | 0.365 ± 0.021 | 0.051 ± 0.002 | 3.300 ± 0.002 | 0.707 ± 0.001 | 0.027 ± 0.006 |
| ZAMB -D- | - ALM - | 0.142 ± 0.002 | 0.383 ± 0.016 | 0.088 ± 0.008 | 3.496 ± 0.106 | 0.688 ± 0.044 | 0.048 ± 0.006 |
| | GRASP (STD) | 0.145 ± 0.002 | 0.366 ± 0.012 | 0.079 ± 0.003 | 3.392 ± 0.002 | 0.697 ± 0.001 | 0.049 ± 0.009 |
| SOLV -A- | - ALM - | 0.132 ± 0.006 | 0.537 ± 0.013 | 0.016 ± 0.001 | 2.140 ± 0.014 | 0.613 ± 0.011 | 0.140 ± 0.006 |
| | GRASP (EXT) | 0.122 ± 0.002 | 0.421 ± 0.012 | 0.012 ± 0.002 | 1.980 ± 0.031 | 0.748 ± 0.014 | 0.120 ± 0.002 |
| SOLV -B- | - ALM - | 0.144 ± 0.015 | 0.631 ± 0.010 | 0.025 ± 0.005 | 2.096 ± 0.081 | 0.570 ± 0.021 | 0.259 ± 0.017 |
| | GRASP (EXT) | 0.157 ± 0.025 | 0.481 ± 0.021 | 0.014 ± 0.002 | 1.992 ± 0.058 | 0.688 ± 0.024 | 0.215 ± 0.008 |
| SOLV -C- | - ALM - | 0.136 ± 0.007 | 0.595 ± 0.035 | 0.039 ± 0.003 | 2.102 ± 0.072 | 0.607 ± 0.024 | 0.298 ± 0.031 |
| | GRASP (EXT) | 0.157 ± 0.037 | 0.472 ± 0.037 | 0.025 ± 0.003 | 2.052 ± 0.171 | 0.721 ± 0.041 | 0.249 ± 0.044 |
| SOLV -D- | - ALM - | 0.168 ± 0.017 | 0.682 ± 0.024 | 0.020 ± 0.002 | 2.296 ± 0.074 | 0.549 ± 0.021 | 0.460 ± 0.029 |
| | GRASP (EXT) | 0.148 ± 0.014 | 0.484 ± 0.009 | 0.012 ± 0.002 | 2.003 ± 0.080 | 0.740 ± 0.035 | 0.401 ± 0.020 |
| BAHR -A- | - ALM - | 0.131 ± 0.010 | 0.521 ± 0.021 | 0.034 ± 0.005 | 1.872 ± 0.164 | 0.597 ± 0.039 | 0.077 ± 0.017 |
| | GRASP (EXT) | 0.141 ± 0.010 | 0.684 ± 0.161 | 0.042 ± 0.003 | 1.982 ± 0.072 | 0.671 ± 0.018 | 0.084 ± 0.019 |
| BAHR -B- | - ALM - | 0.137 ± 0.008 | 0.392 ± 0.013 | 0.032 ± 0.003 | 2.535 ± 0.042 | 0.644 ± 0.004 | 0.155 ± 0.019 |
| | GRASP (EXT) | 0.118 ± 0.008 | 0.470 ± 0.080 | 0.041 ± 0.004 | 2.364 ± 0.090 | 0.660 ± 0.046 | 0.144 ± 0.021 |
| BAHR -C- | - ALM - | 0.139 ± 0.020 | 0.550 ± 0.019 | 0.033 ± 0.001 | 2.077 ± 0.498 | 0.575 ± 0.042 | 0.303 ± 0.082 |
| | GRASP (EXT) | 0.133 ± 0.006 | 0.460 ± 0.010 | 0.029 ± 0.001 | 2.185 ± 0.085 | 0.615 ± 0.034 | 0.326 ± 0.026 |
| BAHR -D- | - ALM - | 0.153 ± 0.003 | 0.393 ± 0.013 | 0.060 ± 0.007 | 2.242 ± 0.042 | 0.580 ± 0.010 | 0.245 ± 0.018 |
| | GRASP (EXT) | 0.145 ± 0.005 | 0.476 ± 0.057 | 0.065 ± 0.004 | 2.320 ± 0.032 | 0.621 ± 0.014 | 0.230 ± 0.026 |
| LANA -A- | - ALM - | 0.156 ± 0.004 | 0.443 ± 0.008 | 0.014 ± 0.002 | 2.236 ± 0.079 | 0.699 ± 0.021 | 0.036 ± 0.003 |
| | GRASP (STD) | 0.159 ± 0.010 | 0.459 ± 0.026 | 0.011 ± 0.001 | 1.941 ± 0.163 | 0.792 ± 0.087 | 0.031 ± 0.004 |
| LANA -B- | - ALM - | 0.168 ± 0.004 | 0.429 ± 0.021 | 0.017 ± 0.005 | 2.145 ± 0.057 | 0.672 ± 0.009 | 0.058 ± 0.005 |
| | GRASP (STD) | 0.172 ± 0.011 | 0.439 ± 0.009 | 0.015 ± 0.003 | 2.135 ± 0.046 | 0.727 ± 0.031 | 0.054 ± 0.006 |
| LANA -C- | - ALM - | 0.284 ± 0.016 | 0.441 ± 0.014 | 0.032 ± 0.003 | 2.546 ± 0.203 | 0.611 ± 0.035 | 0.032 ± 0.008 |
| | GRASP (STD) | 0.258 ± 0.043 | 0.402 ± 0.021 | 0.029 ± 0.005 | 2.560 ± 0.018 | 0.612 ± 0.012 | 0.038 ± 0.020 |





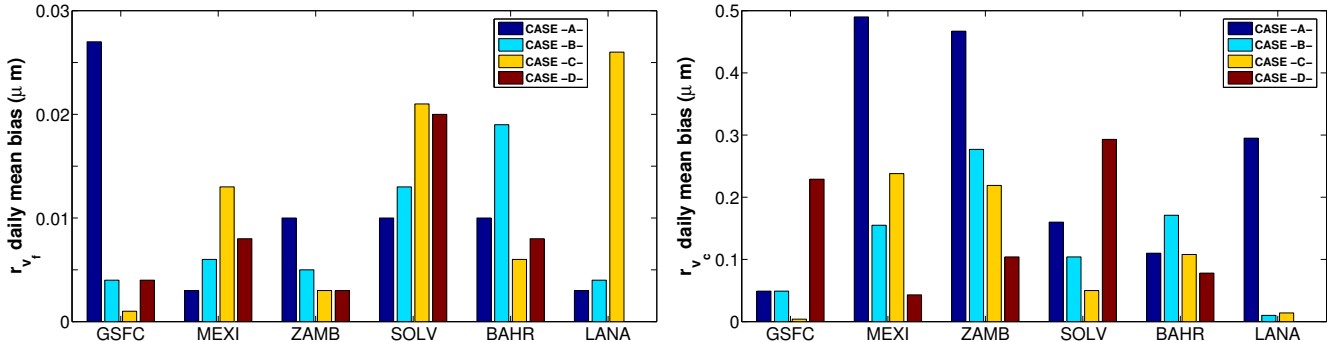

**Figure 7.** Absolute differences between the daily averages of the median volume radii retrieved by GRASP-AOD and AERONET standard inversions (total values can be found in table 11) for the examples considered in the analysis (table 10). The differences for the fine mode are represented in the left panel and for the coarse mode in the right panel. The four different days considered for each station (except for Lanai where we have considered only three) are represented with different colors: -A- in blue, -B- in turquoise, -C- in yellow and -D- in red.

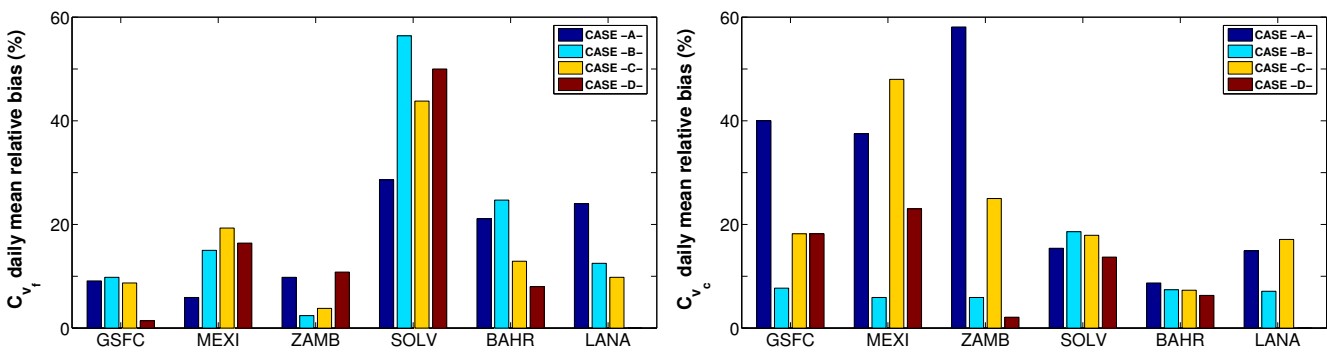

**Figure 8.** Relative differences between the daily averages of the volume concentrations retrieved by GRASP-AOD and AERONET standard inversions (total values can be found in table 11) for the examples considered in the analysis (table 10). The differences for the fine mode are represented in the left panel and for the coarse mode in the right panel. The four different days considered for each station (except for Lanai where we have considered only three) are represented with different colors: -A- in blue, -B- in turquoise, -C- in yellow and -D- in red.




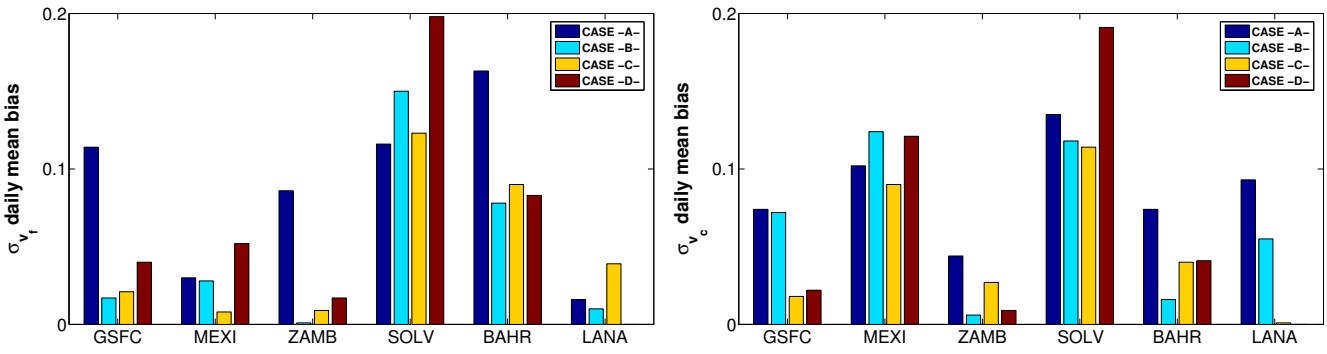

**Figure 9.** Absolute differences between the daily averages of the mode standard deviation retrieved by GRASP-AOD and AERONET standard inversions (total values can be found in table 11) for the examples considered in the analysis (table 10). The differences for the fine mode are represented in the left panel and for the coarse mode in the right panel. The four different days considered for each station (except for Lanai where we have considered only three) are represented with different colors: -A- in blue, -B- in turquoise, -C- in yellow and -D- in red.

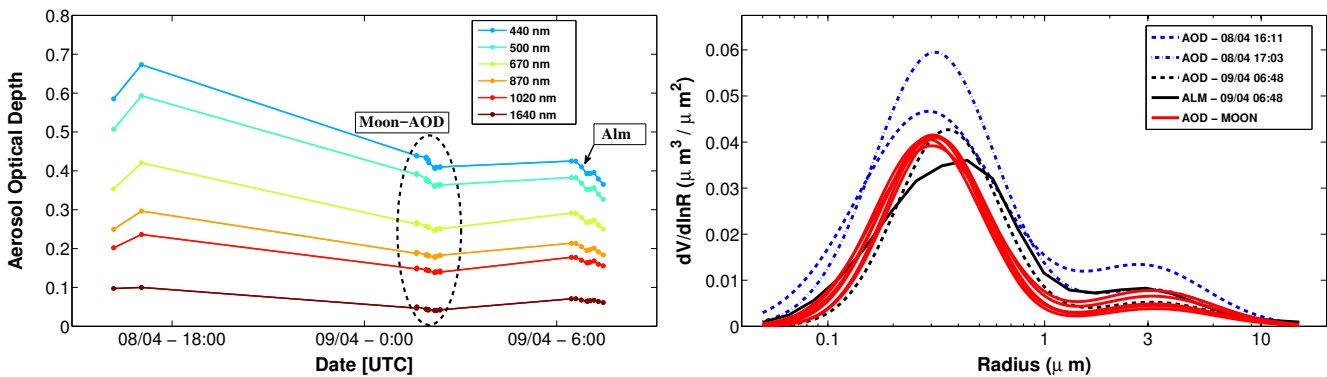

**Figure 10.** In the left, aerosol optical depth values measured at Lille site from the evening of 08/04/2015 until the morning of 09/04/2015. Measurements during daylight were taken by sun-photometer number #741 and night measurements were carried out by the moon-photometer number #841. In the right, size distribution retrieved by GRASP-AOD inversion using the AOD from the evening of 08/04/2015 (blue lines), the measurements during the night (red-lines) and the AOD corresponding to the almucantar in the morning of 09/04/2015. The size distribution from the AERONET standard inversion (6:48 AM) is also represented in the figure as a reference (black solid line).



**Table 12.** Daily averages of the retrieved products $< \tau_f(500) >$ and $r_{eff}$. The values of $< \tau_f(500) >$ are computed from the retrievals of AERONET standard and GRASP-AOD inversion, and from SDA algorithm for the 23 chosen days. The effective radius is computed from AERONET standard and GRASP-AOD inversion.

| Cases | $< \tau_f(500) >$ | | | $< r_{eff} >$ | |
|---|---|---|---|---|---|
| | AERONET | GRASP-AOD | SDA | AERONET | GRASP-AOD |
| GSFC -A- | $0.144 \pm 0.019$ | $0.136 \pm 0.013$ | $0.129 \pm 0.013$ | $0.240 \pm 0.026$ | $0.212 \pm 0.015$ |
| GSFC -B- | $0.262 \pm 0.022$ | $0.250 \pm 0.024$ | $0.254 \pm 0.025$ | $0.254 \pm 0.029$ | $0.260 \pm 0.009$ |
| GSFC -C- | $0.402 \pm 0.030$ | $0.389 \pm 0.037$ | $0.386 \pm 0.037$ | $0.215 \pm 0.020$ | $0.228 \pm 0.007$ |
| GSFC -D- | $0.472 \pm 0.049$ | $0.481 \pm 0.043$ | $0.481 \pm 0.044$ | $0.214 \pm 0.010$ | $0.229 \pm 0.004$ |
| MEXI -A- | $0.170 \pm 0.005$ | $0.176 \pm 0.009$ | $0.156 \pm 0.007$ | $0.271 \pm 0.012$ | $0.236 \pm 0.014$ |
| MEXI -B- | $0.296 \pm 0.007$ | $0.289 \pm 0.012$ | $0.276 \pm 0.013$ | $0.207 \pm 0.009$ | $0.229 \pm 0.015$ |
| MEXI -C- | $0.347 \pm 0.017$ | $0.359 \pm 0.008$ | $0.339 \pm 0.011$ | $0.210 \pm 0.013$ | $0.210 \pm 0.022$ |
| MEXI -D- | $0.536 \pm 0.042$ | $0.548 \pm 0.043$ | $0.541 \pm 0.040$ | $0.235 \pm 0.017$ | $0.243 \pm 0.015$ |
| ZAMB -A- | $0.186 \pm 0.022$ | $0.190 \pm 0.021$ | $0.169 \pm 0.017$ | $0.226 \pm 0.013$ | $0.176 \pm 0.020$ |
| ZAMB -B- | $0.323 \pm 0.025$ | $0.326 \pm 0.023$ | $0.322 \pm 0.024$ | $0.245 \pm 0.021$ | $0.230 \pm 0.015$ |
| ZAMB -C- | $0.384 \pm 0.008$ | $0.379 \pm 0.010$ | $0.369 \pm 0.008$ | $0.182 \pm 0.017$ | $0.200 \pm 0.015$ |
| ZAMB -D- | $0.565 \pm 0.020$ | $0.563 \pm 0.025$ | $0.560 \pm 0.022$ | $0.199 \pm 0.011$ | $0.212 \pm 0.016$ |
| SOLV -A- | $0.075 \pm 0.002$ | $0.064 \pm 0.001$ | $0.062 \pm 0.001$ | $0.731 \pm 0.049$ | $0.710 \pm 0.023$ |
| SOLV -B- | $0.117 \pm 0.003$ | $0.087 \pm 0.014$ | $0.099 \pm 0.037$ | $0.893 \pm 0.018$ | $0.972 \pm 0.013$ |
| SOLV -C- | $0.164 \pm 0.014$ | $0.143 \pm 0.040$ | $0.130 \pm 0.007$ | $0.721 \pm 0.054$ | $0.821 \pm 0.103$ |
| SOLV -D- | $0.130 \pm 0.009$ | $0.095 \pm 0.018$ | $0.097 \pm 0.007$ | $1.249 \pm 0.070$ | $1.159 \pm 0.061$ |
| BAHR -A- | $0.146 \pm 0.014$ | $0.149 \pm 0.011$ | $0.111 \pm 0.010$ | $0.346 \pm 0.084$ | $0.333 \pm 0.033$ |
| BAHR -B- | $0.207 \pm 0.025$ | $0.209 \pm 0.021$ | $0.166 \pm 0.015$ | $0.569 \pm 0.040$ | $0.416 \pm 0.039$ |
| BAHR -C- | $0.211 \pm 0.028$ | $0.202 \pm 0.008$ | $0.165 \pm 0.011$ | $0.793 \pm 0.131$ | $0.851 \pm 0.027$ |
| BAHR -D- | $0.380 \pm 0.013$ | $0.375 \pm 0.017$ | $0.328 \pm 0.013$ | $0.550 \pm 0.029$ | $0.481 \pm 0.042$ |
| LANA -A- | $0.060 \pm 0.002$ | $0.053 \pm 0.004$ | $0.045 \pm 0.002$ | $0.423 \pm 0.026$ | $0.427 \pm 0.060$ |
| LANA -B- | $0.072 \pm 0.013$ | $0.066 \pm 0.015$ | $0.054 \pm 0.012$ | $0.527 \pm 0.058$ | $0.535 \pm 0.055$ |
| LANA -C- | $0.182 \pm 0.018$ | $0.164 \pm 0.033$ | $0.169 \pm 0.020$ | $0.462 \pm 0.066$ | $0.479 \pm 0.097$ |





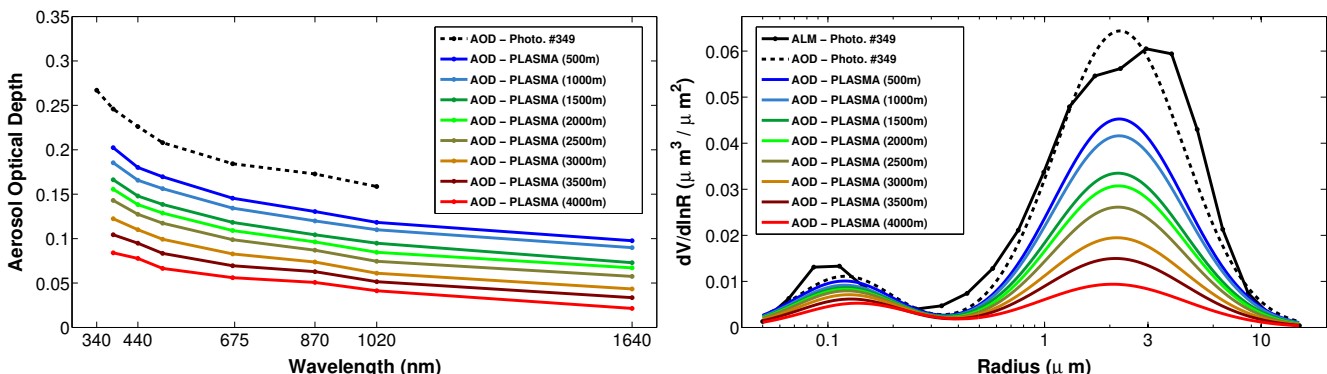

**Figure 11.** Aerosol Optical depth values measured by PLASMA airborne photometer at different height levels on the left, and the corresponding size distribution retrieved by GRASP-AOD inversion on the right. The mean values of the three AOD-data measured by the sun-photometer at AERONET site Cap d'en Font during the PLASMA profile are also represented and its corresponding GRASP-AOD inversion. The size distribution from the closest AERONET standard inversion (9:00 AM) is also represented in the figure as a reference.

**Table A1.** Observation wavelengths of the instruments used in the real data tests of GRASP-AOD inversion (section 4).

| Instrument type | Spectral channels ($nm$) |
|---|---|
| Cimel-318 Extended | 340, 380, 440, 500, 670, 870, 1020, 1640 |
| PLASMA | |
| Cimel-318 Standard | 340, 380, 440, 500, 670, 870, 1020 |
| Cimel-318U (Lunar-photometer) | 440, 500, 670, 870, 1020, 1640 |