# Peer review of "Advanced characterization of aerosol properties from measurements of spectral optical depth using the GRASP algorithm."

_Atmospheric Measurement Techniques, 2016_

## Short Comment (SC1) · 17 Jan 2017

See the attached file

Please also note the supplement to this comment:
http://www.atmos-meas-tech-discuss.net/amt-2016-334/amt-2016-334-SC1-supplement.zip

---

## Editor Comment (EC1) · F. Dulac (Editor) · 18 Jan 2017

I find you methodology interesting and well explained but I wish to question the 0.3-0.9 range of Tau-a(440) values for direct tests: on one side 0.3 is already a significant AOD, significantly larger than the average value in many AERONET stations, especially for places like Lanai where sea-salt is controlling the AOD; conversely, 0.9 does not cover the range of high values regularly observed at stations under desert dust influence; using such a limited range yields the reader to the conclusion that your algorithm is not consolidated at moderate and very high AOD; I would therefore suggest to include both a lower test value (e.g. 0.1) and a larger value for dust only (e.g. 1.5) to offer a better assessment of your algorithm range of application.

---

## Editor Comment (EC2) · F. Dulac (Editor) · 18 Jan 2017

Thank you for your submission to the ChArMEx Special Issue of AMT. I am, however, somewhat frustrated that application to ChArMEx data is only one small sub-section at the end of your manuscript. It is indeed of special interest because it refers to original airborne photometric data with the recent PLASMA instrument. But in order to better relate the study to other available ChArMEx data and justify publication as part of the ChArMEx Special issue, I recommend a more complete link to ChArMEx.

On one side, you miss referring to ChArMEx papers that discuss the size distribution and/or absorption properties of Mediterranean aerosols (e.g. Mallet et al., 2013, doi:10.5194/acp-13-9195-2013; Denjean et al., 2016, doi:10.5194/acp-16-1081-2016;

see also Fig. 14 in Renard et al., 2016, doi:10.5194/amt-9-1721-2016).

On the other side, you might consider a couple of Mediterranean AERONET sites with large data sets in your sensitivity tests. Following Mallet et al. (2013), I could suggest Rome (urban aerosols with relatively low omega-0) and Sede Boker (dusty site with the longest time series in the Med. region). It seems to me that comparisons might also be worth being tested with AERONET inversions at ChArMEx super sites of Ersa and Lampedusa. Other case studies, such as those of July 2012 documented by lidar by Granadoz-Muñoz et al. (2016; doi: 10.5194/acp-16-7043-2016) might also be considered.

Other technical comments:

-p.19, line 15: please specify ">40$^\circ$".

-In the bibliography, please replace "n/a-n/a" by the paper number given before the year of publication in several JGR papers (Kaskaoutis et al., 2009; Kim et al., 2004; O'Neil et al., 2003; Schmid et al., 2003; Schuster et al., 2006; Smirnov et al., 2009).

---

## Referee Comment (RC1) · Anonymous Referee #1 · 19 Jan 2017

The paper deals with the retrieval of aerosol microphysical properties from spectral aerosol optical depth (AOD) using the Generalized Retrieval of Aerosol and Surface Properties (GRASP) algorithm. The main objectives of the paper are well described and discussed. GRASP is becoming a very powerful tool for aerosol characterization from remote sensing measurements and the inclusion of this new capability is of great interest for the scientific community. There are very large databases of spectral AODs alone compared with the classical inversion that also requires sky-radiances measurements. The accuracy and errors in the retrievals are well presented, and it is shown how the final products are below 20% uncertainty. The inclusion of AOD measurements at 1640 nm is very interesting and actually allows retrievals of coarse

mode with good accuracy. This point should be pointed out more as an improvement to previous developments. The applicability to nighttime photometric measurements is great an interestingly presented as such measurements can only provide AODs. Night-time measurements are expecting to increase with the recent developments in moon photometry. Therefore, the research presented in the article is recommended to be published in Atmospheric Measurement Techniques.

However, in my opinion, the paper needs improvements before its final publication. Although is generally well structured, the writing can be improved as there are many unnecessary discussion (e.g. in page 7, lines 20-30 about the multi-pixel capabilities of GRASP seem out of context) and repetitions. Also, there are many editing errors and English misspellings. My major scientific concerns are:

As commented, the use of 1640 nm provides very good retrievals of coarse mode. However, many AERONET measurements do not include this filter. A discussion (extra analyzed if required) about the use of the classical spectral range of measurements 380-1020 nm need to be included. I also agree with the editor that the analysis should be extended to lower and higher AOD values.

It is not clear to me which approach you eventually use about spherical/non-spherical particles. Is it critical for the retrieval?

I also have minor concerns that could be useful to improve the manuscript.

The final products of the approach presented are the parameters of a log-normal bi-modal size distribution. If I am right, you need to retrieve first the size distribution. So I do not understand well what the improvement instead of using 22 bins is. Please clarify.

The authors show the dependence of size distribution with refractive index, but such dependences are within the error claimed. However, it is not clear to me how they select the input refractive index for experimental measurements.

It is not clear to me how you obtained refractive indexes of Table 1. If you used AERONET sky-radiance inversion, how did you obtain values in 340 and 380 nm.

Figure 2 does not show something new and could be removed. Also, section 3.5.1. 'Pre-analysis with the forward code' can be shortened as it is well known by the scientific community.

In my opinion, Appendixes can be skipped and references are enough.

The authors reference many times the results of Dubovik et al., (2002). Why not including a table that summarizes the main results used in the current manuscript? Page 16 lines 14-15. Please add a reference.

Section 3.4 Simulation of aerosol optical depth errors. For wavelengths below 400 nm AERONET instruments have errors of 0.02. Also, moon photometers might have errors of 0.02 or even higher. I suggest adding a brief discussion about the effects of these larger errors.

Section 3.5.2. I miss a general conclusion or a table that summarizes the conclusions. What are the final results adding all the errors you computed?

The variables in Table 3 are confusing. Please choose another way to remark that are differences between model and retrieved parameters.

Figure 10: Please make the points corresponding to experimental measurements bigger. In their current shape they are difficult to see. The same happens in Figure 11.

Why not showing temporal evolution of radius and volume concentrations retrieved in Figures 10 and 11?

Why did you not say anything about your applications to experimental measurements in the conclusions section?

---

## Referee Comment (RC2) · M. King (Referee) · 20 Jan 2017

Review of "Advanced characterization of aerosol properties from measurements of spectral optical depth using the GRASP algorithm" by B. Torres, O. Dubovik, D. Fuertes, G. Schuster, V. E. Cachorro, T. Lapyonok, P. Goloub, L. Blarel, A. Barreto, M. Mallet, C. Toledano, and D. Tanré

Recommendation: This paper presents a sensitivity analysis and application of determining the aerosol size distribution from spectral measurements of aerosol optical thickness, in particular determining 6 parameters of a bimodal aerosol size distribution. After the sensitivity study, it applies the method to real observations from AERONET, airborne measurements, and lunar transmission measurements. The paper is extremely thorough and well written. I recommend that this paper be accepted for publication with only minor editorial changes.

General Comments: This important paper is very easy to read and covers an important topic of determining the aerosol size distribution from (primarily) ground-based measurements of spectral optical thickness in the absence of almucantar measurement. The treatment of calibration (optical thickness), initial guess, and refractive index sensitivity is clearly presented.

Major Comments: All references are missing full initials of authors (like middle initials). The references should be complete in this regard. Please check.

Minor Comments: Page 2, line 12 – change 'one on the first type of measurements' to 'one of the first types of measurements.'

Page 3, line 25 – change ''airborne photometers' to 'airborne sunphotometers'.

Page 3, line 30 – change 'at polar regions' to 'in polar regions'. Later in same sentence, change 'aerosol load' to 'aerosol loading'.

Page 4, line 12 – Consider adding an additional reference to a recent paper by van Donkelaar et al., 2016:

van Donkelaar, A., Martin, R. V., Brauer, M., Hsu, N. C., Kahn, R. A., Levy, R. C., Lyapustin, A., Sayer, A. M., and Winker, D. M.: Global estimates of fine particulate matter using a combined geophysical-statistical method with information from satellites, models, and monitors. Environmental Science & Technology, 50,3762-3772, 2016.

Page 5, line 21 – Consider adding an additional reference to a book chapter by King and Dubovik:

King, M. D., and Dubovik, O.: Determination of aerosol optical properties from inverse methods. Aerosol Remote Sensing, J. Lenoble, L. Remer, and D. Tanré, Eds., Springer-Verlag, 101–136, 2013.

Page 6, Equation (2) – v(r) in equation not defined.

Page 9, Figure 1 – the line types (dashed, and dot-dashed) are hard to distinguish. Please improve.

Page 21, line 21 – reference to 'Chapter 2 of Lenoble et al. 2013' should be to 'Lenoble et al. 2013':

Lenoble, J., Mishchenko, M. I.., and Herman, M.: Absorption and scattering by molecules and particles. Aerosol Remote Sensing, J. Lenoble, L. Remer, and D. Tanré, Eds., Springer-Verlag, 13–51, 2013.

Page 13, line 5 – change 'regardless the initial guess' to 'regardless of the initial guess'.

Page 15, line 5 – after analysis, add reference back to Table 2 (i.e., 'analysis (cf. Table 2). The standard deviation. . .'

Page 16, line 18 – change 'aerosol cases are here' to 'aerosol cases here'.

Page 17, line 32ff – the fact that the refractive index (mostly real part) affects the derived size distribution was also discussed in King et al. (1978), where it was pointed out that the shape of the size distribution remains the same but shifts with a varying real part (with little sensitivity to the imaginary part). This is derived from anomalous diffraction theory of van de Hulst, and was also discussed in Yamamoto and Tanaka (1969).

Page 20, line 11 – change 'MEXI-A' to 'MEXI-C' if I understand this correctly.

Page 20, line 15 – change 'thought' to 'though'.

Page 31, line 25 – the Kieffer and Stone reference needs to have the page range of the publication, not just the first page.

---

## Referee Comment (RC3) · Z.-Q. Li (Referee) · 23 Jan 2017

Review of "Advanced characterization of aerosol properties from measurements of spectral optical depth using the GRASP algorithm" by B. Torres, O. Dubovik, D. Fuertes, G. Schuster, V. E. Cachorro, T. Lapyonok, P. Goloub, L. Blarel, A. Barreto, M. Mallet, C. Toledano, and D. Tanré

General comments: The authors propose to evaluate the using spectral aerosol optical depth (AOD) measurements to characterize the microphysical and optical properties of atmospheric aerosols, based on the new developed Generalized Retrieval of Aerosol and Surface Properties (GRASP) code. The approach is interesting and deserves to be explored for its potential for the retrieval of particle size distribution. The paper is

well written and easy to read.

Major comments: (1) For methodology of this paper, to retrieve the parameters of size distribution, the complex refractive index must be known. However, the refractive index is also a very important parameter, and change with the different place and time. When the GRASP-AOD method can be used, the synchronous measurements of Sun photometer (CE318) are needed to obtain the refractive index. Thus, if the measurements of CE318 are considered, the parameters of size distribution and refractive index could be retrieved directly in daytime, and needn't to refer GRASP-AOD method. By contrast, if the measurements of Lunar photometer are considered, the simultaneous results of refractive index can't be obtained correctly. Therefore, please make clear for the application scenario of GRASP-AOD method in the paper. (2) Just as shown in Table A1, the observation wavelengths of the instruments (CE318, PLASMA and Cimel-318U) are different, thus the information content are also different. However, the authors didn't analyze the retrieval errors for different instruments, please also make clear for this part in the paper.

Minor comments: (1) Page 2 line 12: "one on the first type" → "one of the first type". (2) Page 3 line 31 "aerosol load" → "aerosol loading". (3) Please give definition for the parameter "v(r)" in Eq.(2). (4) Please give definition for the parameter "Nr" in Eq.(3). (5) Page 10 section 3.2: Please add more details for the description of Figure 3 about "Forward Code" and "Inversion Code", it would be better to show the methodology detailed based on the GRASP code. (6) Page 13 line 6: "regardless" → "regardless of". (7) Page 18 line 19: "sub-figures" → "subfigures". (8) Page 19 line 15: ">40" → ">40°". (9) Page 20 line 15: "thought" → "though". (10) Page 31-33: Some expression such as "n/a-n/a" in references should be modified with the software Endnote. (11) Page 34 line 1: "Optica Pura y Aplicada" → "Optica Puray Aplicada". (12) Page 44 Figure 8: Due the captions of Figure 8 is the same as these of Figure 7, thus the expression should be concise, and please change to "Similar as Figure 7, but for the volume concentration". (13) Page 45 Figure 9: change the caption of Figure 9 to "Similar as Figure 7, but for

the geometric standard deviation".

---

## Referee Comment (RC4) · T. F. Eck (Referee) · 25 Jan 2017

Review for Atmospheric Measurement Techniques

Title: Advanced characterization of aerosol properties from measurements of spectral optical depth using the GRASP algorithm.

Authors: Benjamin Torres, Oleg Dubovik, David Fuertes, Gregory Schuster, Victoria Eugenia Cachorro, Tatsiana Lapyonok, Philippe Goloub, Luc Blarel, Africa Barreto, Marc Mallet, Carlos Toledano, and Didier Tanré

General Comments: This paper describes a new application of the GRASP algorithm that utilizes accurate spectral AOD measurements from sunphotometer instruments

(both ground-based and airborne) to retrieve aerosol size distributions. The paper is clearly organized and very well written. This new algorithm provides inversions that may be extremely useful in analysis of particle size related aerosol parameters, since these retrievals can be made much more frequently from AOD spectra alone than from the combined input of AOD spectra and sky radiance distributions, such as in almucantar inversions in AERONET (the Dubovik and King (2000) algorithm).

The sensitivity analysis is clear and thorough, however I am surprised that the perturbations in real refractive index applied in this section are so small. Real refractive index variations of +-0.02 (for fine mode dominated cases; section 3.5.1) are somewhat trivial and it would seem very difficult to provide with such accuracy when day-to-day (and seasonal) variation in real refractive index can be expected at some sites (such as at GSFC). This is due to hygroscopic growth of particles related to variation in relative humidity and low altitude cloud interaction with aerosol, plus relative changes in aerosol species composition variation. Ideally, the analysis should include much larger uncertainty/variation in the initial guess of the real refractive index.

I agree with the editor and other reviewers that you should add the case of AOD(440 nm)=0.1 to your sensitivity analysis section, since most data in the AERONET network are for AOD levels <0.3. Especially for the Lanai site the AOD=0.3 is much too high to be representative. This is a background marine site where AOD is predominately <0.15. The monthly mean AODs at Lanai do not exceed 0.08 at 440 nm except for the dust transport season in spring when they reach a monthly average maximum of 0.12 in April. I also agree with the editor that it would be useful to include very high AOD cases in your sensitivity analysis (1.5 or 2 at 440 nm) for dust and also fine mode smoke, since these cases are important for analysis of major aerosol events.

The estimation of uncertainty in AERONET measured AOD for field instruments of 0.01 in the visible and NIR and increasing to 0.02 in the UV wavelengths should be mentioned on Page 13 or elsewhere in the manuscript. This wavelength dependent uncertainty was taken into account for operational SDA inversions where only the 380

to 870 nm wavelengths (5 channels) were utilized as input to the algorithm in order to avoid somewhat higher uncertainties in the 340 nm AOD due to both calibration and filter instability (degradation in time), and for 1020 nm due to temperature sensitivity of the detector at this wavelength and water vapor absorption. It seems that you should make some tests of wavelength range used as input to the code (with accounting for realistic uncertainty for each wavelength) to determine the optimal set of AOD spectral data that should be used as input, plus characterize how the uncertainty changes when specific wavelengths are not available. In particular some AERONET Cimels operated by PHOTONS only have the 440, 675, 870 and 1020 nm channels for AOD measurement. How is the algorithm performance affected when only these 4 wavelengths are available as input to the GRASP-AOD algorithm?

For the Real Cases (Section 4) analysis it is important again that you include cases with AOD of 0.1 at 440 nm for all sites, and for the Lanai site a case of AOD(440)=0.07 since this is the climatological mean AOD (for non-dust months, i.e. excluding spring season) for that site. Additionally you should include some analysis of sites that are not included in the Dubovik et al. (2002) Table 1. The vast majority of AERONET sites are not characterized in that table so if the algorithm is to be operationally applied a strategy for selection of first guesses of the input parameters is needed. The way the paper is written it sometimes implies using the almucantar retrievals as a source of first guesses for refractive indices and size, therefore it is presented (in some sections) as a companion retrieval to the almucantar sky radiances retrievals and not a fully independent retrieval such as SDA. This should be discussed or clarified in the paper.

Specific Comments:

Title: You say "...characterization of aerosol properties..." but shouldn't this be more specific, since only aerosol size related properties are retrieved as the refractive indices are assumed a priori. Therefore it seems that perhaps the word 'size' should be inserted between 'aerosol' and 'properties' in the title.

[Figure]

Abstract lines 10-11: "Differences in the fine mode volume median radii for the GRASP-AOD and AERONET inversions are less than 0.02 $\mu$m at sites dominated by the fine mode for all cases..." Is this the maximum difference or mean difference?

Abstract line 16: I suggest removing the word 'advance' in this sentence since it is confusing.

Introduction, Page 2, line 4: I suggest that you remove 'Unfortunately' from this sentence, as it is somewhat confusing.

Page 3, lines 4-8: Please provide more details and justification for utilizing bi-modal assumption with rv and sigmas for each mode to describe the size distributions rather than use 22 size bins as is done in the AERONET standard retrieval. I assume that it is a more stable retrieval than the 22 bins, given only the AOD spectra as input, but it would be informative to provide some more details on why you selected this way to make the retrievals (either here or in Section 3).

Page 3, lines 11-13: "For instance, AERONET includes around forty direct Sun measurements per day in its standardized sequence of measurements (in cloudless conditions); only about eight of these sequences are coincident with sky-radiance measurements that are suitable as input to the AERONET inversion code." It would be useful to include information on the new Cimels here, where Hybrid scans make sky radiance retrievals possible each hour throughout the day thereby doubling or more the total number of sky-radiance retrievals. Also many new Cimels are run in the turbo mode for AOD measurement where the time interval of AOD sampling is 3 minutes rather than the 15 minutes for the standard scenario in a Cimel sun-sky radiometer. Therefore the $\sim$40 measurements per day increases to about 200 AOD spectra per day (depending on latitude and season, therefore day length) with new Cimels.

Page 3, lines 13-14: "...a large amount of data containing only direct Sun measurements which are not used apart from the characterization of the aerosol load." I disagree, since the Angstrom Exponent and it's spectral variation are also useful parameters providing basic aerosol size information that are computed by AERONET from the spectral AOD, and are utilized in many studies.

Page 3, lines 14-15: "Moreover, many AERONET sites are plagued by several months of partial cloudiness (especially in wintertime)." I think this is somewhat misleading, since some sites have the seasonal opposite and some tropical sites are cloudy for most seasons. Therefore I suggest dropping the phrase '(especially in wintertime)' from the sentence.

Page 3, lines 14-15: "As a result, the aerosol loads are the only data reported at many sites. . ." Please insert after 'aerosol loads' something like: 'and Angstrom Exponents that parameterize the relative fine versus coarse mode optical influence, depending on the wavelength range used in the computation of AE (Eck et al. 1999; Schuster et al., 2006).'

Page 4, lines 11-12: Please include the reference of van Donkelaar et al. (2010).

Page 6, lines 4-6: For GRASP-AOD please discuss whether there would be any difference in the retrievals if spherical particle shape assumption was utilized/assumed versus spheroidal shape.

Page 7, lines 7-9: In most cases this is true, that lognormal bi-modal size distribution assumptions are sufficient. However, for cloud-processed aerosol a third middle mode sometimes exists, see Eck et al. 2012 for Dubovik almucantar retrievals that are tri-modal, not bi-modal, and these middle modes are supported by independent in situ measurements of fog/cloud processed aerosols. This should be mentioned as a relatively rare case that does occur however.

Page 8, lines 12: Note that Dubovik 2002 presents an AERONET Version 1 database climatology that has been refined significantly with the AERONET Version 2 reprocessing of the entire database that occurred in 2006.

Page 8, lines 16-17: This is weakly absorbing aerosol at GSFC, non-absorbing aerosol

would have SSA=1.

Page 8, lines 17-18: It should be mentioned that this is strongly absorbing biomass burning (BB) aerosol at Mongu, Zambia and that BB aerosol range from very weakly absorbing to strongly absorbing (see Eck et al, 2003, GRL) depending on fuel types and phase of combustion (flaming versus smoldering).

Page 8, lines 27-28: Why would you assume that for a mixed case (Bahrain) that all particles are non-spherical? Solar Village also has many mixed mode aerosol days with predominately spherical fine mode particles from pollution.

Page 8, lines 12-13: Surprising that you do not show the case of AOD=0.9 in Figure 2 since this is the one where the curvature for the fine mode cases would become the most obvious, since the fine mode radius is largest. Note that the 1640 nm AOD showing a departure from a 2nd order fit of AOD versus WL in Figure 2 is not observed in the real GSFC site AERONET measurement data. Perhaps you can show the 2nd order fit to your simulated data in Figure 2 and also show the delta AOD departures from the fit.

Page 9, lines 8-9: Note that the Angstrom Exponents you computed for the Lanai site are much too high, you have 1.22 in Table 2, while the measurement data yield averages of $\sim$0.6 to 0.8 for Lanai data (see the AERONET climatology tables). This possibly suggests an issue with the Dubovik 2002 climatology parameters for this site. Also note that for the Solar Village site only the 0.6 and 0.9 AOD cases can be considered dust dominated, since the AOD=0.3 case is mixed with Angstrom Exponent=0.84 in Table 2.

Page 13, line 29-31: The estimation of uncertainty for AERONET measured AOD for field instruments is 0.01 in the visible and NIR and increasing to 0.02 in the UV wavelengths (Eck et al. 1999). This spectral variation in AOD uncertainty should be mentioned here or elsewhere in the paper.

[Figure]

Page 14, line 8: This is an awkward way to say that uncertainties increase at lower AOD. For the reader it would be much clearer if you did not use abbreviations such as GSFC1 and GSFC3 in the text but instead referred to the AOD levels and associated uncertainties.

Page 19, line 5: Should insert the Dubovik et al. (2006) reference here since these significant algorithm advances were incorporated in the AERONET operational inversions in 2006.

Page 19, line 6: Please add the reference Smirnov et al. (2000 ) here for Level 2 data in Version 2.

Page 19, line 22-23: Should say that these sites did not have the 1640 nm channel of the newer Cimels, as many readers will not know what you mean by 'extended' photometer.

Page 19, line 31-33: I suggest that an additional, perhaps more robust way to compare the retrievals of the fine mode fraction (FMF) of optical depth is to provide a scatterplot of all individual (not just daily means) GRASP-AOD retrievals regardless of SZA (not just time matched to almucantars) for many sites (including sites not in the Dubovik 2002 table) compared to the AERONET almucantar retrievals of FMF. This should include all AOD levels measured at each site. Both can be plotted as a function of AE(440-870) on the x-axis. Regression fit and statistics can then also be shown. Other parameters such as volume median radius could also be compared as scatterplots using time-matched individual retrievals from many sites, with the AERONET standard sky-radiance inversion on the x-axis and the GRASP-AOD inversion on the y-axis.

Page 20, lines 10-11: Need to mention the likely reason for this large difference for the GSFC-A case. It is the lowest AOD of all sites with AOD(440)=0.166. This is important and needs to be quantified further with cases that have lower AOD values such as 0.10 or lower.

[Figure]

Page 20, line 13: This is an anomalously large rv fine for a low AOD case at GSFC. Note that the climatology of GSFC size distributions (as a function of AOD) in Eck et al. (2012; Figure 17 b) and in Dubovik et al. (2002) strongly suggest that this case is an anomaly, therefore a relatively poor choice for a case study. The actual values of the almucantar retrieved individual rv fine at GSFC for this date (Nov 22, 2009) range from 0.21 to 0.26 micron (still very high for this AOD level) so I don't know where you got the 0.27 micron number from.

Page 21, lines 17-20: Should note here in the text that differences in the definition of modes, radius cutoff in AERONET standard retrievals versus tails of modes included in SDA explain a least a portion of these differences.

Page 22, lines 9-10: Need to note that the moon measurements have a spectral range from 440 to 1640 nm, while earlier in the paper you made the sensitivity analysis for the 340 nm to 1640 nm spectral range (see Table 2).

Page 23, lines 20-21: Are you referring to pollution or smoke from biomass burning or a mixture of the two types of aerosols?

Page 24, lines 8-9: Please give the rv for both the fine and coarse modes for the AERONET standard almucantar inversion, for comparison purposes to the GRASP-AOD inverted size distributions.

Page 28, lines 2-3: Calibration uncertainty for AERONET master instruments is better than you stated here. The uncertainty in Vo due to calibration by Langley analysis at Mauna Loa is ∼0.25% in the visible and NIR and 0.5% in the UV. The resulting total uncertainty in AOD (additionally including some other sources of uncertainty) is ∼0.002 to 0.009 for Master Cimels (lower values in visible and NIR and higher in UV ; see Eck et al. 1999). This is for overhead sun (SZA=0 and optical airmass=1) and the uncertainty in measured AOD is less as solar zenith angle increases (optical airmass increases).

Page 28, lines 8-9: Should cite Smirnov et al. (2000) here since the Version 2 Level 2 data are cloud screened and quality controlled as described in this reference.

---

## Author Comment (AC1) · 18 Jun 2017

Just to be above board, this is Norm O'Neill. You should consider the elaborations / corrections that I have made below to the paragraph where the SDA is described in your text (I had tracking on in Word).

O'Neill et al. (2003) developed the Spectral Deconvolution Algorithm (SDA) to discriminate fine and coarse mode extinction at a reference wavelength. That study employed the 0th, 1st and 2nd order mathematical (differential) equations describing a bi-modal, particle size distribution (O'Neill et al., 2001b) to arrive at an expression for the fine Angstrom exponent (a pure spectral derivative) and, in turn, the fine AOD (from

the 1st order Angstrom type equation) and the coarse AOD (from the 0th order AOD equation). The set of 3 equations were then solved for fine and coarse parameters given the total AOD along with its 1st and 2nd order spectral derivatives as input. This solution involved two 2nd order approximations: prescribing low fixed values for the coarse Ångström exponent and its derivative and the use of an empirical relation (between the fine Ångström exponent and its derivative). The algorithm is part of the AERONET processing chain : the value of the fine and coarse AOD at 500 nm is retrieved from every measured AOD spectrum and provided as a standard product of the network (full description in http://aeronet.gsfc.nasa.gov/new_web/PDF/tauf_tauc_technical_memo1.pdf).

Thank you very much for your corrections. We have added the text exactly as you have suggested. We have acknowledged your collaboration at the end of the manuscript.

---

## Author Comment (AC2) · 18 Jun 2017

Interactive comment on "Advanced characterization of aerosol properties from measurements of spectral optical depth using the GRASP algorithm" by B. Torres et al. Anonymous Referee 1 The paper deals with the retrieval of aerosol microphysical properties from spectral aerosol optical depth (AOD) using the Generalized Retrieval of Aerosol and Surface Properties (GRASP) algorithm. The main objectives of the paper are well described and discussed. GRASP is becoming a very powerful tool for aerosol characterization from remote sensing measurements and the inclusion of this new capability is of great interest for the scientific

community. There are very large databases of spectral AODs alone compared with the classical inversion that also requires sky-radiances measurements. The accuracy and errors in the retrievals are well presented, and it is shown how the final products are below $20\%$ uncertainty. The inclusion of AOD measurements at 1640 nm is very interesting and actually allows retrievals of coarse mode with good accuracy. This point should be pointed out more as an improvement to previous developments. The applicability to night-time photometric measurements is great an interestingly presented as such measurements can only provide AODs. Night- time measurements are expecting to increase with the recent developments in moon photometry. Therefore, the research presented in the article is recommended to be published in Atmospheric Measurement Techniques. However, in my opinion, the paper needs improvements before its final publication. Although is generally well structured, the writing can be improved as there are many unnecessary discussion (e.g. in page 7, lines 20-30 about the multi-pixel capabilities of GRASP seem out of context) and repetitions. Also, there are many editing errors and English misspellings.

The editing errors and English misspellings pointed out by the different referees have been corrected. The multi pixel approach is recognised as one of the major advances presented by GRASP code. We consider that the fact that we have not used this capability in this study (not being in the the scope of the work) should be commented. At the same time, we would like to note that checking the possible use in the future could be interesting.

My major scientific concerns are: As commented, the use of 1640 nm provides very good retrievals of coarse mode. However, many AERONET measurements do not include this filter. A discussion (extra analyzed if required) about the use of the classical spectral range of measurements 380-1020 nm need to be included. I also agree with the editor that the analysis should be extended to lower and higher AOD values.

We have added 4 new aerosol cases in the sensitivity analysis: GSFC0 and LANA0 with $\tau(440) = 0.1$ and SOLV4 ZAMB4 with $\tau(440) = 1.5$ (low and large aerosol optical load). We have included a new study for the spectral range of the old polarized photometers from $440$ to $1020$ nm (new section 3.4.3).

It is not clear to me which approach you eventually use about spherical/non-spherical particles. Is it critical for the retrieval?

We have added that we assume the sphericity parameter in the inversion strategy description. We use a sphericity parameter of $0$ (i.e., all particles are non-spherical) for the cases with Ångström exponent smaller than $0.6$. We use a linear approximation with respect to the Ångström exponent to select intermediate values of the sphericity parameter for the cases with Ångström exponents between $0.6$ and $1.1$ (for instance in the sensitivity analysis: SOLV1, BAHR2, and BAHR3). That is, we use sphericity parameters of 0 and 100 for Ångström exponents of 0.6 and 1.1 (respectively), and linearly interpolate the intermediate values. For the cases with Ångström exponents greater than $1.1$, we fixed the sphericity parameter at $100$ (considering all the particles as spheres). At the same time we have included the subsection "3.6 Variations on the sphericity parameter", where the effects of this assumption are described.

I also have minor concerns that could be useful to improve the manuscript. The final products of the approach presented are the parameters of a log-normal bi-modal size distribution. If I am right, you need to retrieve first the size distribution. So I do not understand well what the improvement instead of using 22 bins is. Please clarify.

We have assumed simplified bimodal lognormal functions to directly retrieve size distributions, and we do not pass from 22bins functions. We have specific Kernels to make this possible.

The authors show the dependence of size distribution with refractive index, but such dependences are within the error claimed. However, it is not clear to me how they select the input refractive index for experimental measurements.

To run the GRASP-AOD inversions, we have assumed climatological values of the

refractive index for the different sites in Table 13. This is now clearly stated in the text. In any case, in real applications of the code, we would suggest exploring other alternatives (to use the data from the closest AERONET standard inversion (if there are in the same day), monthly climatologies, etc.)

It is not clear to me how you obtained refractive indexes of Table 1. If you used AERONET sky-radiance inversion, how did you obtain values in 340 and 380 nm.

We did it by extrapolating the values in those cases where they are not constant.

Figure 2 does not show something new and could be removed. Also, section 3.5.1. 'Pre-analysis with the forward code' can be shortened as it is well known by the scientific community. In my opinion, Appendixes can be skipped and references are enough.

Figure 2, section 3.5.1 and the Appendix were commented and corrected by other referees. We would like to keep them in their current form if the editor agrees.

The authors reference many times the results of Dubovik et al., (2002). Why not including a table that summarizes the main results used in the current manuscript?

Please note that Table 1 and Table 2 already summarize the main result that we use in the manuscript.

Page 16 lines 14-15. Please add a reference.

We have tried to retrieve the refractive index in preliminary tests but we quickly realized that the solutions were not stable. Note the difficulty in retrieving 22 parameters (16 refractive index+ 6 parameters describing the size distribution) from only 8 measurements. Moreover, the section 3.5.1 itself shows the low sensibility of aerosol optical depth to refractive index compared to parameters that describe the size distribution.

Section 3.4 Simulation of aerosol optical depth errors. For wavelengths below 400 nm AERONET instruments have errors of 0.02. Also, moon photometers might have errors of 0.02 or even higher. I suggest adding a brief discussion about the effects of these

larger errors.

We have repeated the tests in section 3.4 (Simulation of aerosol optical depth errors) assuming double uncertainty for the wavelengths $340$, $380$ and $1020$ nm. The latter was included due to its dependency on the temperature.

Section 3.5.2. I miss a general conclusion or a table that summarizes the conclusions. What are the final results adding all the errors you computed?

We think that we would not get a realistic uncertainty if we directly added all the errors/differences computed from the sensitivity tests, since they have rather different nature: in some cases we treat assumptions, in others errors in the measurements, etc. Moreover, note that in some cases we have exaggerated the uncertainties in the assumption, given that our main interest was to better understand their effects.

The variables in Table 3 are confusing. Please choose another way to remark that are differences between model and retrieved parameters.

We have added the symbol $\Delta$ to indicate that we talk about differences.

Figure 10: Please make the points corresponding to experimental measurements bigger. In their current shape they are difficult to see. The same happens in Figure 11.

We have modified the figures. Thank you.

Why not showing temporal evolution of radius and volume concentrations retrieved in Figures 10 and 11?

We have stated in the text that in both cases they do not vary considerably. We think that the manuscript is already overloaded and showing a table with such little variations would be unnecessary.

Why did you not say anything about your applications to experimental measurements in the conclusions section?

We have added the following sentence in the conclusions: "Finally, we have presented two practical applications of the GRASP-AOD code: some retrievals during night from moon photometer data and retrievals from data obtained at different heights with the airborne sun-tracking photometer PLASMA."
* * *

---

## Author Comment (AC3) · 18 Jun 2017

Interactive comment on "Advanced characterization of aerosol properties from measurements of spectral optical depth using the GRASP algorithm" by B. Torres et al. M. King (Referee) michael.king@lasp.colorado.edu Review of "Advanced characterization of aerosol properties from measurements of spectral optical depth using the GRASP algorithm" by B. Torres, O. Dubovik, D. Fuertes, G. Schuster, V. E. Cachorro, T. Lapyonok, P. Goloub, L. Blarel, A. Barreto, M. Mallet, C. Toledano, and D. Tanré Recommendation: This paper presents a sensitivity analysis and application of determining the aerosol size distribution from spectral

measurements of aerosol optical thickness, in particular determining 6 parameters of a bimodal aerosol size distribution. After the sensitivity study, it applies the method to real observations from AERONET, airborne measurements, and lunar transmission measurements. The paper is extremely thorough and well written. I recommend that this paper be accepted for publication with only minor editorial changes. General Comments: This important paper is very easy to read and covers an important topic of determining the aerosol size distribution from (primarily) ground-based measurements of spectral optical thickness in the absence of almucantar measurement. The treatment of calibration (optical thickness), initial guess, and refractive index sensitivity is clearly presented. Major Comments: All references are missing full initials of authors (like middle initials). The references should be complete in this regard. Please check.

We have used BibTeX for managing the references. In some of the references, a space was missing between the initials and the compiler was recognizing only the first name. The mistake has been corrected. Thank you very much.

Minor Comments: Page 2, line 12 – change 'one on the first type of measurements' to 'one of the first types of measurements.'

Changed, thanks.

Page 3, line 25 – change ''airborne photometers' to 'airborne sunphotometers'.

Changed, thanks.

Page 3, line 30 – change 'at polar regions' to 'in polar regions'. Later in same sentence, change 'aerosol load' to 'aerosol loading'.

Changed, thanks.

Page 4, line 12 – Consider adding an additional reference to a recent paper by van Donkelaar et al., 2016: van Donkelaar, A., Martin, R. V., Brauer, M., Hsu, N. C., Kahn, R. A., Levy, R. C., Lyapustin, A., Sayer, A. M., and Winker, D. M.: Global estimates of fine particulate matter using a combined geophysical-statistical method with information from satellites, models, and monitors. Environmental Science Technology, 50,3762-3772, 2016.

Added, thanks.

Page 5, line 21 – Consider adding an additional reference to a book chapter by King and Dubovik: King, M. D., and Dubovik, O.: Determination of aerosol optical properties from in- verse methods. Aerosol Remote Sensing, J. Lenoble, L. Remer, and D. Tanré, Eds., Springer-Verlag, 101–136, 2013.

Added, thanks.

Page 6, Equation (2) – v(r) in equation not defined.

$\nu(r)$ is the volume of particle. Added, thank you

Page 9, Figure 1 – the line types (dashed, and dot-dashed) are hard to distinguish. Please improve.

We have changed the order between them, and we think that now, it is easy to distinguish them.

Page 11, line 21 – reference to 'Chapter 2 of Lenoble et al. 2013' should be to 'Lenoble et al. 2013': Lenoble, J., Mishchenko, M. I.., and Herman, M.: Absorption and scattering by molecules and particles. Aerosol Remote Sensing, J. Lenoble, L. Remer, and D. Tanré, Eds., Springer-Verlag, 13–51, 2013.

Changed. Thank you.

Page 13, line 5 – change 'regardless the initial guess' to 'regardless of the initial guess'.

Changed. Thank you.

Page 15, line 5 – after analysis, add reference back to Table 2 (i.e., 'analysis (cf. Table 2). The standard deviation. . .'

Added. However, you may find important changes in the subsection.

Page 16, line 18 – change 'aerosol cases are here' to 'aerosol cases here'.

Changed thanks.

Page 17, line 32ff – the fact that the refractive index (mostly real part) affects the derived size distribution was also discussed in King et al. (1978), where it was pointed out that the shape of the size distribution remains the same but shifts with a varying real part (with little sensitivity to the imaginary part). This is derived from anomalous diffraction theory of Van de Hulst, and was also discussed in Yamamoto and Tanaka (1969).

We have added this interesting result in the discussion. Thank you very much for your comment.

Page 20, line 11 – change 'MEXI-A' to 'MEXI-C' if I understand this correctly.

The whole paragraph was modified in the new version.

Page 20, line 15 – change 'thought' to 'though'.

The whole paragraph was modified in the new version.

Page 31, line 25 – the Kieffer and Stone reference needs to have the page range of the publication, not just the first page.

We have corrected it. Thank you.

---

## Author Comment (AC4) · 18 Jun 2017

Interactive comment on "Advanced characterization of aerosol properties from measurements of spectral optical depth using the GRASP algorithm" by B. Torres et al. Z.-Q. Li (Referee) lizq@radi.ac.cn Review of "Advanced characterization of aerosol properties from measurements of spectral optical depth using the GRASP algorithm" by B. Torres, O. Dubovik, D. Fuertes, G. Schuster, V. E. Cachorro, T. Lapyonok, P. Goloub, L. Blarel, A. Barreto, M. Mallet, C. Toledano, and D. Tanré General comments: The authors propose to evaluate the using spectral aerosol optical depth (AOD) measurements to characterize the microphysical

and optical properties of atmospheric aerosols, based on the new developed Generalized Retrieval of Aerosol and Surface Properties (GRASP) code. The approach is interesting and deserves to be explored for its potential for the retrieval of particle size distribution. The paper is well written and easy to read. Major comments: (1) For methodology of this paper, to retrieve the parameters of size distribution, the complex refractive index must be known. However, the refractive index is also a very important parameter, and change with the different place and time. When the GRASP-AOD method can be used, the synchronous measurements of Sun photometer (CE318) are needed to obtain the refractive index. Thus, if the measurements of CE318 are considered, the parameters of size distribution and refractive index could be retrieved directly in daytime, and needn't to refer GRASP-AOD method. By contrast, if the measurements of Lunar photometer are considered, the simultaneous results of refractive index can't be obtained correctly. Therefore, please make clear for the application scenario of GRASP-AOD method in the paper.

To run the GRASP-AOD inversions, we have assumed climatological values of the refractive index for the different sites in Table 13. This is now clearly stated in the text. In any case, in real applications of the code, we would also suggest exploring other alternatives (to use the data from the closest AERONET standard inversion (if there are in the same day), monthly climatologies, etc.)

(2) Just as shown in Table A1, the observation wavelengths of the instruments (CE318, PLASMA and Cimel-318U) are different, thus the information content are also different. However, the authors didn't analyze the retrieval errors for different instruments, please also make clear for this part in the paper.

We have repeated the tests in section 3.4 (Simulation of aerosol optical depth errors) assuming double uncertainty for the wavelengths $340, 380$ and $1020$ nm. The latter was included given its dependency on the temperature.

Minor comments: (1) Page 2 line 12: "one on the first type" → "one of the first type".

Changed, thanks.

(2) Page 3 line 31 "aerosol load" → "aerosol loading".

Changed, thanks.

(3) Please give definition for the parameter "v(r)" in Eq.(2).

$\nu(r)$ is the volume of particle. Added, thank you

(4) Please give definition for the parameter "Nr" in Eq.(3).

$N_r$ is the number of bins used to represent the size distribution. Added, thank you

(5) Page 10 section 3.2: Please add more details for the description of Figure 3 about "Forward Code" and "Inversion Code", it would be better to show the methodology detailed based on the GRASP code.

Some details of the Forward code and Inversion code are giving in the section 2. Here, we just pretend to describe the methodology used in the tests

(6) Page 13 line 6: "regardless" → "regardless of".

Changed, thanks.

(7) Page 18 line 19: "sub-figures" → "subfigures".

Changed, thanks.

(8) Page 19 line 15: ">40" → ">40âŮę".

Changed, thanks.

(9) Page 20 line 15: "thought" → "though".

The whole paragraph was modified.

(10) Page 31-33: Some expression such as "n/a-n/a" in references should be modified with the software Endnote.

We have corrected the problems with the bibliography.

(11) Page 34 line 1: "Optica Pura y Aplicada" → "Optica Puray Aplicada".

The name of the journal is "Optica Pura y Aplicada". Note that –y- in Spanish means –and- in English and it should go separated.

(12) Page 44 Figure 8: Due the captions of Figure 8 is the same as these of Figure 7, thus the expression should be concise, and please change to "Similar as Figure 7, but for the volume concentration". (13) Page 45 Figure 9: change the caption of Figure 9 to "Similar as Figure 7, but for the geometric standard deviation".

If the editor agrees, we would like to keep the whole descriptions since sometimes they are relative or absolute differences depending on the parameter.

---

## Author Comment (AC5) · 19 Jun 2017

Interactive comment on "Advanced characterization of aerosol properties from measurements of spectral optical depth using the GRASP algorithm" by B. Torres et al. T. F. Eck (Referee) thomas.f.eck@nasa.gov" Review for Atmospheric Measurement Techniques Title: Advanced characterization of aerosol properties from measurements of spectral optical depth using the GRASP algorithm. Authors: Benjamin Torres, Oleg Dubovik, David Fuertes, Gregory Schuster, Victoria Eugenia Cachorro, Tatsiana Lapyonok, Philippe Goloub, Luc Blarel, Africa Barreto, Marc Mallet, Carlos Toledano, and Didier Tanré

[Figure]

General Comments: This paper describes a new application of the GRASP algorithm that utilizes accurate spectral AOD measurements from sunphotometer instruments (both ground-based and airborne) to retrieve aerosol size distributions. The paper is clearly organized and very well written. This new algorithm provides inversions that may be extremely useful in analysis of particle size related aerosol parameters, since these retrievals can be made much more frequently from AOD spectra alone than from the combined input of AOD spectra and sky radiance distributions, such as in almucantar inversions in AERONET (the Dubovik and King (2000) algorithm). The sensitivity analysis is clear and thorough, however I am surprised that the perturbations in real refractive index applied in this section are so small. Real refractive index variations of +-0.02 (for fine mode dominated cases; section 3.5.1) are somewhat trivial and it would seem very difficult to provide with such accuracy when day-to-day (and seasonal) variation in real refractive index can be expected at some sites (such as at GSFC). This is due to hygroscopic growth of particles related to variation in relative humidity and low altitude cloud interaction with aerosol, plus relative changes in aerosol species composition variation. Ideally, the analysis should include much larger uncertainty/variation in the initial guess of the real refractive index.

We have carried out a series of test with real refractive index variations up to $\pm0.05$ for the example of GSFC. We have observed a linear behaviour between the real refractive index variation and the error produced in the retrieved parameters that characterize the fine mode. Thus, it is possible to approximate the error in $r_{V_f}$ and $C_{V_f}$ as: $\Delta r_{V_f} = -0.04\,\Delta n$ and $\Delta C_{V_f} = -0.27\,\Delta n\,\tau_a(440)$. In relative terms, these differences in $r_{V_f}$ represents between $\mp12-13\%$ while in $C_{V_f}$ between $\mp8-10\%$ for the maximum variation considered of $\pm5\sigma_n$.

I agree with the editor and other reviewers that you should add the case of AOD(440 nm)=0.1 to your sensitivity analysis section, since most data in the AERONET network are for AOD levels <0.3. Especially for the Lanai site the AOD=0.3 is much too high to be representative. This is a background marine site where AOD is predominately

[Figure]

<0.15. The monthly mean AODs at Lanai do not exceed 0.08 at 440 nm except for the dust transport season in spring when they reach a monthly average maximum of 0.12 in April. I also agree with the editor that it would be useful to include very high AOD cases in your sensitivity analysis (1.5 or 2 at 440 nm) for dust and also fine mode smoke, since these cases are important for analysis of major aerosol events.

The estimation of uncertainty in AERONET measured AOD for field instruments of 0.01 in the visible and NIR and increasing to 0.02 in the UV wavelengths should be mentioned on Page 13 or elsewhere in the manuscript. This wavelength dependent uncertainty was taken into account for operational SDA inversions where only the 380 to 870 nm wavelengths (5 channels) were utilized as input to the algorithm in order to avoid somewhat higher uncertainties in the 340 nm AOD due to both calibration and filter instability (degradation in time), and for 1020 nm due to temperature sensitivity of the detector at this wavelength and water vapor absorption. It seems that you should make some tests of wavelength range used as input to the code (with accounting for realistic uncertainty for each wavelength) to determine the optimal set of AOD spectral data that should be used as input, plus characterize how the uncertainty changes when specific wavelengths are not available. In particular some AERONET Cimels operated by PHOTONS only have the 440, 675, 870 and 1020 nm channels for AOD measurement. How is the algorithm performance affected when only these 4 wavelengths are available as input to the GRASP-AOD algorithm?

The multiterm LSM formulation allows to account for the uncertainty of each wavelength. Therefore, in real applications of the code, there may be two alternatives: to introduce the known uncertainties of each wavelength in the covariance matrix (Eq 4 and 5) or to reduce the spectral range. Following your advices, we have repeated the tests in section 3.4 (Simulation of aerosol optical depth errors) assuming double uncertainty for the wavelengths $340, 380$ and $1020$ nm. At the same time, we have re-done the study for the spectral range of the old polarized photometers $440$ to $1020$ nm (new section 3.4.3).

For the Real Cases (Section 4) analysis it is important again that you include cases with AOD of 0.1 at 440 nm for all sites, and for the Lanai site a case of AOD(440)=0.07 since this is the climatological mean AOD (for non-dust months, i.e. excluding spring season) for that site. Additionally you should include some analysis of sites that are not included in the Dubovik et al. (2002) Table 1. The vast majority of AERONET sites are not characterized in that table so if the algorithm is to be operationally applied a strategy for selection of first guesses of the input parameters is needed.

The new sites considered in the Section 4, Lampedusa and Rome Tor Vergata, include study cases with $\tau(440) < 0.2$. Thus, four of the eight analysed sites include one case with $\tau(440) < 0.2$ and all the sites has at least one case with $\tau(440) \leq 0.3$. In the simulation study for low aerosol conditions, we point out that the uncertainty for the parameters representing the bimodal log-normal size distribution increase a lot when the aerosol load is low. For practical uses of GRASP-AOD, it would be convenient to establish a lower limit in the $\tau_a$ values for quality assured retrievals. On the other hand, the capacity to discriminate between fine and coarse mode extinction remains stable, in absolute terms, and it is related to the value of $\tau_a$-errors.

The way the paper is written it sometimes implies using the almucantar retrievals as a source of first guesses for refractive indices and size, therefore it is presented (in some sections) as a companion retrieval to the almucantar sky radiances retrievals and not a fully independent retrieval such as SDA. This should be discussed or clarified in the paper.

To run the GRASP-AOD inversions, we have assumed climatological values of the refractive index for the different sites in Table 13. This is now clearly stated in the text. In any case, in real applications of the code, we would suggest exploring other alternatives (to use the data from the closest AERONET standard inversion (same day), monthly climatologies, etc.)

Specific Comments: Title: You say ". . .characterization of aerosol properties. . ." but

shouldn't this be more specific, since only aerosol size related properties are retrieved as the refractive indices are assumed a priori. Therefore it seems that perhaps the word 'size' should be inserted between 'aerosol' and 'properties' in the title.

Changed, thanks.

Abstract lines 10-11: "Differences in the fine mode volume median radii for the GRASP-AOD and AERONET inversions are less than 0.02 $\mu$m at sites dominated by the fine mode for all cases. . ." Is this the maximum difference or mean difference?

We have redone the sentence: Differences in the fine mode volume median radii for the GRASP-AOD and AERONET inversions are in average $0.010$ $\mu$m at sites dominated by the fine mode and $0.013$ $\mu$m including all cases.

Abstract line 16: I suggest removing the word 'advance' in this sentence since it is confusing.

Removed, thanks.

Introduction, Page 2, line 4: I suggest that you remove 'Unfortunately' from this sentence, as it is somewhat confusing.

Removed, thanks.

Page 3, lines 4-8: Please provide more details and justification for utilizing bi-modal assumption with rv and sigmas for each mode to describe the size distributions rather than use 22 size bins as is done in the AERONET standard retrieval. I assume that it is a more stable retrieval than the 22 bins, given only the AOD spectra as input, but it would be informative to provide some more details on why you selected this way to make the retrievals (either here or in Section 3).

Now it is shown in Section 3. Initially we have tried with 22-bis size distribution but we rapidly observed that strong smoothness constraints, in terms of size distribution, were required to assure realistic retrievals. That is why we changed our strategy to a

log-normal approximation.

Page 3, lines 11-13: "For instance, AERONET includes around forty direct Sun measurements per day in its standardized sequence of measurements (in cloudless conditions); only about eight of these sequences are coincident with sky-radiance measurements that are suitable as input to the AERONET inversion code." It would be useful to include information on the new Cimels here, where Hybrid scans make sky radiance retrievals possible each hour throughout the day thereby doubling or more the total number of sky-radiance retrievals. Also many new Cimels are run in the turbo mode for AOD measurement where the time interval of AOD sampling is 3 minutes rather than the 15 minutes for the standard scenario in a Cimel sun-sky radiometer. Therefore the âĹij40 measurements per day increases to about 200 AOD spectra per day (depending on latitude and season, therefore day length) with new Cimels.

We have added a sentence including the possibilities of the new instruments.

Page 3, lines 13-14: ". . .a large amount of data containing only direct Sun measurements which are not used apart from the characterization of the aerosol load." I disagree, since the Angstrom Exponent and it's spectral variation are also useful parameters providing basic aerosol size information that are computed by AERONET from the spectral AOD, and are utilized in many studies.

Yes, we agree. We have added: and to obtain some basic aerosol size information computed from its spectral variation

Page 3, lines 14-15: "Moreover, many AERONET sites are plagued by several months of partial cloudiness (especially in wintertime)." I think this is somewhat misleading, since some sites have the seasonal opposite and some tropical sites are cloudy for most seasons. Therefore I suggest dropping the phrase '(especially in wintertime)' from the sentence.

Thanks, removed.

Page 3, lines 14-15: "As a result, the aerosol loads are the only data reported at many sites. . ." Please insert after 'aerosol loads' something like: 'and Angstrom Exponents that parameterize the relative fine versus coarse mode optical influence, depending on the wavelength range used in the computation of AE (Eck et al. 1999; Schuster et al., 2006).'

Thanks, added.

Page 4, lines 11-12: Please include the reference of van Donkelaar et al. (2010).

Thanks, added.

Page 6, lines 4-6: For GRASP-AOD please discuss whether there would be any difference in the retrievals if spherical particle shape assumption was utilized/assumed versus spheroidal shape.

We have added in the inversion strategy description that we assume the sphericity parameter. At the same time we have included the subsection 3.6 Variations on the sphericity parameter, were the effects of this assumption are described.

Page 7, lines 7-9: In most cases this is true, that lognormal bi-modal size distribution assumptions are sufficient. However, for cloud-processed aerosol a third middle mode sometimes exists, see Eck et al. 2012 for Dubovik almucantar retrievals that are tri-modal, not bi-modal, and these middle modes are supported by independent in situ measurements of fog/cloud processed aerosols. This should be mentioned as a relatively rare case that does occur however.

We have added the following paragraph: However, some AERONET retrievals suggest that the particle size distribution is not always perfectly log-normal, as some size distributions are characterized by asymmetrical mode shapes (e.g. Dubovik et al., 2002a; Eck et al., 2005, 2010). Moreover, some size distribution retrievals have a pronounced tri-modal structure, such as observed in volcanic aerosol plumes (Eck et al., 2010) or aerosol retrievals located near fog or cloud (Eck et al., 2012; Li et al., 2014). Obviously,

our strategy that is based upon simplified bimodal size distributions would not provide correct retrievals in these specialized situations.

Page 8, lines 12: Note that Dubovik 2002 presents an AERONET Version 1 database climatology that has been refined significantly with the AERONET Version 2 reprocessing of the entire database that occurred in 2006.

We have modified the paragraph as follows: The numerical tests in this paper are based upon the climatology provided by Dubovik et al. (2002a), which utilizes about 10 years of real aerosol retrievals with AERONET's Version 1 processing. (We note that AERONET has subsequently implemented a Version 2 aerosol retrieval product, but the single-scatter albedo climatology that is based upon this newer processing scheme is within 0.02 of the Dubovik et al. (2002a) climatology for the same aerosol type; Giles et al., 2012).

Page 8, lines 16-17: This is weakly absorbing aerosol at GSFC, non-absorbing aerosol would have SSA=1.

Changed, thank you.

Page 8, lines 17-18: It should be mentioned that this is strongly absorbing biomass burning (BB) aerosol at Mongu, Zambia and that BB aerosol range from very weakly absorbing to strongly absorbing (see Eck et al, 2003, GRL) depending on fuel types and phase of combustion (flaming versus smoldering).

Added, thank you.

Page 8, lines 27-28: Why would you assume that for a mixed case (Bahrain) that all particles are non-spherical? Solar Village also has many mixed mode aerosol days with predominately spherical fine mode particles from pollution.

We use a sphericity parameter of $0$ (i.e., all particles are non-spherical) for all the Solar Village cases except the one with $\tau_a(440) = 0.3$ (SOLV1), which has an Ångström exponent greater than $0.6$ and can not be considered as pure desert dust. We use a

linear approximation with respect to the Ångström exponent to select intermediate values of the sphericity parameter for the three cases with Ångström exponents between $0.6$ and $1.1$ (SOLV1, BAHR2, and BAHR3). That is, we use sphericity parameters of 0 and 100 for Ångström exponents of 0.6 and 1.1 (respectively), and linearly interpolate the intermediate values. The rest of the examples have Ångström exponents greater than $1.1$, so we fixed the sphericity parameter at $100$ (considering all the particles as spheres).

Page 8, lines 12-13: Surprising that you do not show the case of AOD=0.9 in Figure 2 since this is the one where the curvature for the fine mode cases would become the most obvious, since the fine mode radius is largest. Note that the 1640 nm AOD showing a departure from a 2nd order fit of AOD versus WL in Figure 2 is not observed in the real GSFC site AERONET measurement data. Perhaps you can show the 2nd order fit to your simulated data in Figure 2 and also show the delta AOD departures from the fit.

The curvature for case with AOD=0.9 is just a little bit more obvious than for the other two cases and it does not include Lanai case. Therefore, we prefer to keep the selected figures in the work. Note also that the purpose of figure is just to offer a graphical idea of the tendencies from the AOD simulated values. A study of the 2nd order fit and the differences would be interesting but we are aware of the length of the article and this analysis is out of the scope of the work.

Page 9, lines 8-9: Note that the Angstrom Exponents you computed for the Lanai site are much too high, you have 1.22 in Table 2, while the measurement data yield averages of âĹ0.6 to 0.8 for Lanai data (see the AERONET climatology tables). This possibly suggests an issue with the Dubovik 2002 climatology parameters for this site. Also note that for the Solar Village site only the 0.6 and 0.9 AOD cases can be considered dust dominated, since the AOD=0.3 case is mixed with Angstrom Exponent=0.84 in Table 2.

It is true that values of Angstrom Exponents for Lanai site are higher than the data averages in AERONET climatology table. This may be due to an overestimation of the concentration of fine mode in the analysis of Dubovik 2002. At the same point, we would like to note that values of 1.2 are registered and they are not rare at Lanai site (see figure 9 in Smirnov 2001 or figure 1 in Smirnov 2003). On the other hand, we totally agree with the comment regarding Solar Village. In the new version, when we introduce the sphericity parameter, we comment that SOLV1 cannot be studied as pure desert dust and should be considered as mixed desert dust.

Page 13, line 29-31: The estimation of uncertainty for AERONET measured AOD for field instruments is 0.01 in the visible and NIR and increasing to 0.02 in the UV wavelengths (Eck et al. 1999). This spectral variation in AOD uncertainty should be mentioned here or elsewhere in the paper.

We have included it and redone all the tests taking into the account this fact in the covariance matrix.

Page 14, line 8: This is an awkward way to say that uncertainties increase at lower AOD. For the reader it would be much clearer if you did not use abbreviations such as GSFC1 and GSFC3 in the text but instead referred to the AOD levels and associated uncertainties.

Thank you, done.

Page 19, line 5: Should insert the Dubovik et al. (2006) reference here since these significant algorithm advances were incorporated in the AERONET operational inversions in 2006.

Thank you, added.

Page 19, line 6: Please add the reference Smirnov et al. (2000) here for Level 2 data in Version 2.

Thank you, added.

[Figure]

Page 19, line 22-23: Should say that these sites did not have the 1640 nm channel of the newer Cimels, as many readers will not know what you mean by 'extended' photometer.

I have added a reference to table A1.

Page 19, line 31-33: I suggest that an additional, perhaps more robust way to compare the retrievals of the fine mode fraction (FMF) of optical depth is to provide a scatterplot of all individual (not just daily means) GRASP-AOD retrievals regardless of SZA (not just time matched to almucantars) for many sites (including sites not in the Dubovik 2002 table) compared to the AERONET almucantar retrievals of FMF. This should include all AOD levels measured at each site. Both can be plotted as a function of AE(440-870) on the x-axis. Regression fit and statistics can then also be shown. Other parameters such as volume median radius could also be compared as scatterplots using time-matched individual retrievals from many sites, with the AERONET standard sky-radiance inversion on the x-axis and the GRASP-AOD inversion on the y-axis.

We consider that the paper is already quite overloaded, and that the main point of section 4 is to do a first validation with the daily means of retrieved products. As a next step, we would like to do a large validation with AERONET data, with individual (or point by point) studies like the one proposed here. Ideally, we would like to have some members of AERONET staff involved in that future analysis.

Page 20, lines 10-11: Need to mention the likely reason for this large difference for the GSFC-A case. It is the lowest AOD of all sites with AOD(440)=0.166. This is important and needs to be quantified further with cases that have lower AOD values such as 0.10 or lower.

The new analysis in subsection "3.4.2. Low and high aerosol load conditions" does show the large uncertainties of the retrieved parameters when the aerosol load is low. Given the characteristics of actual sun-photometer measurements, and particularly AERONET $\tau - errors$, the quality of the retrieval of the size distribution parameters

can only be assured from $\tau(440) > 0.2, 0.3$.

Page 20, line 13: This is an anomalously large rv fine for a low AOD case at GSFC. Note that the climatology of GSFC size distributions (as a function of AOD) in Eck et al. (2012; Figure 17 b) and in Dubovik et al. (2002) strongly suggest that this case is an anomaly, therefore a relatively poor choice for a case study. The actual values of the almucantar retrieved individual rv fine at GSFC for this date (Nov 22, 2009) range from 0.21 to 0.26 micron (still very high for this AOD level) so I don't know where you got the 0.27 micron number from.

The value of 0.27 micron was a typo error in the text but not in (-old) Table 11 where the value of rv was 0.23 (now in table 14), which agrees with the values pointed out by the referee. On the other hand, we could find a day where fine rv fits better with the AOD values following the climatological functions. However, we reckon that the case is of a great interest since it shows that rv value retrieved by GRASP-AOD is not far from the value obtained by AERONET standard inversion, even thought, it is very different from the "initial guess" used (around 0.138 applying equations in Table 4, which are based on climatology results from Dubovik 2002).

Page 21, lines 17-20: Should note here in the text that differences in the definition of modes, radius cutoff in AERONET standard retrievals versus tails of modes included in SDA explain a least a portion of these differences.

This was written as a footnote and we have kept it

Page 22, lines 9-10: Need to note that the moon measurements have a spectral range from 440 to 1640 nm, while earlier in the paper you made the sensitivity analysis for the 340 nm to 1640 nm spectral range (see Table 2).

Done.

Page 23, lines 20-21: Are you referring to pollution or smoke from biomass burning or a mixture of the two types of aerosols?

I'm referring to only pollution.

Page 24, lines 8-9: Please give the rv for both the fine and coarse modes for the AERONET standard almucantar inversion, for comparison purposes to the GRASP-AOD inverted size distributions.

Added, thanks.

Page 28, lines 2-3: Calibration uncertainty for AERONET master instruments is better than you stated here. The uncertainty in Vo due to calibration by Langley analysis at Mauna Loa is $\sim 0.25\%$ in the visible and NIR and $0.5\%$ in the UV. The resulting total uncertainty in AOD (additionally including some other sources of uncertainty) is $\sim 0.002$ to $0.009$ for Master Cimels (lower values in visible and NIR and higher in UV ; see Eck et al. 1999). This is for overhead sun (SZA=0 and optical airmass=1) and the uncertainty in measured AOD is less as solar zenith angle increases (optical airmass increases).

We have modified it, thank you.

Page 28, lines 8-9: Should cite Smirnov et al. (2000) here since the Version 2 Level 2 data are cloud screened and quality controlled as described in this reference.

Added, thank you.
* * *

---

## Author Comment (AC6) · 19 Jun 2017

Interactive comment on "Advanced characterization of aerosol properties from measurements of spectral optical depth using the GRASP algorithm" by B. Torres et al. F. Dulac (Editor) francois.dulac@cea.fr

I find you methodology interesting and well explained but I wish to question the 0.3-0.9 range of Tau-a(440) values for direct tests: on one side 0.3 is already a significant AOD, significantly larger than the average value in many AERONET stations, especially for places like Lanai where sea-salt is controlling the AOD; conversely, 0.9 does not cover the range of high values regularly observed at stations under desert dust influence;

[Figure]

using such a limited range yields the reader to the conclusion that your algorithm is not consolidated at moderate and very high AOD; I would therefore suggest to include both a lower test value (e.g. 0.1) and a larger value for dust only (e.g. 1.5) to offer a better assessment of your algorithm ran

We have added 4 new aerosol cases: GSFC0 and LANA0 with $\tau(440) = 0.1$ and SOLV4 ZAMB4 with $\tau(440) = 1.5$ (low and large aerosol optical load).
* * *

---

## Author Comment (AC7) · 19 Jun 2017

I have realised that there was a comment without an answer due to an error during the (latex) compilation.

– I agree with the editor and other reviewers that you should add the case of AOD(440 nm)=0.1 to your sensitivity analysis section, since most data in the AERONET network are for AOD levels <0.3. Especially for the Lanai site the AOD=0.3 is much too high to be representative. This is a background marine site where AOD is predominately <0.15. The monthly mean AODs at Lanai do not exceed 0.08 at 440 nm except for the dust transport season in spring when they reach a monthly average maximum of

0.12 in April. I also agree with the editor that it would be useful to include very high AOD cases in your sensitivity analysis (1.5 or 2 at 440 nm) for dust and also fine mode smoke, since these cases are important for analysis of major aerosol events.

We have added 4 new aerosol cases: GSFC0 and LANA0 with $\tau(440) = 0.1$ and SOLV4 ZAMB4 with $\tau(440) = 1.5$ (low and large aerosol optical load). –
* * *

---

## Author Comment (AC8) · 19 Jun 2017

Interactive comment on "Advanced characterization of aerosol properties from measurements of spectral optical depth using the GRASP algorithm" by B. Torres et al. F. Dulac (Editor) francois.dulac@cea.fr Thank you for your submission to the ChArMEx Special Issue of AMT. I am, however, somewhat frustrated that application to ChArMEx data is only one small sub-section at the end of your manuscript. It is indeed of special interest because it refers to original airborne photometric data with the recent PLASMA instrument. But in order to better relate the study to other available ChArMEx data and justify publication as part of the

[Figure]

ChArMEx Special issue, I recommend a more complete link to ChArMEx. On one side, you miss referring to ChArMEx papers that discuss the size distribution and/or absorption properties of Mediterranean aerosols (e.g. Mallet et al., 2013, doi:10.5194/acp-13-9195-2013; Denjean et al., 2016, doi:10.5194/acp-16-1081-2016; see also Fig. 14 in Renard et al., 2016, doi:10.5194/amt-9-1721-2016). On the other side, you might consider a couple of Mediterranean AERONET sites with large data sets in your sensitivity tests. Following Mallet et al. (2013), I could suggest Rome (urban aerosols with relatively low omega-0) and Sede Boker (dusty site with the longest time series in the Med. region). It seems to me that comparisons might also be worth being tested with AERONET inversions at ChArMEx super sites of Ersa and Lampedusa. Other case studies, such as those of July 2012 documented by lidar by Granadoz-Muñoz et al. (2016; doi: 10.5194/acp-16-7043-2016) might also be considered.

The climatology by Dubovik et al. (2002a) provides dynamical models where the parameters defining the aerosol properties (size distribution approximated by a bimodal log-normal function, and complex refractive indices) can be calculated from the values of the aerosol optical thickness. This is not the case in the study Mallet et al. 2013, where the authors give climatological values of the mediterranean sites but they do not define such dynamical models. Therefore, and as discussed in private, to include them in the sensitivity analysis was not possible.

Nevertheless, we were aware that we have submitted the paper in the special issue of ChArMEx. Therefore, the sites of Rome and Lampedusa were included in the analysis in the section 4.1.2

Other technical comments: p.19, line 15: please specify $> 40°$.

Corrected, thank you.

In the bibliography, please replace "n/a-n/a" by the paper number given before the year of publication in several JGR papers (Kaskaoutis et al., 2009; Kim et al., 2004; O'Neil et al., 2003; Schmid et al., 2003; Schuster et al., 2006; Smirnov et al., 2009).

Corrected, thank you.